Resource

# A CRISPRi/a screening platform to study cellular nutrient transport in diverse microenvironments

Christopher Chidley [1] ✉, Alicia M. Darnell[2], Benjamin L. Gaudio[1], Evan C. Lien[2], Anna M. Barbeau[2,3], Matthew G. Vander Heiden [2,3,4] & Peter K. Sorger [1,5] ✉

Blocking the import of nutrients essential for cancer cell proliferation represents a therapeutic opportunity, but it is unclear which transporters to target. Here we report a CRISPR interference/activation screening platform to systematically interrogate the contribution of nutrient transporters to support cancer cell proliferation in environments ranging from standard culture media to tumours. We applied this platform to identify the transporters of amino acids in leukaemia cells and found that amino acid transport involves high bidirectional flux dependent on the microenvironment composition. While investigating the role of transporters in cystine starved cells, we uncovered a role for serotonin uptake in preventing ferroptosis. Finally, we identified transporters essential for cell proliferation in subcutaneous tumours and found that levels of glucose and amino acids can restrain proliferation in that environment. This study establishes a framework for systematically identifying critical cellular nutrient transporters, characterizing their function and exploring how the tumour microenvironment impacts cancer metabolism.

Altered cellular metabolism is a common feature of tumours, enabling cancer cells to maximize proliferation in nutrient-limited environments[1]. Cellular adaptations to the tumour microenvironment (TME) generally include an increase in nutrient uptake, a diversification of uptake mechanisms and a rewiring of metabolism so tumours can more efficiently convert available nutrients into biomass[2]. Amino acids contribute substantially to the formation of cellular biomass[3], and many cancers are dependent on environmental supplies of non-essential amino acids for growth and survival. For example, acute lymphoblastic leukaemias are dependent on asparagine, luminal breast cancers and breast cancer brain metastases exhibit serine auxotrophy, and certain cancers silence arginine biosynthesis or upregulate glutamine metabolism[2,4–7]. Such dependencies can potentially be targeted therapeutically by inhibiting nutrient uptake[8].

Nutrients are actively transported across cell membranes by a large class of transporter proteins called solute carriers (SLCs)[9]. The activity of many SLCs is influenced by the composition of the microenvironment[10]. Studying the function of these SLCs, and identifying potential therapeutic targets, requires characterizing transport under physiologically relevant conditions. Much of our current knowledge derives from experiments in cell-free systems, and many SLC substrates identified in these experiments (hereafter 'annotated' substrates) have not been evaluated in cells or organisms[11]. For example, over 60 SLCs are annotated as amino acid transporters, but it is unclear which are dominant or growth-limiting in cells[11–13]. Recent studies have uncovered the transporters for essential metabolites such as NAD+ (refs. 14–16), choline[17] and glutathione[18,19], but 20–30% of SLCs, including many that are essential in human cells, have no identified function[9,20].

[1]Laboratory of Systems Pharmacology, Harvard Medical School, Boston, MA, USA. [2]Koch Institute for Integrative Cancer Research, Massachusetts Institute of Technology, Cambridge, MA, USA. [3]Department of Biology, Massachusetts Institute of Technology, Cambridge, MA, USA. [4]Department of Medical Oncology, Dana-Farber Cancer Institute, Boston, MA, USA. [5]Department of Systems Biology, Harvard Medical School, Boston, MA, USA. ✉e-mail: christopher_chidley@hms.harvard.edu; peter_k_sorger@hms.harvard.edu

The development of clustered regularly interspaced short palindromic repeats (CRISPR) screening[21] and efforts to better characterize the composition of the TME[22,23] present opportunities to better define the role of nutrient transporters in human cells.

In this Resource, we describe the use of CRISPR interference (CRISPRi) and CRISPR activation (CRISPRa) screening using custom single guide RNA (sgRNA) libraries to systematically interrogate the contributions to nutrient uptake or release of all SLC and ATP-binding cassette (ABC) transporters found in human cells. We screen for transporters responsible for amino acid import in K562 chronic myelogenous leukaemia cells and assess transporter function using mass spectrometry-based transport assays. We also use CRISPRi to identify transporters essential for cell proliferation in standard and physiological growth media, and in subcutaneous tumours in mice. Finally, we use CRISPRa to identify nutrients limiting for proliferation across environments, providing a framework for characterizing cancer-relevant nutrient transporters and how their interaction with the TME shapes metabolism.

## Results

### Cell-based screens for amino acid transporters

To assess transporter function, we employed a dual approach: transporter knockdown (KD) via CRISPRi and overexpression (OE) via CRISPRa. Given previous findings that transporter genes are often top hits in CRISPR screens[21,24–26], we hypothesized that KD/OE would alter substrate transport rates and that such changes could be isolated in growth-based pooled screens (Fig. 1a). Specifically, in conditions where a nutrient limits proliferation, KD of a transporter needed for its import would reduce proliferation and, conversely, OE of a transporter capable of importing that nutrient would increase proliferation. Transporters involved in nutrient export would exhibit opposite phenotypes compared with importers.

To validate this system, we targeted four expressed SLC7 amino acid transporters (SLC7A1, SLC7A5, SLC7A6 and SLC7A8) and three not expressed (SLC7A2, SLC7A3 and SLC7A7) in K562 cells expressing either dCas9-KRAB (CRISPRi) or dCas9-SunTag (CRISPRa). CRISPRi/a of transporters resulted in strong and specific changes in messenger RNA expression (Fig. 1b). SLC7A5, SLC7A6, SLC7A7 and SLC7A8 transporters function as heterodimers with SLC3A2. Quantitative reverse transcription polymerase chain reaction (RT–qPCR) analysis revealed upregulation of *SLC3A2* levels in SLC7A5, SLC7A7 and SLC7A8 CRISPRa cells, consistent with higher levels of functional heterodimers (Fig. 1b). Changes in cell surface protein levels in these cells mirrored RNA-level changes (Fig. 1c). We conclude that CRISPR-based perturbation of SLCs resulted in the anticipated changes in protein levels.

Next, we constructed CRISPRi and CRISPRa sgRNA pooled lentiviral libraries targeting 489 annotated members of the SLC and ABC transporter families (hereafter 'transporters')[20,27,28]; the library includes 10 sgRNAs/gene and 730 non-targeting control (NTC) sgRNAs

(Extended Data Fig. 1a and Supplementary Table 1). The primary function of ABC transporters is to export xenobiotics from cells, but some transporters export metabolites[29,30], suggesting a role in nutrient homeostasis. Libraries of single transporter KD or OE cells were prepared by transduction of K562 CRISPRi or CRISPRa parental lines (Extended Data Fig. 1b).

To enable selection for transporter phenotypes, we identified all amino acids in RPMI-1640 culture medium (RPMI) whose absence would limit proliferation (Fig. 1d). For 13 of the 19 amino acids in RPMI, reduced levels decreased proliferation, indicating net consumption of these amino acids by cells[3]. Asn, Asp, Glu, Gly and Pro removal from the medium did not impact proliferation, precluding transporter identification for these amino acids. K562 cells initially responded to low Ser but proliferation returned to normal, probably due to upregulation of serine biosynthesis enzymes[31]. We were therefore only able to screen for serine transporters under transient and mild growth limitation. Cystine (the oxidized conjugate of Cys most abundant in culture medium) deprivation induced substantial cell death, through ferroptosis[32,33].

We performed pooled CRISPRi/a screens at amino acid concentrations that reduced K562 proliferation by ~50% for all 13 growth-limiting amino acids to identify the dominant and growth-limiting amino acid transporters (Supplementary Tables 2 and 3). Libraries of single transporter KD or OE cells underwent three rounds of low amino acid exposure followed by one recovery day in complete medium (Fig. 1e). Changes in library content were compared with a control screen in complete RPMI medium, and scores were calculated for each transporter[25,34] (Methods).

Amino acid limitation activates the GCN2 starvation response and represses mTOR activity, resulting in changes in transporter gene expression[12]. To assess the extent to which this occurred under screening conditions we used western blotting to assay phosphorylation of mTORC1 and GCN2 targets (Extended Data Fig. 1c). K562 cells cultured in low Arg, Lys or His screening conditions showed mild GCN2 activation and mTOR repression. Refreshing the medium after 1 day incubation relieved those effects in low amino acid conditions, but not in complete starvation conditions (Extended Data Fig. 1c). We investigated if low amino acid conditions altered transporter gene expression; low-Arg and low-Leu screening conditions increased expression of most genes in our SLC7 transporter panel, but not all of them, consistent with results for HEK293T cells in single amino acid dropout media[35]. Critically, however, despite changes in transporter expression caused by screening media, CRISPRi/a of SLC7 genes resulted in additional and specific strong up- or downregulation of mRNA levels and concomitant changes in protein levels at the plasma membrane (Extended Data Figs. 1d,e and 2a). We conclude that while low amino acid screening conditions induce a slight upregulation in the expression of transporters, this does not interfere with the identification of specific phenotypes resulting from transporter CRISPRi/a.

**Fig. 1 | CRISPRi/a screens identify the transporters of amino acids in K562 cells. a**, A cartoon of the general approach used to identify transporters in cells. Individual transporter genes are knocked down via CRISPRi or overexpressed via CRISPRa, and modified transport activity is detected by changes in proliferation. **b**, CRISPRi/a of transporters leads to specific changes in gene expression. Expression levels in K562 CRISPRi/a cells with specific or non-targeting control (NTC) sgRNAs were quantified by RT–qPCR relative to the housekeeping gene GAPDH. Data are mean ± s.e.m of $n = 3$ technical replicates. The horizontal dashed lines represent the average of both NTCs. n.d., not detected. **c**, CRISPRi/a of transporters leads to changes in protein level at the plasma membrane. K562 CRISPRi/a cells were incubated with a cell-impermeable biotinylation reagent, and plasma membrane proteins were isolated by streptavidin affinity purification and analysed by western blotting. **d**, The identification of amino acids that limit proliferation of K562 cells when their level in growth medium is reduced. Data were determined using a luminescent cell viability assay and

represent the average $\log_2$FC relative to time of 0 days (T0) of $n = 4$ biologically independent samples. **e**, A cartoon of the pooled screening strategy used to identify amino acid transporters. Library pools were grown in RPMI media where specific amino acids were present at a level that reduced proliferation by 50% relative to complete RPMI (low amino acid). a.a., amino acid. **f**, Volcano plots of transporter CRISPRi/a screens in K562 cells in low lysine and low arginine. Black circles represent transporter genes, and red circles represent negative control genes. $n = 2$ screen replicates. **g**, Bubble plots displaying CRISPRi screen scores determined for all 64 transporters annotated as capable of amino acid transport[12]. **h**, Same as **g** for CRISPRa screens. In **f–h**, the phenotype scores represent averaged and normalized sgRNA enrichments in low amino acid versus RPMI, and $-\log_{10}(P$ value) was determined using a Mann–Whitney test of sgRNA enrichments compared with all NTC sgRNAs. Source numerical data and unprocessed blots are available in Source data.

## CRISPRi/a screening results

We found that CRISPRi screens in low amino acid conditions mainly found hits with negative phenotype scores, consistent with a role for these transporters in net amino acid import. For all 13 conditions, we identified at least one transporter with significantly depleted sgRNAs (Fig. 1f,g and Extended Data Fig. 2b). Screens with one strong negative hit suggest that a single transporter is responsible for the bulk of the import of the limiting amino acid. For example, we found that SLC7A5

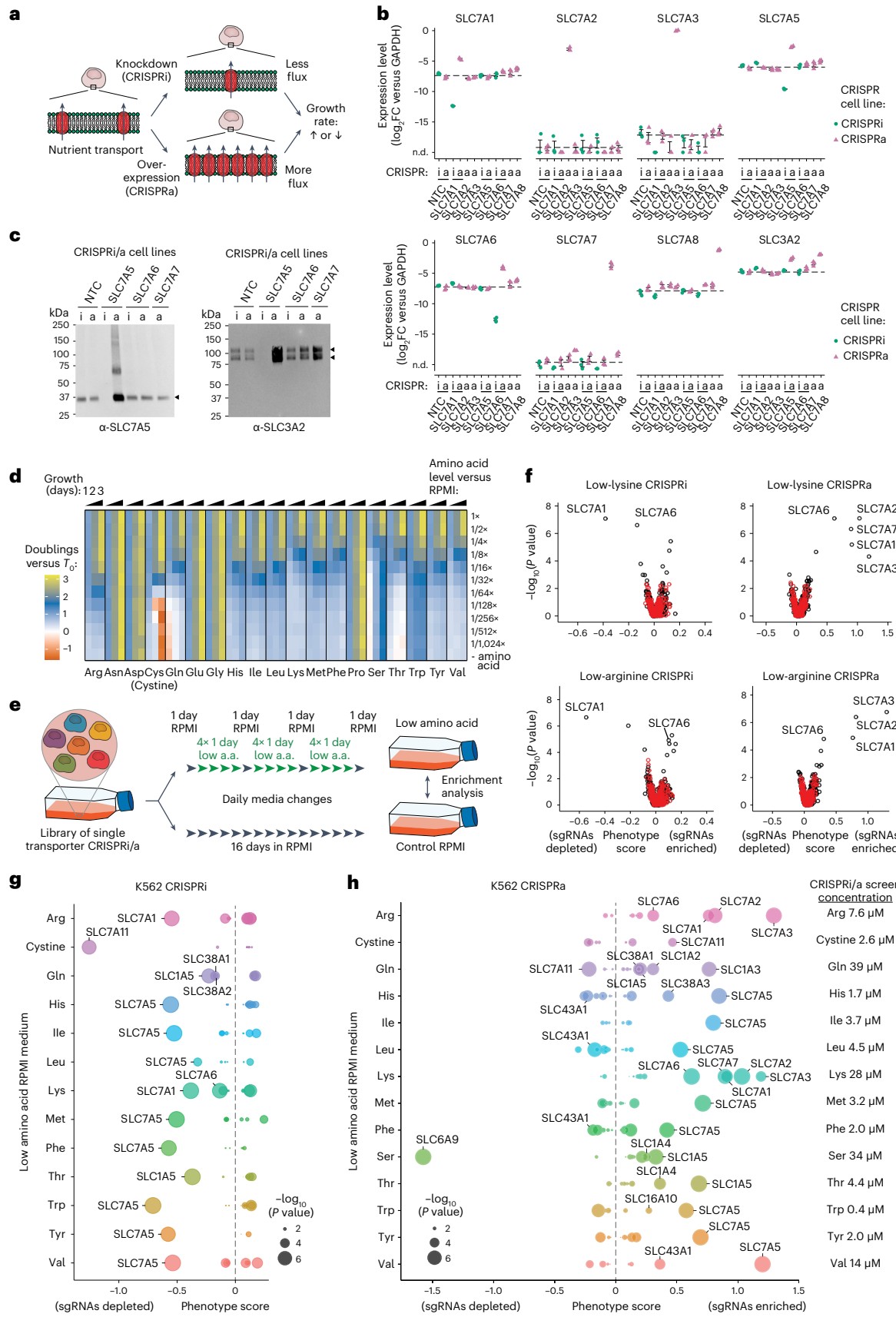

(LAT1)[36] is probably the primary importer for all large neutral amino acids in growth-limited conditions[12,28]. Screens with more than one negative hit suggest a role for multiple partially redundant transporters; this was observed for CRISPRi of SLC1A5, SLC38A1 and SLC38A2 in glutamine-limiting conditions (Fig. 1g).

In contrast, CRISPRa screens in low amino acid conditions primarily yielded hits with positive scores, consistent with increased amino acid import (Fig. 1f,h and Extended Data Fig. 2b). CRISPRa identified more hits than CRISPRi since it queries the role of all transporters, not only the ~50% (ref. 37) that are well expressed in K562 cells. For example, SLC7A1, SLC7A2 and SLC7A3 (CAT1, CAT2 and CAT3), annotated as arginine and lysine transporters, were strong hits in low-Lys and low-Arg CRISPRa screens. However, only SLC7A1 was a hit in those CRISPRi screens, as only SLC7A1 is expressed (Fig. 1b,f). Thus, while SLC7A1 is the primary transporter for Lys and Arg in K562 cells, SLC7A2 and SLC7A3 are also capable of transport. CRISPRi/a screens were reproducible across biological replicates (Extended Data Fig. 2c,d), and strong CRISPRi hits consistently appeared in corresponding CRISPRa screens, indicating that transporter activity is not saturated under screening conditions.

Our parallel screening across many low amino acids conditions enabled us to derive biological insight from negative results for transporters with significant phenotypes in at least one condition, demonstrating that CRISPRi/a was effective. Negative findings can be interpreted as evidence of the transporter's lack of activity for those tested amino acids (Extended Data Fig. 2b).

To delve deeper into transporter phenotypes, we constructed cell lines expressing a single sgRNA identified in screens. Transporter KD/OE was confirmed by RT–qPCR, and growth phenotypes were measured in competition assays (Extended Data Fig. 3a,b and Methods). We validated screens results using seven SLC7 family genes having varying phenotypes in low His, Lys and Arg conditions. The validation closely aligned with screen outcomes, reinforcing the reliability of the approach (Fig. 2a). For example, SLC7A7 OE increased proliferation in low Lys but not in low Arg in both screens and competition assays (Fig. 2a). To explore the potential impact of substrate competition, we reduced Lys levels in RPMI and again observed no phenotype for SLC7A7 OE in low Arg. These data suggest that, at growth-limiting concentrations, SLC7A7 imports Lys but not Arg or any other tested amino acid (Fig. 2a).

Because transporter function can be influenced by the composition of the environment, we tested the effect of medium composition more broadly by assaying CRISPRi/a cells targeting the seven SLC7 genes in a medium we formulated to contain all amino acids and five additional known SLC7 family substrates (citrulline, ornithine, creatine, creatinine and carnitine) at levels found in human plasma (physiological amino acid/PAA-RPMI)[22]. In contrast with PAA-RPMI, RPMI contains many amino acids at 0.4× to 10× the levels found in human plasma and only trace amounts of alanine, cysteine and the five additional SLC7 substrates (Supplementary Table 2). However, in competition assays, we found that RPMI phenotypes were reproduced in PAA-RPMI

(Pearson correlation coefficient 0.94), suggesting that differences in amino acid and other substrate levels between RPMI and human plasma minimally influence SLC7 family phenotypes.

In contrast to other SLC7 transporters, we found no phenotype associated with SLC7A8 (LAT2) KD or OE in RPMI or PAA-RPMI, despite expression comparable to SLC7A5. In addition, *SLC7A8* mRNA levels induced by CRISPRa surpassed those of *SLC7A5* induced by CRISPRa (Fig. 1b and Extended Data Fig. 3c). While SLC7A8 is annotated as an importer for neutral amino acids such as Leu, Ile, His and Phe[12,36], our data suggest that SLC7A8 does not import amino acids in K562 cells under conditions of amino acid limitation.

## CRISPRi/a directly changes amino acid transport rates

We next asked whether screen hits reflect direct changes to transport of the limiting amino acid by the transporter targeted by CRISPRi/a. We quantified amino acid import by incubating K562 cells in medium containing 16 heavy-labelled amino acids and measuring intracellular isotope accumulation over time using gas chromatography–mass spectrometry (GC–MS) (Methods, Fig. 2b and Extended Data Fig. 3d). As many transporters function as exchangers, mediating both import and export of amino acids[10,12], we also determined net transport across the plasma membrane by measuring amino acid consumption from the culture medium (Extended Data Fig. 3d). We determined absolute import and net transport rates using an external standard curve for all 20 amino acids[23] (Fig. 2c and Extended Data Fig. 3e–g).

We found that, while most amino acids were consumed from the medium, five (Asn, Asp, Glu, Gly and Pro) were secreted, consistent with previous findings in non-small cell lung cancer cells[3], and our observation that removing these amino acids from RPMI did not impact proliferation (Figs. 1d and 2c). For 9 out of the 11 amino acids consumed by K562 cells, import exceeded net transport by more than sevenfold, indicating high bidirectional flux (Extended Data Fig. 4a). We also measured intracellular free amino acid levels and extrapolated free amino acid pool turnover rates by dividing import rates by intracellular levels. We found that most amino acids were turned over in minutes, further demonstrating high flux in cells (Fig. 2d).

Using transport assays, we assessed the impact of SLC7A5 KD on amino acid transport rates. SLC7A5, required for tumour growth in many settings[38], showed no proliferation defect in CRISPRi screens in RPMI but a strong defect in a medium low in His, Ile, Leu, Met, Phe, Trp, Tyr or Val (Fig. 2e). We observed a large reduction in the import rates of these amino acids in either RPMI or low Leu (Fig. 2f), whereas import of all other amino acids was unchanged (Extended Data Fig. 4b). Consistent with an absence of proliferation phenotype for SLC7A5 KD in RPMI, overall amino acid consumption rates were unchanged (Fig. 2g). Despite a strong reduction in neutral amino acid transport caused by SLC7A5 KD in RPMI, import rates still exceeded consumption rates (Fig. 2f,g). Thus, a strong perturbation to transport rates does not necessarily result in a growth defect.

In low-Leu medium, however, Leu import rates fell below RPMI consumption rates in control cells and were even lower in SLC7A5

**Fig. 2 | Measurement of amino acid transport rates in K562 cells reveals that CRISPRi of SLC7A5 specifically reduces transport of large neutral amino acids. a**, The phenotype scores obtained in transporter CRISPRi/a screens for SLC7 family genes were validated in competition assays in K562 cells in RPMI and in RPMI with amino acids adjusted to human plasma levels (PAA-RPMI). **b**, The import of amino acids into K562 cells in RPMI was determined by quantifying the intracellular accumulation of stable heavy-isotope-labelled amino acids over time by GC–MS. One example representative of six independent experiments. **c**, Amino acid import and cellular consumption rates of K562 cells growing in RPMI. Import rates were determined by linear regression of the early phase of heavy-isotope-labelled amino acid accumulation. Consumption rates were determined by linear regression of amino acid levels in the growth medium of K562 cells over time. *n* = 6 biologically independent samples for import and *n* = 5 for consumption. Data are mean ± s.e.m. **d**, Intracellular amino acid levels

(*n* = 7 biologically independent samples) and pool turnover rates of K562 cells growing in RPMI. Pool turnover rates were inferred by dividing amino acid import rates in **c** by intracellular levels. Data are mean ± s.e.m. **e**, Screen scores for K562 SLC7A5 CRISPRi/a. **f**, Amino acid import rates for K562 SLC7A5 and non-targeting control (NTC) CRISPRi in RPMI and in low leucine. Data represent the slope ± SE determined from the linear regression of *n* = 7 biologically independent samples. **g**, Cellular consumption rates for K562 SLC7A5 and NTC CRISPRi in RPMI. **h**, Consumption of leucine from low-leucine medium by K562 SLC7A5 and NTC CRISPRi cells. In **g** and **h**, the data represent the slope ± SE determined from the linear regression of *n* = 6 biologically independent samples. The shaded area represents 95% confidence interval. **i**, Intracellular amino acid levels of K562 SLC7A5 and NTC CRISPRi cells growing in RPMI or in low leucine (*n* = 8 biologically independent samples; data are mean ± s.e.m.). Source numerical data are available in Source data.

KD cells, consistent with the reduced proliferation imposed by the low-amino-acid medium and the strong additional growth defect imposed by KD of SLC7A5 (Fig. 2f,g). Specifically, Leu consumption rates in low-Leu medium were 6-fold lower in control cells and 14-fold lower in SLC7A5 KD cells compared with Leu consumption in RPMI

(Fig. 2h). Moreover, reduced import of SLC7A5 substrates correlated with lower intracellular levels in both RPMI and low Leu (Fig. 2i and Extended Data Fig. 4c). Of note, import of SLC7A5 substrates other than Leu was higher in low Leu than in RPMI (Fig. 2f and Extended Data Fig. 4d,e), revealing competition among multiple substrates for

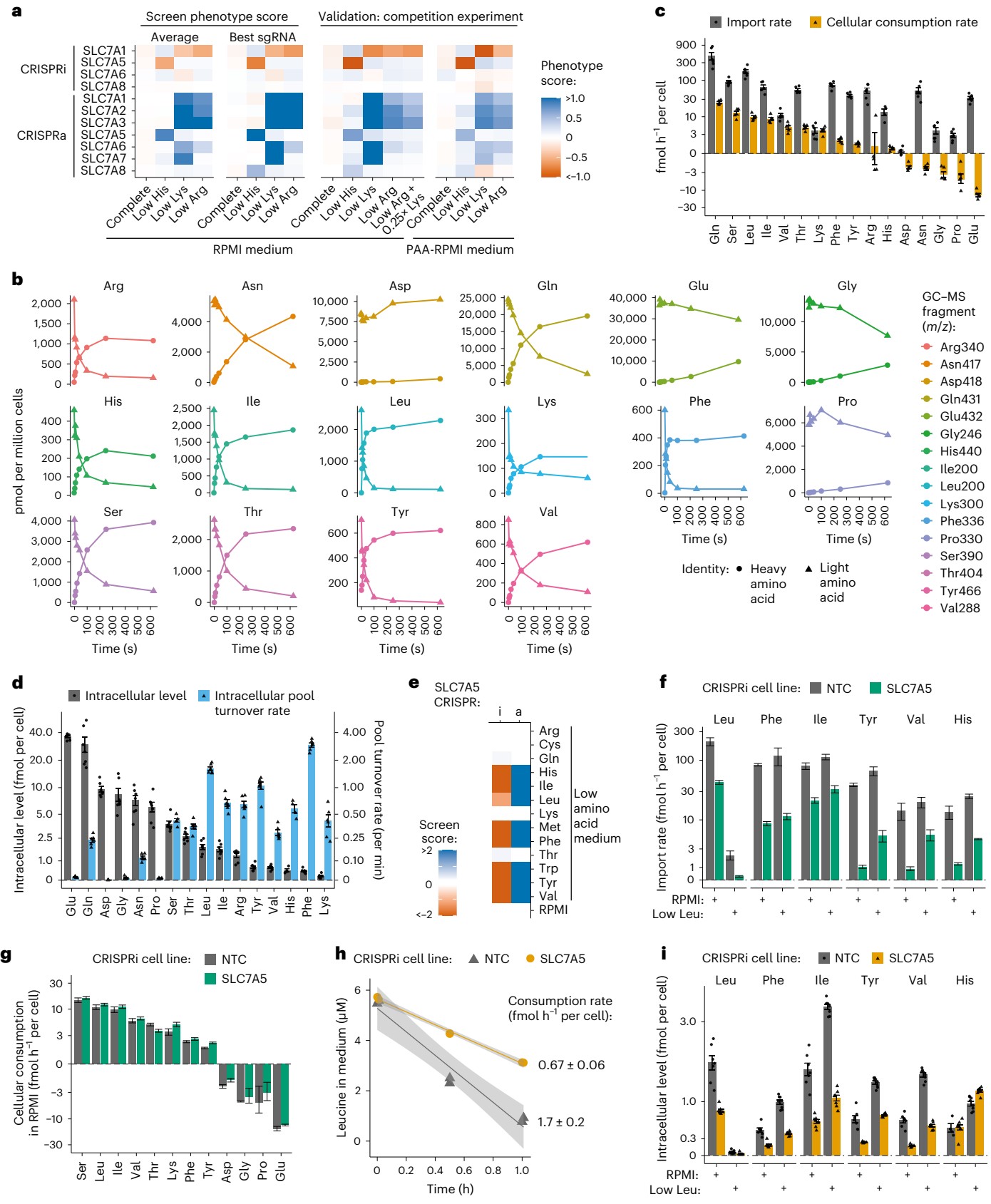

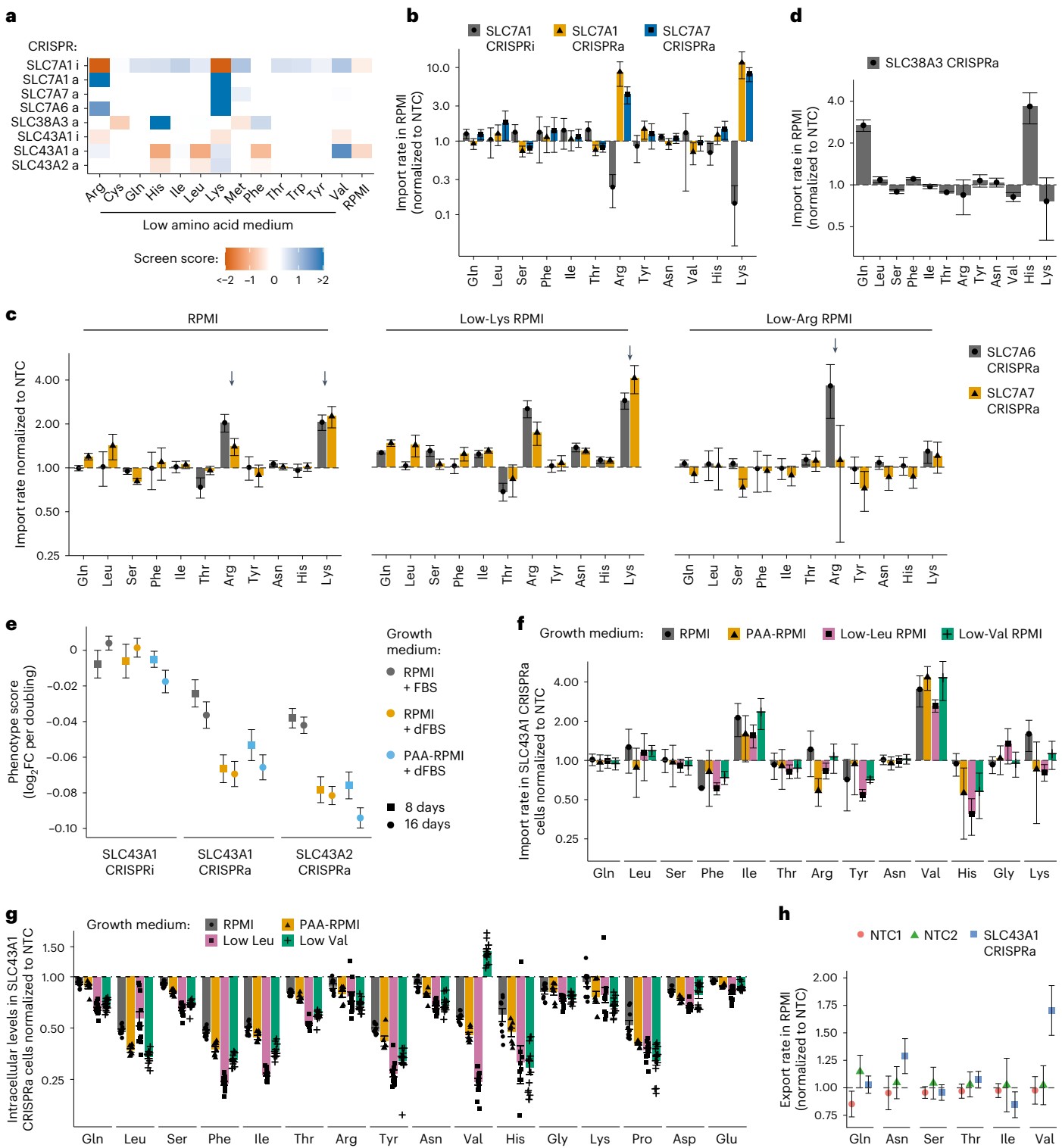

**Fig. 3 | SLC43A1/LAT3 is a net exporter of large neutral amino acids. a**, Scores obtained in CRISPRi/a screens in low amino acid and in RPMI. **b**, SLC7A1 CRISPRi/a and SLC7A7 CRISPRa specifically alter the import of arginine and lysine in K562 cells in RPMI. **c**, SLC7A7 CRISPRa increases the import of lysine when lysine is present in the medium at growth-limiting concentrations. SLC7A6 CRISPRa increases the import of arginine and lysine in all conditions tested. The arrows highlight the import of Arg and Lys in RPMI, import of Lys in low-Lys conditions and import of Arg in low-Arg conditions. **d**, SLC38A3 CRISPRa increases import of histidine and glutamine into K562 cells in RPMI. **e**, CRISPRa of SLC43A1 and SLC43A2 induces a proliferation defect in K562 over a range of conditions (RPMI with regular FBS, RPMI with dialyzed FBS (dFBS), and RPMI modified such that

amino acids match human plasma levels (PAA-RPMI)). Data are the mean ± s.e.m. of two biological replicates each with six technical replicates. **f**, CRISPRa of SLC43A1 increases import of isoleucine and valine into K562 cells. **g**, CRISPRa of SLC43A1 decreases intracellular levels of large neutral amino acids in K562 cells. Levels were determined from import assays in **f** (*n* = 7 biologically independent samples; data are mean ± s.e.m.). **h**, CRISPRa of SLC43A1 increases the export rate of valine from K562 cells cultured in RPMI. In **b**–**d**,**f** and **h**, the rates were determined from a linear regression of *n* = 6 biologically independent samples. The data represent the slope ± SE normalized to a non-targeting control (NTC). Source numerical data are available in Source data.

import by a single transporter. Overall, these results show that SLC7A5 KD reduces import of its amino acid substrates, impacting cell proliferation when the import of a specific amino acid falls below normal consumption rates. We conclude that many of our CRISPRi/a screen phenotypes reflect direct changes to transport of the limiting amino acid by the targeted transporter gene.

### SLC7A6 and SLC7A7 can scavenge cationic amino acids

While SLC7A1, SLC7A2 and SLC7A3 use membrane potential to drive import of Lys and Arg, the exchangers SLC7A6 and SLC7A7 are thought to export Arg and Lys in physiological conditions[12,36]. We characterized SLC7A6 and SLC7A7 phenotypes using transport assays (Figs. 2a and 3a). Surprisingly, we found that SLC7A7 OE conferred an advantage in low Lys, and SLC7A6 OE conferred an advantage in low Lys and low Arg, suggesting a potential role for these proteins as net importers. We measured amino acid import in RPMI and observed that OE of SLC7A6 or SLC7A7 increased Lys and Arg import (Fig. 3b,c). The increase in import for SLC7A7 OE was similar to that for SLC7A1 OE (four- to eightfold), the primary importer of these amino acids in K562 cells (Fig. 3b and Extended Data Fig. 5a). In screening conditions, SLC7A6 OE increased Arg and Lys import, while SLC7A7 OE increased Lys, but not Arg, import, consistent with screen scores (Fig. 3c). These results show that SLC7A6 and SLC7A7 are strong cationic amino acid importers and highlight differences in Arg affinity, with SLC7A7 emerging as the lower-affinity Arg transporter.

### SLC38A3 is a high-affinity histidine transporter

SLC38A3 (SNAT3) can transport His and Asn but is primarily considered a Gln transporter, resulting in net import or export depending on environmental conditions[39]. However, we found that OE of SLC38A3 conferred a proliferation advantage in low His but not low Gln (Fig. 3a). When we measured amino acid import in RPMI, we observed an increase for both His and Gln (Fig. 3d and Extended Data Fig. 5b). These data suggest that SLC38A3 functions as a high-affinity His and low-affinity Gln transporter in K562 cells.

### SLC43A1 is a net exporter of large neutral amino acids

In addition to SLC7A5 and SLC7A8, the L-type amino acid transporter (LAT) family includes SLC43A1 (LAT3) and SLC43A2 (LAT4), annotated as low-affinity facilitated diffusers of neutral amino acids Leu, Phe, Ile, Met and Val[10,40,41]. We found that SLC43A1/A2 OE reduced proliferation in RPMI, and SLC43A1 OE conferred resistance to low Val but hypersensitivity to low His, Leu and Phe, indicating varying affinities for structurally similar amino acids (Fig. 3a). These phenotypes suggest that SLC43A1 and SLC43A2 function as net exporters of amino acids in most conditions. We confirmed proliferation defects in RPMI and other complete media and also observed that SLC43A1 OE increased Ile import by 1.6–2.4-fold and Val import by 2.6–4.4-fold with minimal changes for other amino acids, in RPMI, PAA-RPMI, low-Leu and low-Val media (Fig. 3e,f and Extended Data Fig. 5c–e). Despite increased import, intracellular Ile and Val levels were substantially reduced in SLC43A1 OE cells across conditions, as were the intracellular levels of other neutral amino acids, which fell by at least 50% in all four media (Fig. 3g). An exception to this trend was an increase in intracellular Val in low Val conditions, consistent with the screen resistance phenotype (Fig. 3a,g). SLC43A2 OE caused similar changes to amino acid import and levels (Extended Data Fig. 5f,g). To test the hypothesis that SLC43A1 and SLC43A2 act as net exporters of neutral amino acids, we measured amino acid export in RPMI and found that export of Val was specifically increased upon SLC43A1 OE (Fig. 3h). Despite reduced intracellular levels, export of Leu, Ile and Phe in SLC43A1 OE was similar to control, suggesting a higher intrinsic export capacity (Extended Data Fig. 5h). Overall these data show that SLC43A1 and SLC43A2 primarily facilitate neutral amino acid export but are capable of net importing Ile and Val in low Ile/Val conditions. These results illustrate the environment's

influence on transport and the phenotypic complexity arising from bidirectional transport, underscoring the importance of quantifying both import and export flux to determine net transporter activity.

### Serotonin is an anti-ferroptosis endogenous antioxidant

CRISPRi/a screens in low amino acid conditions also reveal indirect effects, such as changes in the demand for the limiting amino acid or in the cell's tolerance to the low amino acid environment. For instance, OE of the Gly transporter SLC6A9 caused hypersensitivity to low Ser, as Gly-to-Ser conversion depletes the one-carbon pool (Fig. 1h)[42]. Also, OE of the Glu/Asp transporter SLC1A2 caused resistance to low Gln, as it reduced demand for exogenous Gln, consistent with previous observations (Extended Data Fig. 6a,b)[35,43,44].

In K562 cells, low cystine was unique because it caused cell death (Fig. 1d), probably via glutathione depletion and consequent induction of ferroptosis[32,33]. We hypothesized that screens performed in low cystine would also yield hits related to ferroptosis sensitivity. SLC7A11, the primary cystine importer in animal cells, was one of the strongest hits (Fig. 4a)[45]. Iron transporters exhibited robust phenotypes in the CRISPRi screen, consistent with the role of iron in enabling ferroptosis. KD of ABCC1, a major multidrug efflux pump, conferred resistance, probably by reducing intracellular glutathione efflux[46], suggesting that ABCC1 OE may render cells susceptible to ferroptosis, a potentially druggable vulnerability (Fig. 4a).

Unexpectedly, we found that overexpressing SLC6A4, the serotonin transporter, conferred resistance to low cystine (Fig. 4a). SLC6A4 is poorly expressed in K562 cells whereas SLC7A11 is well expressed, and CRISPRa induced strong and specific OE (Extended Data Fig. 6c). We validated the resistance of SLC6A4 and SLC7A11 OE in screen-like competition assays and observed that addition of fluoxetine, a selective serotonin reuptake inhibitor (SSRI)[47], to block serotonin transport by SLC6A4, reduced the resistance of SLC6A4 OE by at least 50% while addition of exogenous serotonin increased resistance (Fig. 4b). The same effects were observed in short-term viability assays in single cell lines (Fig. 4c,d). Thus, transport of serotonin into cells by SLC6A4 seems to underlie ferroptosis resistance. While not part of the RPMI formula, serotonin can be generated via oxidative degradation of Trp[48]. Addition of 5-hydroxytryptophan (5-OH Trp), the metabolic intermediate in the conversion of Trp into serotonin, had no effect on viability, suggesting it was either not protective or not imported (Fig. 4b,d). We hypothesize that SLC6A4 OE confers an advantage in low cystine by increasing the transport of low levels of serotonin present in the culture medium and consequent inhibition of ferroptosis.

To understand serotonin's impact in the ferroptosis pathway, we tested its ability to mitigate ferroptosis in the presence of two inducers targeting different steps: erastin (SLC7A11 inhibitor), and RSL3 (glutathione peroxidase 4 (GPX4) inhibitor). Exposure of K562 cells to RSL3 alone or RSL3 plus erastin in low cystine did not prevent rescue by serotonin present in the medium or added exogenously, indicating serotonin acts downstream of GPX4 (Fig. 4e). Similar results were obtained in the BRAF[V600E] melanoma cell line A375 (Fig. 4f). Serotonin has affinity for unsaturated lipid bilayers and its indole ring has strong antioxidant activity[49,50]. We therefore measured lipid peroxide levels and found that serotonin almost completely abrogated the strong increase in peroxide levels caused by medium lacking cystine and containing RSL3; this effect was dependent on SLC6A4 OE (Fig. 4g and Extended Data Fig. 6d). At high serotonin concentrations, rescue was independent of SLC6A4, probably due to other import mechanisms or passive diffusion.

K562 and A375 cells, which express SLC6A4 poorly, require SLC6A4 OE or high serotonin concentrations for anti-ferroptosis activity. We tested ferroptosis protection by serotonin via SLC6A4 in intestinal Caco-2 cells, which express high levels of SLC6A4 (Extended Data Fig. 6c). In Caco-2 cells, serotonin addition to low-cystine medium prevented cell death and KD of SLC6A4 blocked rescue (Fig. 4h). Rescue

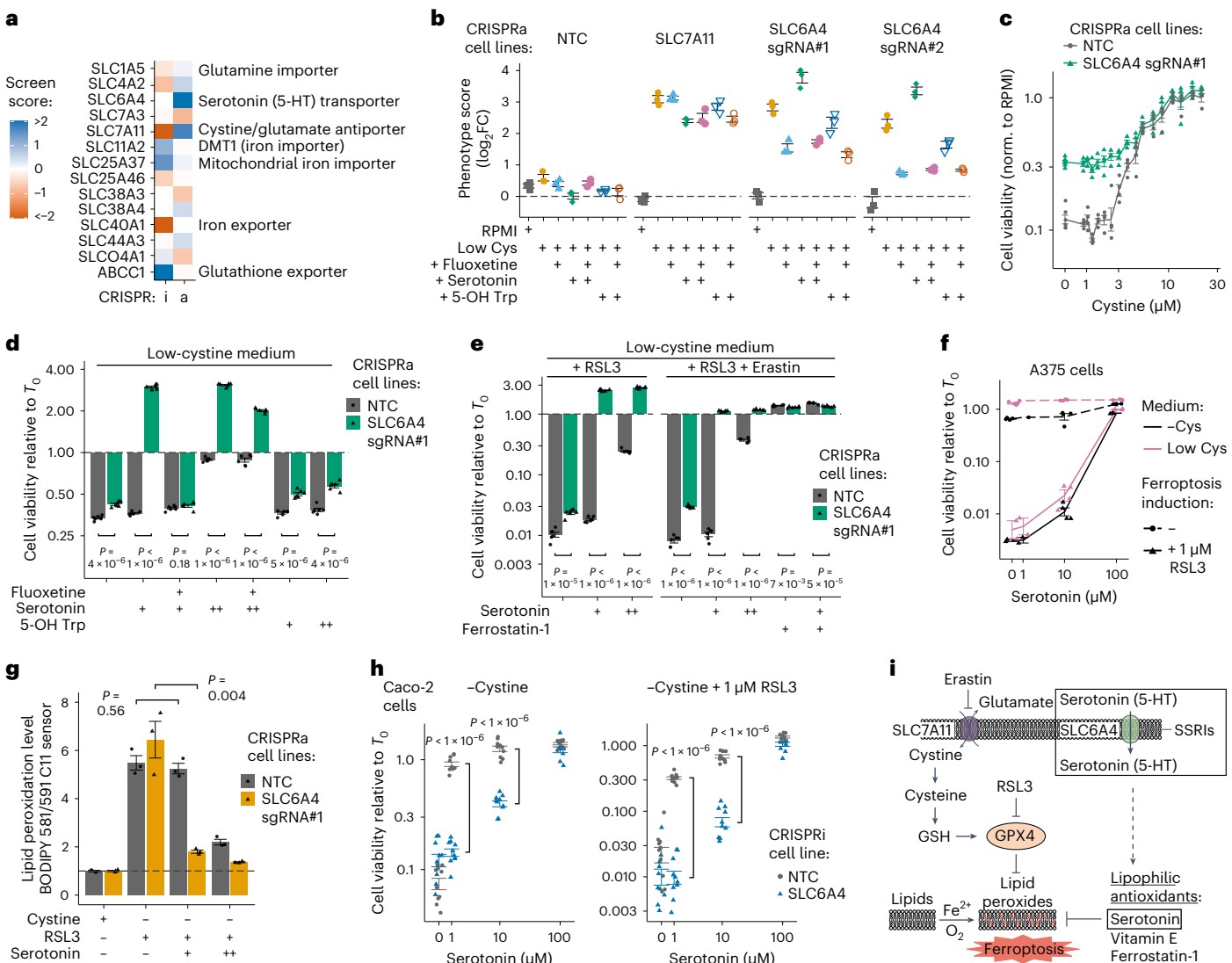

**Fig. 4 | Serotonin imported by SLC6A4 protects cells from ferroptosis via endogenous antioxidant activity. a**, Scores obtained in transporter CRISPRi/a screens in K562 cells in low-cystine medium. **b**, SLC6A4 CRISPRa provides a growth advantage in low cystine dependent on SLC6A4 activity. Phenotype scores were determined in competition assays in low-cystine medium replicating screen conditions and represent the average log₂FC between a specific and non-targeting control (NTC) sgRNA. Fluoxetine, 10 μM; serotonin, 1 μM; 5-OH Trp, 1 μM. $n$ = 3 biologically independent samples. **c**, SLC6A4 CRISPRa provides resistance to K562 cells over a range of cystine levels. Cell viability was quantified in a luminescent assay after 48 h in low cystine. $n$ = 4 biologically independent samples. **d**, Serotonin protects K562 cells from death in low cystine and expression of SLC6A4 increases protection. Assay as in **c**, and viability was normalized to the initial cell count (T0). Fluoxetine, 5 μM; serotonin: 1 (+) or 10 (++) μM; 5-OH Trp, 1 (+) or 10 (++) μM. $n$ = 6 biologically independent samples. **e**, Serotonin protects K562 cells from death independent of GPX4 activity. Assay

as in **d**. Ferrostatin-1, 0.1 μM; RSL3, 1 μM; erastin, 5 μM. **f**, Serotonin protects A375 cells from ferroptosis only at high concentration. Assay as in **d**. $n$ = 4 biologically independent samples. **g**, Addition of serotonin (1 (+) or 10 (++) μM) reduces lipid peroxide levels in K562 cells induced for ferroptosis, dependent on SLC6A4 expression. Each data point represents the average of 20,000 cells determined by flow cytometry. $n$ = 3 biologically independent samples. Peroxide levels were normalized to the '+Cys' condition represented by the horizontal dashed line (normalized to a value of 1). **h**, Serotonin protects Caco-2 cells from ferroptosis dependent on SLC6A4 expression and independent of GPX4 activity. Assay as in **d**. $n$ = 10 biologically independent samples. **i**, A cartoon illustrating the role of serotonin and SLC6A4 in suppressing ferroptosis. Boxed areas are from this study; core ferroptosis pathway components and ferroptosis-modulating molecules were adapted from others[33,61]. In **d**–**f** and **h**, $P$ values were determined using two-tailed unpaired Student's $t$-tests. In **b**–**h**, data are mean ± s.e.m. Source numerical data are available in Source data.

occurred downstream of GPX4, as demonstrated by RSL3 independence. Low-cystine conditions did not induce SLC6A4 expression in K562 and Caco-2 cells, in contrast to SLC7A11 (Extended Data Fig. 6e). These data demonstrate that serotonin suppresses ferroptosis by quenching lipid peroxides and expression of SLC6A4 substantially potentiates its effect (Fig. 4i).

**Transporter essentiality is highly condition specific**
We next assessed transporter essentiality using CRISPRi screening across a range of conditions in which growth was not deliberately

nutrient limited. We hypothesized that condition-specific transporter phenotypes would not only shed light on transporter function but also provide insights into cellular metabolism and environmental effects, as transport flux is influenced by cellular demands and extracellular nutrient levels. First, CRISPRi screens in K562 cells cultured in complete RPMI revealed that about 9% of all KDs were significantly and reproducibly depleted (46 essential transporters) (Fig. 5a,b and Extended Data Fig. 7a–c). We measured essentiality across three complete synthetic media and found that the majority of transporters exhibited condition-specific phenotypes (Fig. 5c). For example,

the folate transporter SLC19A1 was essential only in RPMI at low cell density or with dialysed foetal bovine serum (dFBS). Additionally, under high-density growth conditions, KD of the choline importer FLVCR1 (SLC49A1) and the bicarbonate transporter SLC4A7 resulted in strong defects while KD of the mitochondrial citrate transporter SLC25A1 conferred an advantage[17,28]. In contrast, a minority of KD phenotypes (for example, ABCE1, SLC25A10 and SLC39A9) were insensitive to changes in extracellular nutrient levels (Fig. 5c,d).

We next identified the nutrients causing condition-specific phenotypes in complementation assays. For example, comparing essentiality in cells cultured in RPMI with dFBS versus regular foetal bovine serum (FBS) revealed three transporters (SLC39A10, MPC1 and MPC2) with markedly different phenotypes (Fig. 5d). SLC39A10 KD conferred a strong defect in dFBS but not in FBS, unlike KD of the closely related gene SLC39A9, which showed consistent deleterious effects in both conditions. SLC39 transporters are annotated as zinc importers, although some members import other metal ions such as cadmium and manganese[51]. Since dialysis removes small molecules and ions, and RPMI lacks transition metals, we hypothesized that SLC39A10 imports one or more essential metals for K562 cells. In contrast, SLC39A9 is probably buffered from extracellular metal ion concentration changes due to its localization in the Golgi[51]. We added specific metal ions to RPMI with dFBS and observed that zinc rescued the defect of SLC39A10 KD and manganese exacerbated it (Fig. 5e and Extended Data Fig. 7d). These results suggest that SLC39A10 is a major zinc importer in K562 cells and that manganese can compete for import.

KD of MPC1 and MPC2, the two subunits of the mitochondrial pyruvate carrier (MPC) that transports cytoplasmic pyruvate into the mitochondria (Fig. 5f), was also selectively detrimental in RPMI with dFBS. The defect of MPC KD in dFBS was rescued by addition of alanine (Fig. 5g and Extended Data Fig. 7e), consistent with previous observations showing that MPC knockouts are deleterious in low-alanine conditions because mitochondrial pyruvate is required to synthesize alanine[52]. Lactate and pyruvate, present in FBS but removed by dialysis (Extended Data Fig. 7f), displayed differential effects when added to MPC KD cells. Lactate exacerbated proliferation defects, whereas pyruvate alleviated them. In addition, supplementation with either pyruvate or alanine rescued defects caused by lactate (Fig. 5g and Extended Data Fig. 7g). Given lactate and pyruvate's impact on the $NAD^+/NADH$ ratio, we hypothesized that growth defects were caused by a reduction in the $NAD^+/NADH$ ratio[53]. Supplementing media with α-ketobutyrate (AKB), which increases the $NAD^+/NADH$ ratio without contribution to the tricarboxylic acid (TCA) cycle or to ATP generation[54], rescued defects caused by MPC KD (Fig. 5g). These results are consistent with mitochondrial pyruvate being required for alanine synthesis and $NAD^+$ regeneration.

We noted cell-type-specific transporter requirements by comparing transporter essentialities in K562 and A375 cells, in which CRISPRi/a is highly effective[55]. While many transporters showed similar essentiality between cell lines, 17 transporters exhibited strong cell-line-specific phenotypes (Fig. 5h). For example, KD of the mitochondrial serine transporter SFXN1 (ref. 56) was deleterious in K562 cells but had no effect in A375 cells, despite the KD being similarly effective in the two cell lines (Extended Data Fig. 7h). These findings suggest that K562 is unable to compensate for the loss of mitochondrial one-carbon metabolism by activating the cytosolic counterpart[57] and highlight the potential of comparing transporter essentiality to reveal cell-specific changes in metabolism.

## Transporter CRISPRi/a screens in subcutaneous tumours

Given our findings that transporters are highly sensitive to the environment, we performed screens in xenografted tumours as a more physiological environment. We performed CRISPRi and CRISPRa transporter screens in K562 and A375 cells growing in subcutaneous tumours in immunodeficient mice. To ensure sufficient library coverage, we first determined engraftment efficiencies using green fluorescent protein (GFP)-labelled cells (Extended Data Fig. 8a,b). Across all screens we identified transporter perturbations affecting proliferation in subcutaneous tumours (Fig. 6a,b). To discern in vivo-specific effects, we screened in RPMI and also in human plasma-like medium (HPLM)[22] and adult bovine serum (ABS), which approximate blood nutrient levels[45] (Fig. 6c,d). We quantified similarity across environments and found that CRISPRi phenotypes in HPLM and RPMI were strongly correlated (Pearson's $r > 0.9$), and both of these had weaker but similar correlations to xenografts ($r = 0.7$–$0.8$) (Fig. 6e,f). CRISPRa correlations were generally lower, suggesting greater environmental dependency. In K562, xenograft phenotypes correlated more strongly with those in HPLM ($r = 0.7$) than in RPMI ($r = 0.5$), suggesting HPLM mimics the tumour environment better than RPMI[23], and ABS scores poorly correlated with other conditions.

These results highlight nutrient dependencies and limitations in subcutaneous tumours and how they change across environments. For example, KD of the main glucose importer, SLC2A1, was deleterious in A375 cells in all tested conditions, while OE enhanced tumour growth. In contrast, SLC2A1 KD in K562 cells showed no phenotype, and OE was deleterious in tumours and RPMI. We observed comparable SLC2A1 expression levels at baseline in the two cell lines, and OE was similarly effective at increasing transcript (approximately tenfold) and plasma membrane protein levels (Extended Data Fig. 8c,d), indicating that these differences probably stem from changes in glucose metabolism rather than in SLC2A1 activity, and a higher glucose demand in A375 cells is consistent with reports of increased glycolysis due to the $BRAF^{V600E}$ oncogene[58]. In a second example, the defects induced by either SLC6A9 OE, SLC19A1 KD or SFXN1 KD in K562 tumours suggest that maintaining the one-carbon pool is especially critical in the subcutaneous environment, as these transport perturbations all impair the folate cycle[56].

Our findings in low-amino-acid media show that OE of nutrient transporters confers a growth advantage when the substrate is limiting proliferation. For instance, the advantage conferred by SLC2A1 OE in

**Fig. 5 | Transporter essentiality is highly condition specific, which can be leveraged to decipher transporter function. a**, Essential transporters in K562 cells growing in RPMI determined using CRISPRi screening. $n = 2$ screens each with two technical replicates. $P$ values were determined using a Mann–Whitney test. **b**, A chord plot displaying all essential transporters from **a**. **c**, A tile plot of growth scores determined in CRISPRi screens in K562 cells across culture conditions ($n = 2$ replicates). All transporters significantly enriched or depleted in at least one condition are included. '#' highlights transporters discussed in the main text. **d**, A comparison of essentiality between conditions. Data represent growth scores for all transporters as determined in **c**. **e**, Growth complementation assays identifying SLC39A10 as the main zinc importer in K562 cells and manganese as a competing ion. Phenotypes were determined in RPMI + FBS, in RPMI + dFBS or in RPMI + dFBS supplemented with 10 µM metal ion. Data are mean ± s.e.m. of three technical replicates for non-targeting control (NTC) and three technical replicates of two biological replicates for SLC39A9 and

SLC39A10. **f**, A cartoon illustrating the role of the mitochondrial pyruvate carrier (MPC1/MPC2) and the major sources and uses of cytoplasmic and mitochondrial pyruvate. Exogenous AKB was used to restore $NAD^+$ levels. ETC, electron transport chain. **g**, The addition of either alanine or of molecules restoring $NAD^+$ levels alleviates the defect of MPC1/2 CRISPRi. Phenotypes were determined in competition assays in RPMI + FBS and in RPMI + dFBS with 130 µM alanine, 1 mM AKB, 1 mM pyruvate, 4.5 mM lactate or 130 µM alanine + 4.5 mM lactate. Data are mean ± s.e.m. of two biological replicates each with three technical replicates. Horizontal dashed lines highlight the growth phenotype in the '+dFBS' condition. **h**, A comparison of transporter essentiality in K562 and A375 cells. Data ($n = 2$ replicates) are the phenotype scores determined in CRISPRi screens in RPMI for all transporters significantly depleted in at least one cell line. In **a** and **d**, black circles indicate transporter genes and red circles indicate negative control genes. In **e** and **g**, $P$ values were determined using two-tailed unpaired Student's $t$-tests. Source numerical data are available in Source data.

A375 tumours indicates that glucose is probably limiting proliferation of A375 cells in that environment (Fig. 6d). Additionally, the advantage conferred by OE of amino acid transporters SLC7A3, SLC38A2 and SLC7A11 suggests that Arg, Gln and Cys levels are also limiting proliferation. Conversely, no growth advantage from transporter OE was observed in K562 tumours, suggesting a lack of nutrient limitations.

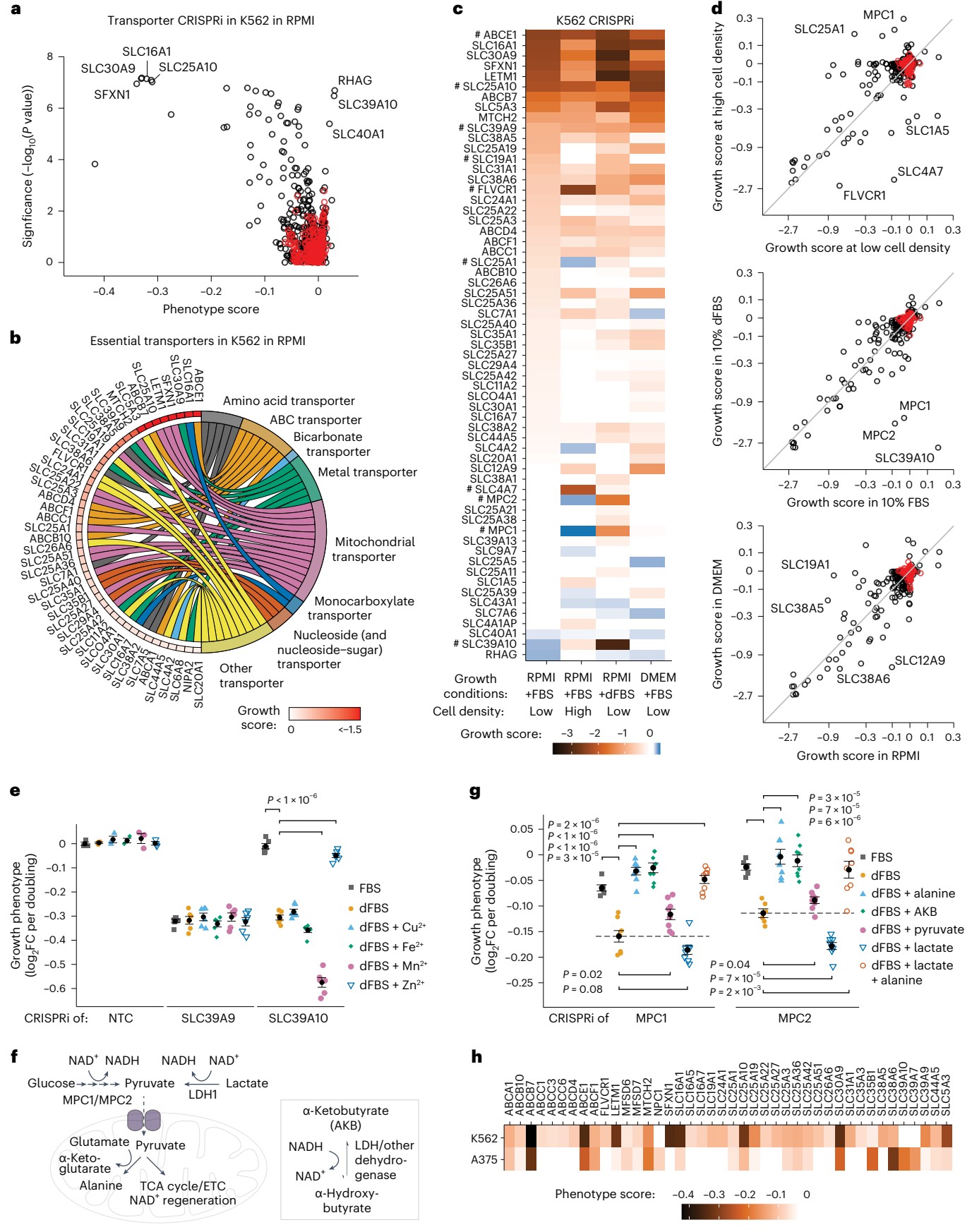

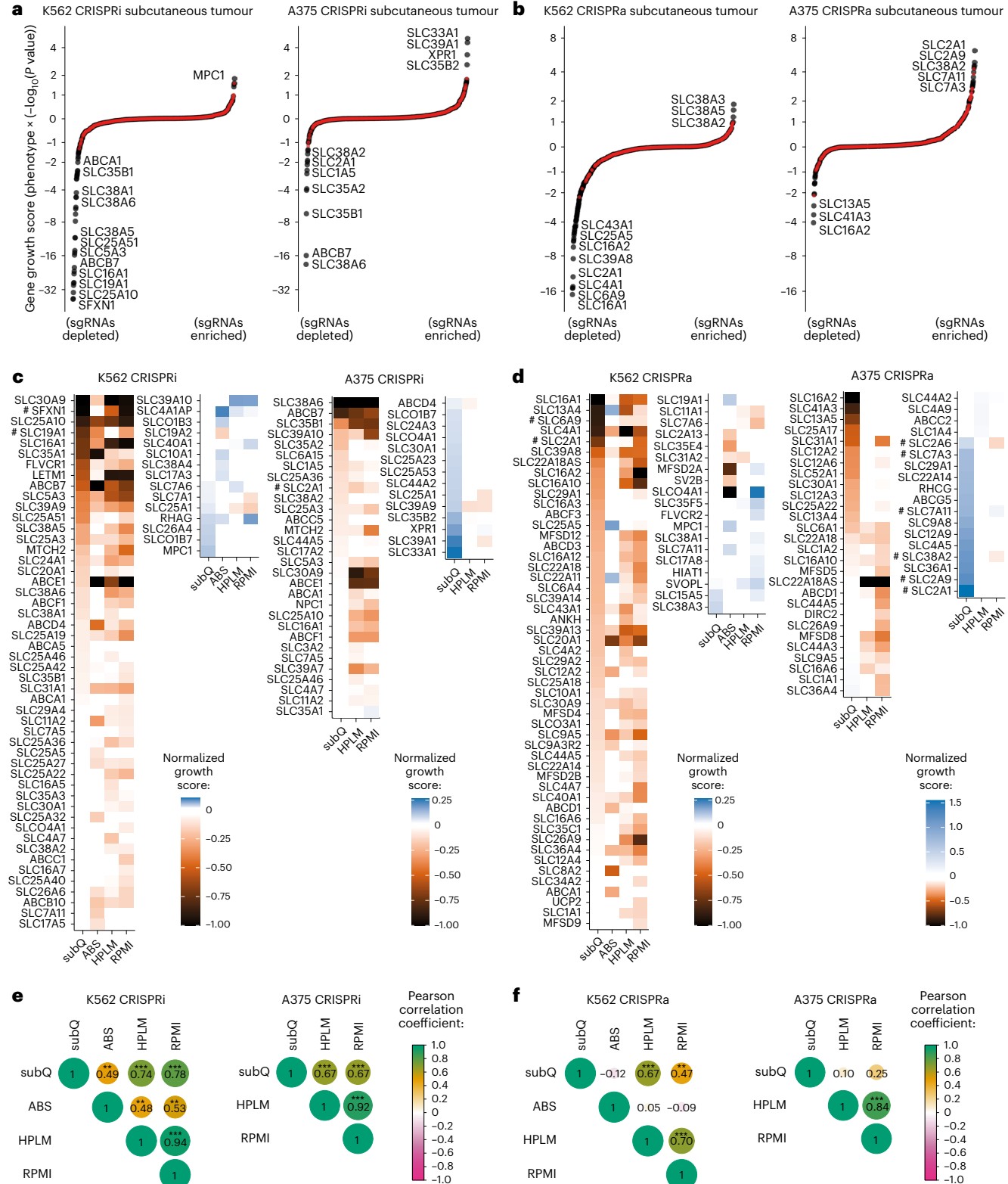

**Fig. 6 | Identification of essential transporters and environmental growth limitations in subcutaneous tumours. a,b**, Pools of K562 and A375 transporter CRISPRi/a libraries were injected subcutaneously in the flank of immunodeficient mice. Growth scores were determined from enrichment/depletion of sgRNAs in whole tumour homogenates (n = 4). The red circles represent negative control genes. **c,d**, A comparison of transporter CRISPRi/a phenotypes in subcutaneous (subQ) tumours to phenotypes in growth culture media identifies condition-specific effects. Pools of cells prepared in **a** and **b** were cultured in ABS (100% bovine serum), in HPLM and in RPMI. Data represent all transporters significantly enriched or depleted in at least one condition, and growth scores determined from two replicates were normalized to the most depleted transporter in each screen. '#' highlights transporters discussed in the main text. **e,f**, Pearson correlation coefficients determined from pairwise comparison of growth scores for enrichments/depletions in **c** and **d** and P values were determined using a two-sided test. **:$P < 1 \times 10^{-4}$; ***:$P < 1 \times 10^{-6}$. In **a** and **b**, P values were determined using a Mann–Whitney test. In **a**, **c** and **e**, data are from CRISPRi screens. In **b**,**d** and **f**, data are from CRISPRa screens. Source numerical data are available in Source data.

This is consistent with the absence of proliferation defects upon KD of the major amino acid transporters (SLC1A5, SLC7A1, SLC7A5 and SLC7A11) in K562 tumours. Differences between A375 and K562 in nutrient limitations may reflect differences in demand or concentrations within the tumour environments, warranting further investigation.

## Discussion

Efforts are ongoing to identify essential nutrients for tumour growth and leverage this for improved anti-cancer therapy[59]. In this study, we developed a CRISPRi/a screening strategy to systematically identify nutrient transporters in cells, focusing on characterizing amino acid transport in the K562 leukaemia cell line. While major amino acid transporter families are well described[10,12], our work enables the simultaneous interrogation of the contribution of each of the 64 annotated amino acid transporters to import and export across different conditions. To understand which transporters are essential or capable of import in nutrient-poor conditions, we performed screening at amino acid concentrations that limit proliferation. As these concentrations are generally lower than plasma or tissue interstitial fluid levels[22,23], our growth-based screens probably identify high-affinity transporters, and not all the low-affinity transporters that also contribute to physiological amino acid homeostasis. We explored low-affinity transport mechanisms in individual CRISPRi/a cell lines in transport assays, and by performing screens in nutrient-rich and physiological conditions. Overall, we identified one or more SLCs required for the transport of 13 different amino acids, and complementary transporters able to import amino acids when overexpressed.

A notable finding is that amino acid transport involves high bidirectional import and export flux at the membrane, aligning with proposed models of cellular amino acid homeostasis[12,13]. While net transport typically favours amino acid import for cell proliferation, some transporters like SLC43A1 and SLC43A2 act as net exporters, only becoming importers when their substrates are limiting. This challenges assumptions about SLC43A1 as an importer and its pursuit as an anti-cancer drug target[41]. Similar phenotypes were observed for SLC16A10 (TAT1), indicating potential net exporter functions (Extended Data Fig. 2b). These results emphasize the importance of measuring net flux across the membrane and considering the environment when assessing transport.

Our study shows that serotonin reduces lipid peroxidation, protecting cells from ferroptosis. Consistent with these findings, a recent study by Liu et al. also found that serotonin prevents ferroptosis by acting as a radical-trapping antioxidant[60]. Serotonin's antioxidant effect requires transport across the plasma membrane by SLC6A4, highlighting that lipid peroxides causing ferroptosis are probably located in intracellular membranes[33,61]. Varying across human tissues, serotonin levels and SLC6A4 expression peak in the gut[62], central nervous system or in diseases like carcinoid tumours[63]. Our results suggest that serotonin's antioxidant role may be prominent in such tissues with elevated serotonin and SLC6A4 levels[64]. Importantly, inhibitors of SLC6A4, such as SSRIs, negate the protective effect of serotonin and might increase ferroptotic sensitivity in those environments[65].

CRISPRi/a screening in subcutaneous xenografts identified nutrient dependencies and limitations in the TME. Although phenotypes in subcutaneous tumours were strongly correlated with in vitro conditions, certain transporters displayed large differences in essentiality between the in vivo and in vitro environments and further studies will be required to understand these effects. Although the TME is generally considered nutrient poor[2], specific constraints on tumour growth remain unclear and have not been systematically explored. Our study, along with others[35,66], demonstrates that identifying growth advantages from transporter OE highlights nutrients functionally limiting tumour growth. A375 melanoma cells faced glucose and amino acid limitation in xenograft tumours, while K562 cells showed no notable limitations, indicating potential variations in nutrient accessibility or environmental metabolite levels within tumours. These results align with observations that tissue interstitial fluid metabolite levels in lung cancer and pancreatic ductal adenocarcinoma xenografts are markedly different[23].

In conclusion, our CRISPRi/a screening platform systematically queries transporter activity and function, providing insights into nutrient uptake and secretion under diverse growth conditions in vivo and in vitro. Beyond nutrients, this method is suitable for studying drug transport and the impact of metabolism on drug sensitivity, as most small-molecule drugs are actively imported into cells via SLCs and can be exported via ABC transporters[67–69]. We anticipate broader applications of our approach in exploring various aspects of transporter biology.

## Online content

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

## Methods

The research in this manuscript complies with relevant ethical regulations. The research was approved by Harvard University's Committee on Microbiological Safety, and animal experiments were approved by the Massachusetts Institute of Technology Institutional Animal Care and Use Committee.

### Cell culture and chemicals

Cell lines were from ATCC: K562 (CCL–243), A375 (CRL–1619), C2BBe1 clone of Caco-2 (CRL–2102) and HEK293T (CRL–3216). Cells used in this study were from low-passage cultures from primary stocks, and cell lines were not further authenticated. K562 cells were grown in RPMI (Corning 10–040) supplemented with 10% (v/v) heat-inactivated FBS (Gibco 10438026). HEK293T and A375 cells were grown in Dulbecco's modified Eagle medium (DMEM, Corning 10–013) supplemented with 10% (v/v) FBS. Caco-2 cells were grown in Eagle's minimal essential medium (EMEM, ATCC 30–2003) supplemented with 10% (v/v) FBS. All cell lines were grown at 37 °C and 5% $CO_2$, and penicillin and streptomycin were added to all growth media to final concentrations of 100 U ml$^{-1}$ and 100 µg ml$^{-1}$, respectively (Corning 30–002–CI). Cells were tested for mycoplasma contamination using the MycoAlert mycoplasma detection kit (Lonza LT07-318).

(1$S$,3$R$)-RSL3 (Cayman Chemical 19288), erastin (MedChemExpress HY–15763) and ferrostatin-1 (Selleckchem S7243) were dissolved in dimethyl sulfoxide (DMSO), quality controlled and stored by the ICCB-Longwood screening facility. Fluoxetine HCl (Millipore Sigma F132), serotonin HCl (Tocris 354750) and 5-OH Trp (Millipore Sigma H9772) were prepared in DMSO. For metal ion complementation assays, $ZnSO_4 \cdot 7H_2O$ (Fluka 96500), $CuSO_4$ (VWR VW 3312–2), $Fe(NO_3)_3 \cdot 9H_2O$ (Sigma F–1143) and $MnCl_2 \cdot 4H_2O$ (Sigma M8530) were dissolved in $H_2O$ to 100 mM. Lactic acid (Fluka 69775) was adjusted to pH 7 with NaOH and diluted with $H_2O$ to 0.5 M, and sodium pyruvate (Sigma P2256) and AKB (Sigma K401) were dissolved in $H_2O$ to 300 mM and 1 M, respectively.

### Transporter CRISPRi/a sgRNA library cloning

Transporter libraries include sgRNAs targeting 413 SLC genes as defined by the Human Genome Organization gene nomenclature, 28 atypical SLCs and 48 ABC transporters (45 as defined by the Human Genome Organization and 3 atypical). sgRNA sequences were from Horlbeck et al.[34] and included 10 sgRNAs per gene for all genes, except 37 of them that had 2 transcription start sites and were therefore represented by 20 sgRNAs, and 730 NTC sgRNAs (Supplementary Table 1). The design and cloning of sgRNAs was performed as previously reported[34] with minor modifications. A single pool of oligonucleotides (12,000) for both CRISPRi and CRISPRa libraries was synthesized by Twist Biosciences. PCR reactions were set up using primers specific to either CRISPRi or CRISPRa sequences using Phusion polymerase (NEB) according to the manufacturer's instructions and using three different conditions to minimize amplification bias (HF buffer, GC buffer and GC buffer + 3% DMSO). Each 50 µl reaction included 4.5 ng template and 0.4 µM primers, and was amplified using eight cycles of 30 s at 98 °C, 30 s at 60 °C (57 °C for the DMSO-containing reaction) and 10 s at 70 °C preceded by 1 min at 98 °C and followed by 1 min at 72 °C. Two replicate sets of reactions were run, and all CRISPRi and CRISPRa amplified libraries were pooled separately. The amplified library (88 bp) was purified by agarose gel electrophoresis and extracted using the QIAquick gel extraction kit (Qiagen). Purified PCR products were digested with BlpI and BstXI, and the digested fragment was purified on a 20% polyacrylamide Tris–borate–EDTA (TBE) gel (Thermo Fisher Scientific). Digested fragments were purified by isopropanol precipitation, and DNA concentrations were quantified by fluorescence using the Qubit dsDNA high-sensitivity assay kit (Thermo Fisher Scientific). Plasmid pU6-sgRNA EF1Alpha-puro-T2A-BFP (Addgene #60955) was digested using BlpI and BstXI, purified by agarose gel electrophoresis and extracted using the QIAquick gel extraction kit (Qiagen). Digested PCR fragments were ligated into the restricted vector at a 1:1 molar ratio using T4 DNA ligase and were purified using the MiniElute PCR purification kit (Qiagen). Purified ligated plasmids were electroporated into MegaX DH10B (Thermo Fisher Scientific) and plated on LB/Amp plates. Library coverage was 10,000× for both libraries and was determined by serial dilution and colony counting. After 15 h at 30 °C, the lawn of bacteria was scraped off the plates, and plasmid DNA was prepared using the Plasmid Plus Maxi kit (Qiagen).

### Lentivirus preparation

HEK293T cells were transfected with the lentiviral plasmid, psPAX2 (Addgene #12260) and pCMV-VSV-G (Addgene #8454) in a 2:2:1 molar ratio using Lipofectamine 3000 (Invitrogen) according to the manufacturer's instructions. The growth medium was replaced 6 h post-transfection and was then collected at 24–30 h and 48–54 h post-transfection. The two collected growth medium fractions were pooled, centrifuged at 1,000$g$ for 10 min and filtered through a 0.45 µm low-protein-binding membrane. Lentivirus-containing supernatants were stored at −80 °C.

### Media preparation

To prepare complete RPMI lacking all amino acids, 8.59 g of RPMI without amino acids, sodium phosphate (US Biological R8999–04 A), 2.00 g sodium bicarbonate (Sigma S6014) and 0.80 g sodium phosphate dibasic (Sigma S0876) were diluted in 945 ml de-ionized $H_2O$. After addition of 100 ml dFBS (Gibco, Thermo Fisher Scientific 26400044) and 10 ml of penicillin–streptomycin 100× solution (Corning 30–002–CI) and homogenization, the medium was sterilized via 0.22 µm membrane filtration (RPMI + dFBS without amino acids). Amino acids were added to this base medium as needed from stock solutions to RPMI levels or to low amino acid screen concentrations, as described in Supplementary Tables 2 and 3. To make complete RPMI used as the control arm in low amino acid screens and experiments, all 19 amino acids were added to complete RPMI without amino acids to RPMI levels (RPMI rich).

Complete RPMI with amino acids present at physiological levels (PAA-RPMI) was prepared from RPMI + dFBS without amino acids as above, except that 965 ml de-ionized $H_2O$ was used. To that, amino acids were added to their concentration in human plasma[22] using the same stock solutions as for RPMI above. In addition to the amino acids present in RPMI, alanine, cysteine, carnitine, citrulline, creatine, creatinine, $N$-acetylglycine and ornithine were also added to their level in human plasma. Quantities added are described in Supplementary Table 2. For low amino acid conditions, the corresponding amino acid was added such that the final concentration in PAA-RPMI matched the concentration used in the CRISPRi/a screens.

To prepare RPMI medium for amino acid import assays (RPMI with 16 amino acids present as heavy isotopes), we first added unlabelled cystine, OH-Pro, Met and Trp to RPMI + dFBS without amino acids. This base medium consisting of complete RPMI lacking 16 amino acids was then used to prepare heavy-labelled RPMI and RPMI with low Leu or low Val. Stock solutions of heavy-labelled amino acids (Cambridge Isotope Laboratories) were added to RPMI lacking 16 amino acids to their concentration in RPMI or to the screen concentration (for Leu and Val), as outlined in Supplementary Table 2.

To prepare PAA-RPMI for amino acid import assays, we prepared a complete PAA-RPMI base as above but lacking 16 amino acids. We then added heavy-labelled amino acids from the same stocks as for RPMI to the concentration in PAA-RPMI, as outlined in Supplementary Table 2.

RPMI + FBS and DMEM + FBS were prepared as outlined in 'Cell culture and chemicals' section. HPLM (Thermo A4899101) was adjusted with 10% dFBS. ABS (Sigma-Aldrich B9433) thawed at 4 °C was adjusted to 50 µM with cystine·2HCl (Sigma C6727). After 1 h at 37 °C to ensure dissolution, ABS was filtered through a 0.2 µm low-protein-binding membrane.

## Amino acid titrations

Single amino acid dropout RPMI medium was prepared for each of the 19 amino acids present in RPMI by adding 18 amino acids to complete RPMI without amino acids. K562 cells grown in complete RPMI were washed 3× using cycles of centrifugation for 5 min at 300$g$ and resuspension of the cell pellet in phosphate-buffered saline (PBS, Corning 21–040 CV). The final cell pellet was first resuspended in PBS at a density of 10 million ml$^{-1}$ and then added to single amino acid dropout RPMI or complete RPMI to a final density of 0.1 million ml$^{-1}$. Thirty microlitres of these cell suspensions was pipetted into the wells of 384-well microplates (Thermo Fisher Scientific 164610). Amino acid stock solutions in H$_2$O (as described in Supplementary Table 2) were adjusted to 0.005% Triton X-100 using a 100× stock solution in H$_2$O and were added to wells of the microplates using a D300 digital dispenser (Hewlett-Packard). We confirmed that the highest concentration of Triton X-100 (0.33 parts per million) did not affect proliferation of K562 cells. Each amino acid was tested along a twofold dilution series from 1× to 1/1,024× its concentration in RPMI and including a no amino acid control. Wells on plate edges were filled but not used for any measurements. The drug dispensing arrangement for each amino acid was spatially randomized (and re-organized during data analysis) to minimize bias. Each data point was present in quadruplicates, and three separate identical plates were prepared for viability measurement after 24, 48 and 72 h incubation. Assay plates were incubated at 37 °C/5% CO$_2$ inside containers humidified by sterile wet gauze. After 24, 48 or 72 h, plates were removed from incubation and cooled at room temperature for 10 min, before dispensing 30 μl of CellTiter-Glo (CTG) (Promega) (1:1 dilution in PBS) into each well. Following a 10 min incubation at room temperature, luminescence was measured in a plate reader (BioTek Synergy H1). A control plate with K562 cells dispensed in complete RPMI was prepared and luminescence was monitored at the onset of the experiment ($T = 0$) and after 24, 48 and 72 h incubation in the same conditions to determine K562 doubling rates in RPMI. Each data point was averaged across four replicates, divided by luminescence at $T = 0$, was internally normalized within each plate and was displayed on a log$_2$ scale to represent the total number of population doublings.

## Preparation of CRISPRi/a parental cell lines

To generate a K562 cell line stably expressing dCas9-KRAB (K562 CRISPRi), K562 cells were transduced with lentiviral particles produced using vector pMH0001 (Addgene #85969; which expresses dCas9-BFP-KRAB from a spleen focus forming virus promoter with an upstream ubiquitous chromatin opening element) in the presence of 8 mg ml$^{-1}$ polybrene (Sigma). A pure polyclonal population of dCas9-KRAB-expressing cells was generated by two rounds of fluorescence-activated cell sorting (FACS) gated on the top half of BFP-positive cells (BD FACS Aria II) (Extended Data Fig. 8e). In a third round of FACS sorting, single cells from the top half of BFP-positive cells were sorted into microplates to establish monoclonal cell lines. The performance of K562 CRISPRi monoclonal lines in knocking down endogenous genes was evaluated by individually targeting three control genes (ST3GAL4, SEL1L and DPH1) and measuring gene expression changes by RT–qPCR, and the best-performing monoclonal cell line was selected for all work described in this study. The preparation of monoclonal K562 cells expressing CRISPRa machinery (K562 CRISPRa) and of A375 expressing CRISPRi (polyclonal) and CRISPRa (monoclonal) has been described elsewhere[25,55].

## Preparation of CRISPRi/a cell lines using sgRNAs targeting individual genes and expression analysis by RT–qPCR

Pairs of complementary synthetic oligonucleotides (Integrated DNA Technologies) forming sgRNA protospacers flanked by BstXI and BlpI restriction sites were annealed and ligated into BstXI/BlpI double-digested plasmid pU6-sgRNA EF1Alpha-puro-T2A-BFP (Addgene #60955). Oligonucleotides used to build sgRNAs targeting

individual genes are listed in Supplementary Table 4. The sequence of all sgRNA expression vectors was confirmed by Sanger sequencing, and lentiviral particles were produced using these vectors as described above ('Lentivirus preparation' section). Parental CRISPRi/a cells were infected with individual sgRNA expression vectors by addition of lentivirus supernatant to the culture medium in the presence of 8 μg ml$^{-1}$ polybrene. Transduced cells were selected using puromycin (2 μg ml$^{-1}$ for K562 cells) starting 48 h post-transduction and over the course of 7 days with daily addition of the antibiotic. After 24 h growth in puromycin-free medium, 0.1 million cells were collected and total RNA was extracted using the RNeasy Plus Mini kit (Qiagen). Complementary DNA was synthesized from 0.1–0.5 μg total RNA using Superscript IV reverse transcriptase (Invitrogen) and oligo(dT)$_{20}$ primers (Invitrogen) following the manufacturer's instructions. Reactions were diluted two- to fivefold with H$_2$O, and qPCR was performed using PowerUp SYBR Green PCR Master mix (Thermo Fisher Scientific), 2 μl diluted cDNA preparation and 0.4 μM of primers using a QuantStudio 6 Real-Time PCR system (Thermo Fisher Scientific). All qPCR primers are listed in Supplementary Table 4. The expression level of target genes was determined relative to the housekeeping gene GAPDH (log$_2$ fold change (FC) versus GAPDH) by subtracting Ct values (ΔCt). Cells were grown in RPMI or in low amino acid medium for 72 h at low confluence with daily media changes before expression profiling.

For Caco-2 cells, we used a one-vector CRISPRi system (Addgene #71236) to create polyclonal KD cell lines. Pairs of complementary synthetic oligonucleotides (Integrated DNA Technologies) forming sgRNA protospacers flanked by BsmBI restriction sites were annealed and ligated into BsmBI digested plasmid pLV hU6-sgRNA hUbC-dCas9-KRAB-T2a-Puro. Caco-2 cells were infected using lentiviral supernatant produced from these plasmids. Transduced cells were selected using EMEM containing puromycin at 2–5 μg ml$^{-1}$ over 12 days. Changes in expression level due to CRISPRi were quantified as above. Cells were grown in RPMI or in low-cystine medium for 72 h at low confluence with daily media changes before expression profiling.

## Transporter CRISPRi/a screens in low amino acid medium

Lentiviral supernatant was prepared for both the transporter CRISPRi and CRISPRa sgRNA libraries as described above ('Lentivirus preparation' section) and was stored at −80 °C. The multiplicity of infection (MOI) of both preparations was determined by titration onto target cell lines and quantification of the percentage of BFP$^+$ cells 2–3 days post-transduction by flow cytometry (BD Biosciences LSR II).

For transporter CRISPRi/a screens in low amino acid, K562 CRISPRi (or CRISPRa) parental cells (85–100 million) were transduced with lentiviral supernatant at an MOI of 0.25–0.3 in 250 ml culture medium + 8 μg ml$^{-1}$ polybrene in a 225 cm$^2$ cell culture flask (Costar). Twenty-four hours post-transduction, cells were collected and resuspended in 200 ml fresh medium in 2 × 225 cm$^2$ flasks. Starting 48 h post-transduction, the culture medium was exchanged daily and cells were maintained at 0.5–1.0 million ml$^{-1}$ in puromycin (1.75 μg ml$^{-1}$) in 400–500 ml. After 6 days in puromycin, the proportion of BFP$^+$ cells determined by flow cytometry increased to 94–96% of the fraction of viable cells. After recovery for 1 day in puromycin-free medium, 2 × 10 million library cells were collected and stored at −80 °C ($T = 0$ samples). The remaining cells were grown for 24 h in RPMI/rich (complete RPMI with all amino acids; 'Media preparation' section). A total of 120 million cells were collected by centrifugation at 300$g$ for 5 min and washed twice with 25 ml PBS, and the final cell pellet was resuspended in 4.8 ml PBS. Complete RPMI medium with 18 amino acids present at RPMI level and 1 amino acid present at a concentration that limits growth of K562 ('Media preparation' section) was prepared, and 50 ml was added to 150 cm$^2$ tissue culture flasks (Falcon 355001) that were preheated and equilibrated to 37 °C/5% CO$_2$. Library cells were added to the flasks to a final density of 0.1 million ml$^{-1}$. For all amino acids except Trp and Cys, screens were conducted over the course of 16 days with three cycles of

1 day in RPMI/rich and 4 days in RPMI/low amino acid, followed by a final day in RPMI/rich. The growth medium was changed daily on all flasks using centrifugation of a fraction of the culture and resuspension of the pellet into pre-equilibrated growth medium. For medium change after growth in RPMI/rich, cell pellets were washed twice with PBS before resuspension in low-amino-acid medium. The amount of culture centrifuged was adjusted such that the final cell density after medium exchange was 0.1 million ml$^{-1}$. For Trp and Cys, screens were conducted over the course of 16 days and medium was changed every 36 h (1× 24 h RPMI/rich; 4× pulses of 36 h RPMI/low amino acid; 1× 36 h RPMI/rich; 4× pulses of 36 h RPMI/low amino acid; 1× 36 h RPMI/rich). The RPMI/rich control arm and the low His, low Arg and low Lys were run in technical replicates. At the end of the screen, 12–15 million cells were collected by centrifugation, washed once with PBS and stored at −80 °C. The number of population doublings for each screen is indicated in Supplementary Table 3. Genomic DNA (gDNA) was extracted from cell pellets using the QIAamp DNA Blood Mini Kit according to the manufacturer's instruction, except that the elution was performed using 10 mM Tris HCl pH 8.5. Typical yields from 15 million cells ranged from 90 to 240 µg gDNA. sgRNA barcodes were amplified by PCR using the gDNA from at least 6 million cells as template and Phusion (NEB M0530) as polymerase. An equimolar mix of primers with stagger regions of different length (CC_LSP_025 to CC_LSP_032_c) was used as forward primer, and barcoded index primers (CC_Cri_a_rev1 to CC_Cri_a_rev10) were used as reverse primers. Reactions were composed of 1× HF buffer, 0.2 mM dNTPs, 0.4 µM forward primer mix, 0.4 µM indexed reverse primer, 0.5 µl Phusion, 1.5 mM MgCl$_2$ and 5 µg gDNA in a volume of 50 µl. After 30 s at 98 °C, the reactions were subjected to 23 cycles of 98 °C for 30 s, 62 °C for 30 s and 72 °C for 30 s, and were followed by 72 °C for 5 min. All reactions from each screen were pooled, and the amplified PCR product (~240–250 bp) was purified by agarose gel electrophoresis using the QIAquick gel extraction kit (Qiagen). Purified PCR products were quantified by fluorescence using the Qubit dsDNA high-sensitivity assay kit (Thermo Fisher Scientific). Individual indexed libraries were mixed in equimolar ratio and were further purified using a QIAquick PCR purification kit (Qiagen). After quantification by qPCR using the NEBnext library quant kit for Illumina (NEB), pooled libraries were sequenced on an Illumina HiSeq 2500 platform using a 50 bp single read on a high-output standard v4 flow cell with a 10–20% PhiX spike-in. About 15 million reads were obtained for each indexed screen.

For some of the screens, an alternative strategy was used to accommodate sequencing on an Illumina NextSeq 500 platform. Changes to the protocol above include amplification of extracted gDNA by PCR using a barcoded forward primer (CC_fwd1 to CC_fwd20) and a barcoded reverse primer (CC_Cri_a_rev1 to CC_Cri_a_rev20) using Q5 polymerase (NEB M0491L). Pooled libraries were sequenced on an Illumina NextSeq 500 platform using a 75 bp single read on a high-output flow cell with a 2–5% PhiX spike-in. A total of 15–30 million reads were obtained for each indexed screen.

Sequencing data were analysed as previously reported[25] with the following modifications. Trimmed sequences were aligned to the library of protospacers present in the transporter CRISPRi/a sgRNA libraries (Supplementary Table 1), and 89–92% of the number of raw reads typically aligned to the library of protospacers. To estimate technical noise in the screen, simulated negative control genes (the same number as that of real genes) were generated by randomly grouping 10 sgRNAs from the pool of 730 NTC sgRNAs present in the libraries. For each gene (and simulated control gene), which is targeted by ten sgRNAs, two metrics were calculated: (1) the mean of the strongest 7 rho phenotypes by absolute value ('phenotype score') and (2) the $P$ value of all 10 rho phenotypes compared with the 730 NTC sgRNAs (Mann–Whitney test). To display data compactly, we calculated a single score ('screen score') by multiplication of the phenotype score with −log$_{10}$($P$ value). sgRNAs were required to have a minimum of 100 counts in at least one of the two conditions tested to be included in the analysis.

To deal with the noise associated with potential low count numbers, a pseudocount of 10 was added to all counts. Gene-level phenotype scores and $P$ values are available in Supplementary Tables 5 and 6.

For screens in low amino acid conditions, scores were calculated using two technical replicates for Arg, Lys and Val. For all other conditions, scores were calculated using one replicate for the low amino acid condition and two technical replicates for both T0 and untreated samples. The highest-scoring negative control gene in any condition was used as the significance cut-off for all conditions (for CRISPRa, a significant hit had a screen score >0.425 or <−0.45; for CRISPRi, >0.49 or <−0.40). We excluded significant hits that had single scores <−0.12 in the untreated control arm of the screen unless they were either hypersensitive or very strong resistant (>1.5) hits in any of the low amino acid conditions. This exclusion was performed because a slowdown in proliferation rate leads to pan-resistance across screen conditions, as previously reported[25].

For in vivo screens, scores were determined without normalization to the number of population doublings. The number of sgRNAs required in at least one of two conditions tested was lowered to 25. For comparison with in vitro screens, screen scores were normalized to the most hypersensitive hit (set to −1) in a given screen. Significance cut-offs were determined by the highest-scoring pseudo negative gene (K562 CRISPRi, <−0.075 and >0.025; K562 CRISPRa, <−0.15 (−0.20 for subcutaneous) and >0.05 (0.10 for subcutaneous and ABS); A375 CRISPRi, <−0.05 and >0.03 (0.06 for subcutaneous); A375 CRISPRa, <−0.17 (−0.25 for subcutaneous) and >0.1 (0.7 for subcutaneous).

Phenotype similarity across environments was determined using all genes with a significant deleterious or beneficial phenotype in any of the environments tested. Pairwise Pearson correlations (and $P$ values) were calculated using the function rcorr in the R package Hmisc (v. 4.7.1), and data were displayed using the R package corrplot (v. 0.84).

For CRISPRi screens in K562 in RPMI, two biological replicates each consisting of two technical replicates were collected. To compute a volcano plot of the data, the read counts of the two technical replicates were averaged after normalization to the total number of reads and the resulting two replicates were processed in the analysis pipeline, as explained above. Essential transporters were determined by averaging screen scores determined for each biological replicate. A stringent significance threshold was determined by the highest-scoring pseudo-gene in any of the two replicates before averaging (score <−0.13). Essential transporters were arranged in a chord plot using GOPlot (v. 1.0.2).

For K562 CRISPRi screens in different culture conditions, significance cut-offs for each screen were determined by the highest-scoring pseudogene. Genes that were significant in at least one condition were displayed in tile plots. However, genes that did not have a growth score <−0.20 or >0.10 in any condition were excluded from the plots. For the comparison between K562 and A375 transporter essentiality, phenotype scores were used instead of growth scores to account for potential differences in sgRNA efficacy. Genes having a phenotype score <−0.065 were significant and were displayed in the tile plot.

## Transporter CRISPRi/a screens in rich medium

For K562 CRISPRi/a transporter screens in non-growth-limited conditions, pooled libraries were prepared as above. Cells were collected by centrifugation at 300$g$ for 5 min, washed in PBS and resuspended in PBS at 100 million ml$^{-1}$. Screens were initiated by addition of 7.5 million cells into 50 ml medium in 150 cm$^2$ flasks and were conducted by passaging the libraries in growth medium for 14 days with medium exchange at least each 2 days and keeping the density between 0.1 million ml$^{-1}$ and 0.4 million ml$^{-1}$. The pools of cells were collected at $T = 14$ days and $T = 0$ days, and enrichment was analysed as outlined above. For RPMI + 10% FBS at high cell density, cells were kept at a density of 0.3–1.2 million ml$^{-1}$ with medium exchange every 2 days.

For A375 screens, CRISPRi/a parental cells grown in RPMI + 10% FBS in 4× 15 cm cell culture dishes (Falcon 353025) to 80–90% confluence

were transduced with transporter library lentiviral supernatant at an MOI of 0.25–0.3 in presence of 8 µg ml$^{-1}$ polybrene. Starting 48 h post-transduction, cells were passaged daily to maintain <95% confluence in RPMI + 10% FBS + puromycin (1.0 µg ml$^{-1}$). After 4 days in puromycin, the proportion of BFP$^+$ cells determined by flow cytometry increased to 90–95% of the fraction of viable cells. After recovery for 1 day in puromycin-free medium, library cells were trypsinized and then quenched. Cells were collected by centrifugation at 300$g$ for 5 min, washed in PBS and resuspended in PBS at 100 million ml$^{-1}$. Screens were initiated by addition of 15 million cells into 50 ml medium in 15 cm dishes and were conducted by passaging the libraries in growth medium for 14 days with medium exchange at least every 2 days and keeping the confluence between 20% and 80%. Cells were collected at $T = 14$ days and $T = 0$ days, and enrichment was analysed as outlined above.

## Transporter screens in subcutaneous tumours in immunodeficient mice

All animal experiments conducted in this study were approved by the MIT Institutional Animal Care and Use Committee. A maximum tumour burden of 2 cm was permitted per Institutional Animal Care and Use Committee protocol, and these limits were not exceeded. Male mice between 3 and 4 months old were used in this study. All animals were housed at ambient temperature and humidity (18–23 °C, 40–60% humidity) with a 12 h light and 12 h dark cycle and co-housed with littermates with ad libitum access to water. For animal injections, pooled cells were washed twice with PBS and filtered through a 40 µm cell strainer. After centrifugation at 300$g$ for 5 min, pelleted cells were resuspended at 100 million ml$^{-1}$ in PBS and the suspensions were kept on ice until injection (<1 h). NOD.Cg-$Prkdc^{scid}$ $Il2rg^{tm1Wjl}$/SzJ mice (NSG mice; The Jackson Laboratory, strain #005557) were injected subcutaneously into both flanks with 0.1 ml of pooled cell suspension (10 million cells per tumour). Each screen was conducted with three mice ($N = 6$ tumours) over 14 days. At the endpoint, animals were killed and whole tumours (typically 100–500 mg) were excised and kept at −80 °C. gDNA was extracted from homogenized whole tumours (at least 160 µg for each tumour) using the DNeasy Tissue kit (Qiagen). sgRNA barcodes were amplified by PCR using 160 µg of gDNA, as described above for transporter screens in low amino acid screens. Libraries were quantified, pooled and analysed as described earlier. For each screen, the two samples that had the greatest number of NTC counts not within 1 log$_2$ of the median were excluded from the analysis to reduce technical noise. Remaining replicates were paired and read counts were averaged after correcting to match total read counts. These averaged samples were then processed with the $T = 0$ days samples to determine gene-level scores as outlined above.

## Determination of engraftment frequency in subcutaneous tumours in NSG mice

Lentivirus was prepared from plasmid pLJM1–eGFP (Addgene #19319). K562 CRISPRi + transporter library cells were infected with pLJM1–eGFP lentivirus, and the proportion of GFP$^+$ cells was determined by flow cytometry 72 h post-transduction. GFP$^+$ cells were spiked into K562 CRISPRi + transporter library to a final ratio of 1 to 1 × 10$^3$, 1 to 1 × 10$^4$ or 1 to 1 × 10$^5$. For each dilution and unspiked control, cells were prepared as above, diluted with PBS, and 100 µl (0.1, 1 or 10 million cells) was injected into both flanks of NSG mice. In addition, a portion of each preparation was collected for $T = 0$ days samples and for growth in RPMI + FBS. Tumours were allowed to form over the course of 19 (10 million cells injected), 26 (1 million) and 34 days (0.1 million). At the endpoint, animals were killed and whole tumours were excised and kept on ice. Tumours were dissociated, and the presence of GFP$^+$ cells was determined by flow cytometry analysis. To determine the skewness of the NTC sgRNA distribution, tumours were analysed in the same way as the screens.

## Immunoblotting

RPMI medium containing low amino acid was made in the same way as for CRISPRi/a screens. In addition, RPMI containing no Arg, His or Lys was prepared analogously, and we used RPMI/rich and RPMI without amino acids as controls. For the time course, K562 cells growing in RPMI/rich were collected, washed twice with treatment medium, and resuspended in treatment medium at 0.35–0.45 million ml$^{-1}$. After 24 h, cells were pelleted and resuspended in fresh medium at 0.35–0.45 million ml$^{-1}$. For each timepoint, 0.6 million K562 cells were pelleted by centrifugation at 300$g$ for 3 min and flash frozen in liquid nitrogen. Pellets were lysed in 1× RIPA buffer (50 mM Tris HCl, 150 mM NaCl, 0.5% sodium deoxycholate, 0.1% sodium dodecyl sulfate and 1% NP40) and lysates clarified by centrifugation at 21,000$g$ for 10 min at 4 °C. Protein concentration in lysates was quantified by bicinchoninic acid assay (Pierce). Then, 20–25 µg protein was run on a 4–20% Bis-Tris Bolt gel (NuPAGE) in 1× MES buffer (NuPAGE) at 125 V for 90 min, and transferred to 0.45 µm nitrocellulose at 18 V for 60 min using the Trans-Blot SD semi-dry transfer system (Bio-Rad) in Tris–glycine transfer buffer (10 mM Tris base, 0.1 M glycine and 20% MeOH). The membrane was blocked with 5% skimmed milk or bovine serum albumin (for phospho-antibodies) in TBST for 1 h at room temperature, and incubated overnight at 4 °C with primary antibody in the same solution used for blocking. After 3× 5 min washes with TBST, incubation for 1–3 h at room temperature with 1:5,000 secondary antibody (Cell Signaling Technology goat HRP-linked anti-rabbit IgG #7074), and three more 5 min washes with TBST, blots were developed in 2 ml Western Lighting Plus chemiluminescent substrate (PerkinElmer) and imaged on the ImageQuant LAS4000 (Cytiva). After imaging, the blot was incubated for 15 min in Restore Western Blot Stripping Buffer (Thermo Fisher Scientific) followed by 1 h in 30% H$_2$O$_2$ at 37 °C. After extensive washes with TBST, the blot was blocked and reprobed as above. For each experiment, a single blot was serially assayed with the following rabbit antibodies (Cell Signaling Technology): vinculin (E1E9V) #13901 dilution 1:2,000; phospho-p70 S6 kinase (Thr389) #9234 dilution 1:1,000; phospho-eIF2α (Ser51) (D9G8) #3398 dilution 1:1,000; phospho-4E-BP1 (Ser65) #9451 dilution 1:1,000; p70 S6 kinase #9202 dilution 1:1,000; eIF2α #9722 dilution 1:2,000; 4E-BP1 (53H11) #9644 dilution 1:2,000.

For plasma membrane fractions, we prepared a fresh 10 mM solution of EZ-link-sulfo-NHS-SS-biotin reagent (Thermo Fisher Scientific A39258) in H$_2$O. Cells were grown in low amino acid media for 72 h at low confluence with daily media changes or in regular RPMI. Cells (5–20 million) were washed twice with ice-cold PBS and resuspended to 40 million ml$^{-1}$ in PBS. EZ-link-sulfo-NHS-SS-biotin was added to 800 µM, and cells were incubated for 30 min at room temperature with gentle rocking. Cells were pelleted and resuspended in 500 µl 150 mM glycine and incubated for 2 min. After washing with 1 ml PBS, cell pellets were resuspended in lysis buffer (50 mM Tris HCl pH 7.5, 100 mM NaCl, 5 mM EDTA pH 8, 1% Triton X-100, 1× protease and phosphatase inhibitor cocktail (Thermo Fisher Scientific 1861280)) at 22.2 µl per 1 million cells. Cells were lysed at 4 °C for 10 min, and lysates were clarified by centrifugation at 21,000$g$ for 10 min at 4 °C. Supernatants ('total extracts'; about 3.6 µg µl$^{-1}$) were stored at −80 °C. For western blots, 1/6 volume of 6× Laemmli sample buffer was added and samples were incubated at 95 °C for 5 min. For pull-downs, 100–300 µl total protein extract was added to a tube containing 25 µl of streptavidin agarose resin (Thermo Fisher Scientific 20357) washed once with lysis buffer. After rotating samples at 4 °C for 60 min, the resin was washed with 3× 500 µl lysis buffer and centrifugation at 500$g$ for 3 min. Biotinylated proteins were eluted by incubating washed beads with 1× Laemmli sample buffer (1 µl per 7.5 µg input total protein) at 95 °C for 5 min.

For plasma membrane fraction western blotting, 10–30 µg total protein extract and the equivalent of 30–90 µg input protein of pull-down samples were run on 4–15% Mini Protean TGX precast gels (Bio-Rad) and transferred to nitrocellulose using the iBlot3 system and reagents (Invitrogen) at 25 V for 3 min. Membranes were blocked

in Intercept (TBS) Blocking Buffer (LI-COR) for 1–2 h at room temperature, and then incubated overnight at 4 °C with primary antibody in blocking buffer + 0.2 % Tween-20 (SLC7A5 (Proteintech, 28670-1-AP) 1:5,000; SLC3A2/4F2hc (Cell Signaling, D6O3P, #13180) 1:1,000; SLC7A6 (Novus Biologicals, NBP2-75086) 1:500; SLC7A7 (Novus Biologicals, NBP1-82826) 1:500; SLC2A1 (Cell Signaling, D3J3A, #12939) 1:1,000). After 3× 5 min washes with TBST, incubation for 1–3 h at room temperature with 1:20,000 anti-rabbit secondary antibody (IRDye800CW LI-COR 926-32211) or 1:10,000 anti-goat secondary antibody for SLC7A6 only (donkey IgG H&L, Alexa Fluor 750 conjugate, Abcam ab175744) in blocking buffer + 0.2% Tween-20 + 0.01% sodium dodecyl sulfate. After 3× 5 min washes with TBST, blots were imaged on the iBright FL1500 Imaging System (Invitrogen). For vinculin staining post-imaging, blots were re-processed as above starting with an overnight incubation in 1:2,000 vinculin (Cell Signaling, E1E9V, #13901) in blocking buffer + 0.2 % Tween-20.

### Growth competition assays

Growth phenotypes were measured in competition assays where a test cell line was mixed with its corresponding NTC control at a 1:1 ratio. Cell lines were grown in complete RPMI, and cell densities were determined using a TC20 automated cell counter (Bio-Rad). Cell mixes were prepared by combining 1 million test cells with 1 million NTC cells, washing twice with 10 ml PBS, and resuspending the final pellet in 1 ml PBS. Fifty microlitres of resuspended cells (0.1 million) was added to wells of a 12-well plate containing 1 ml of growth medium, and an aliquot was stored at −80 °C to determine the initial ratio for each mix. Cultures were passaged as necessary (that is, daily for low amino acid screen validation) by centrifugation of a portion of the culture and resuspension in fresh medium. At the end of the experiment, the total number of population doublings was determined for all conditions and 0.1–0.2 million cells were collected by centrifugation. gDNA was extracted from cell pellets using the QIAamp DNA Blood Mini Kit according to the manufacturer's instruction, except that the elution was performed using 10 mM Tris HCl pH 8.5 (typical yields were 2–10 µg). Each extracted gDNA sample was assessed in two qPCR reactions using either a forward primer complementary to the test cell sgRNA sequence or to the NTC sgRNA sequence, and a universal reverse primer. Primers were tested beforehand to ensure linearity and specificity of detection. The composition of the mix was estimated by the difference in Ct value between the two reactions ($\Delta$Ct). Growth phenotypes were determined by comparing the $\Delta$Ct post-growth in the test medium with the $\Delta$Ct of the sample taken at $T = 0$ ($\Delta\Delta$Ct), and then normalizing by the number of population doubling differences between the test medium and a vehicle control (for example, low amino acid versus RPMI/rich). To determine growth phenotypes in complete media, $\Delta\Delta$Ct values were normalized by the number of population doublings between $T = 0$ and the end of the assay.

### Amino acid transport assays in K562 cells

The polymer coverslip in a 35 mm µ-Dish (Ibidi 81156) was coated with Cell-Tak. For each dish, 12.25 µg Cell-Tak (Corning 354240) diluted to 360 µl with $H_2O$ was neutralized with 40 µl 1 M bicarbonate pH8 solution and rapidly spread over the surface of the coverslip. After 45 min at room temperature, the solution was removed and the coverslip was thoroughly washed with 2× 800 µl $H_2O$ and subsequently left to dry for a few minutes. To prepare cells for immobilization, K562 were collected (1 million cells per dish), washed once with RPMI without FBS and resuspended in RPMI without FBS (400 µl per dish). Resuspended cells were added to coated dishes and incubated at room temperature for 30–45 min. Medium and unattached cells (typically, 0.8 million cells are needed to form a homogeneous monolayer of K562 cells) were gently removed by aspiration. After one wash with 500 µl RPMI/rich, cells were incubated for at least 2 h at 37 °C/5% $CO_2$ in 500 µl RPMI/rich to reach steady state.

For import assays, one dish is required for each timepoint for each cell line. We typically assayed amino acid import over seven to eight timepoints (0, 5, 10, 20, 30, 40, 100 and 250 s) to capture the initial slope for all 16 amino acids. RPMI/rich was removed by aspiration and was replaced by 400 µl pre-warmed and pre-equilibrated RPMI + 16 heavy-labelled amino acids. After incubation at 37 °C for the duration of the timepoint (on a heat block for short timepoints and in the incubator for longer timepoints), cells were thoroughly washed by sequentially submerging dishes into 4× 400–600 ml ice-cold PBS. For the 0 s timepoint, the same procedure as above was performed but using RPMI/rich instead of heavy-labelled medium and incubating for 10 s. PBS in the washes was refreshed regularly to ensure limited contamination of intracellular amino acid pools. After the last wash, PBS was removed by aspiration and 600 µl of ice-cold extraction buffer (80% MeOH/20% $H_2O$ spiked with 4 µg ml⁻¹ norvaline (Sigma N7627)) was added. Cells were scraped off the plate on ice (United Biosystems MCS-200) and were transferred to 1.5 ml tubes. Samples were homogenized in a thermomixer at 2,000 rpm and 4 °C for 15 min. Tubes were spun at 21,100$g$ for 10 min at 4 °C, and supernatants were transferred to new tubes containing 10 µl of 0.1 N HCl and were stored at −80 °C until analysis by GC–MS.

For amino acid consumption assays, a single dish was required for all timepoints for each cell line. We typically assayed consumption over five timepoints (0, 1, 2, 3 and 4 h) in RPMI and over four timepoints (0, 5, 15 and 60 min) in low amino acid medium. RPMI/rich was removed by aspiration and was replaced by 300 µl pre-warmed and pre-equilibrated RPMI/rich. At each timepoint, 15 µl of medium was removed and kept on ice. All samples were spun at 1,000$g$ for 5 min at 4 °C, and 10 µl of the supernatant was transferred to a tube containing 600 µl of extraction buffer + 10 µl of 0.1 N HCl. Samples were stored at −80 °C until analysis by GC–MS.

For GC–MS analysis, samples were dried at room temperature using a flow of nitrogen. To each tube, we added 24 µl methoxamine reagent (ThermoFisher TS-45950) and incubated the resuspended samples at 37 °C for 1 h. We next added 30 µl N-methyl-N-(tert-butyldimethylsilyl) trifluoroacetamide + 1% tert-butyldimethylchlorosilane (Sigma 375934) and incubated homogenized samples at 80 °C for 2 h. After centrifugation at 21,100$g$ for 10 min, supernatants were transferred to glass vials. Derivatized samples were analysed on a DB-35MS column (Agilent Technologies) in an Agilent 7890B gas chromatograph linked to an Agilent 5977B mass spectrometer. One microlitre of sample was injected at 280 °C and mixed with helium carrier gas at a flow rate of 1.2 ml min⁻¹. After injection, the oven was held at 100 °C for 1 min and ramped to 250 °C at 3.5 °C min⁻¹. The oven was then ramped to 320 °C at 20 °C min⁻¹ and held for 3 min at 320 °C. Electron impact ionization in the mass spectrometer was performed at 70 eV, and the MS source and quadrupole was held at 230 °C and 150 °C, respectively. Scanning mode was used for detection with a scanned ion range of 100–650 $m/z$. Acetone washes were performed after every three to five samples and before and after each run. GC–MS raw data were quantified using El-MAVEN (v0.11.0), and, for each amino acid, we extracted the area of the peak for the fragment with the highest signal over noise for the parent ion and all detectable isotopologues (fragments are highlighted in Fig. 2b). Natural isotope abundance was subsequently corrected using IsoCorrectoR to determine total ion counts for each amino acid fragment[70]. For import assays, each sample was first normalized using the norvaline internal standard. We corrected each sample by a factor determined by calculating the ratio of the norvaline signal to the average of the norvaline signal over all timepoints. As the experiment was run under steady state conditions, we applied a second correction that took advantage of the fact that the total ion count for all amino acid fragments (labelled and unlabelled) remains constant over all timepoints. We calculated the total ion count (excluding norvaline) for each sample and corrected each sample by a factor determined by calculating the ratio of the total ion count for the sample to the average total ion count over all timepoints. To determine absolute amounts in

each sample, we used a standard mix (Cambridge Isotope Laboratories MSK–A2) containing 17 amino acids to which we added Gln and Asn. We made dilutions of this mix in 0.1 N HCl and added them to 600 µl K562 cell extract prepared as for the import assay. We analysed these samples as above and applied a linear regression to the data. We then used the linear regression data and the mean norvaline signals of the standard and the sample to convert ion counts into pmol amino acid. Data were normalized by the number of cells used in the assay (0.8 million in cell monolayer minus an estimated 20% cell loss during the washing steps). Intracellular amino acid levels were calculated by averaging the sum of unlabelled and heavy-labelled amino acid over all the timepoints. To calculate import rates, we determined the initial slope of the increase in intracellular levels of heavy-labelled amino acids. We applied a linear regression to heavy-labelled amino acid levels over time, and the number of data points included was variable depending on the amino acid and the conditions. We typically used data from 0 to 100 s for Asn, Asp, Gln, Glu and Pro; data from 0 to 40 s for Arg, Gly, His, Ile, Lys, Ser, Thr and Val; data from 0 to 20 s for Leu and Tyr; and data from 0 to 10 s for Phe.

For import assays in low-arginine and low-lysine conditions (and RPMI control), we used liquid chromatography–mass spectrometry (LC–MS) to quantify light- and heavy-labelled amino acids in methanol cell extracts prepared in the same way as for GC–MS. Samples were dried down using a flow of nitrogen, resuspended in 100 µl 80% MeOH/20% $H_2O$ and filtered through a 10 kDa PES filter. LC–MS analysis was performed using a QExactive orbitrap mass spectrometer using an Ion Max source and heated electro-spray ionization probe coupled to a Dionex Ultimate 3000 UPLC system (Thermofisher). For this, 2.5 µl sample was injected onto a SeQuant ZIC-pHILIC 2.1 mm × 150 mm (5 µm particle size) column (Millipore Sigma). The flow rate was set to 0.15 ml min$^{-1}$, and temperatures were set to 30 °C for the column compartment and 4 °C for the autosampler tray. The following conditions were used to achieve chromatographic separation: mobile phase A was 20 mM ammonium carbonate, 0.1% ammonium hydroxide, and mobile phase B was 100% acetonitrile. The chromatographic gradient was as follows: 0–20 min, linear gradient from 80% to 20% B; 20–20.5 min, linear gradient from 20% to 80% B; 20.5–28 min, hold at 80% B. The mass spectrometer was operated in full scan, polarity-switching mode, the spray voltage was set to 4.2 kV and the heated capillary was held at 320 °C. The MS data acquisition was performed in a range of 70–1,000 $m/z$, with the resolution set at 70,000, the AGC target at $1 \times 10^6$ and the maximum injection time at 20 ms. Quantification of light- and heavy-labelled amino acid levels from LC–MS raw data and import rates were determined as for GC–MS without natural abundance correction and calculation of absolute levels.

For consumption experiments, each sample was normalized using the norvaline standard as above. We next adjusted sample signals to account for the decrease in volume of medium at each timepoint. Finally, we made use of the fact that the experiment was run under steady-state conditions to correct samples on the basis of the total ion count (excluding norvaline) of each sample. We applied a linear regression to the total ion count over the time course of each experiment. We then corrected each sample by a factor calculated by taking the ratio of the total ion count of the sample to the computed level determined by the regression. To determine absolute amounts in each sample, we diluted 10 µl of the standard mix prepared above into 600 µl extraction buffer + 10 µl RPMI/rich and analysed these standards as detailed for the samples above. We applied a linear regression to the standard data and converted sample ion counts to pmol using the regression data as well as the mean norvaline signals of the standards and samples. Consumption rates were calculated from the slope of the changes in amino acid levels over the time course of the assay determined by linear regression.

Amino acid export was measured by exposing cells to RPMI containing heavy-isotope-labelled amino acids, rapidly switching to regular RPMI and measuring the release of labelled amino acids into the medium. While this approach was technically challenging due to the

rapidity of export and the effects of differences in intracellular amino acid levels on export rates, we were able to quantify export rates for a subset of amino acids. K562 cells attached in a monolayer on the surface of an Ibidi dish were prepared as for the import assay. Cells were incubated for at least 2 h with RPMI containing heavy-labelled amino acids at 37 °C/5% $CO_2$. Growth medium was removed by aspiration, and cells were thoroughly washed by sequentially submerging dishes into 4× 400–600 ml PBS at room temperature. After aspiration of the last PBS wash, 160 µl pre-warmed and pre-equilibrated RPMI/rich was added to the dish. Plates were kept at 37 °C, and the medium was homogenized by regularly swirling the dishes. Samples of the media (12.5 µl) were taken over the course of the assay (0, 5, 10, 20, 30, 40 and 100 s) and kept on ice. Samples were centrifuged at 300$g$ for 5 min at 4 °C, and 10 µl of the supernatant was transferred to a tube containing 600 µl of extraction buffer + 10 µl of 0.1 N HCl. Samples were stored at −80 °C until analysis by GC–MS that was performed and analysed in the same way as the consumption samples.

## Cell viability assays

For K562, cells growing in RPMI/10% FBS were collected by centrifugation, washed 2× with PBS and resuspended at 10 million ml$^{-1}$ in PBS. Medium was prepared for each growth condition by addition of small molecules to RPMI low cystine using DMSO stocks. K562 cells were added to each medium to a final concentration of 0.1 million ml$^{-1}$, and suspensions were dispensed into wells of a 48-well plate in triplicates. After 72 h incubation at 37 °C/5% $CO_2$, 50 µl of each culture was transferred to a 96-well plate (in duplicates), 50 µl CTG reagent was added and luminescence was measured in a plate reader (BioTek Synergy H1). Viability was quantified by normalization to luminescence at $T = 0$ h. For the cystine titration, K562 cells washed in PBS were resuspended in RPMI −Cys (0.1 million ml$^{-1}$) and dispensed in a 96-well plate. Cystine was added using a D300 digital drug dispenser (HP), and viability at 72 h was determined by addition of CTG reagent and measurement of luminescence.

For A375, cells were trypsinized and resuspended in RPMI −Cys, then washed 2× with RPMI −Cys and finally resuspended at 10 million ml$^{-1}$ in the same medium. Cells were added to growth medium to 0.1 million ml$^{-1}$ and dispensed into wells of a 96-well plate. After 2 h at 37 °C/5% $CO_2$, small molecules were added to wells using a D300 digital drug dispenser (HP) using stocks in DMSO. Viability was determined at $T = 48$ h by addition of CTG reagent and measurement of luminescence.

For Caco-2 assays, 96-well microplates were coated with poly-D-lysine to avoid cell loss during washing steps. Forty microlitres of a 1:1 (v/v) solution of poly-D-lysine (Sigma A–003–M) 0.1 mg ml$^{-1}$ in $H_2O$ and PBS was added to each well and incubated for 1 h at room temperature. Wells were washed 3× with PBS and left to air dry. Caco-2 cells were seeded into the 96-well plates in EMEM/10% FBS at a density of about 1,000 cells per well. After 48 h at 37 °C/5% $CO_2$, wells were washed twice with RPMI −Cys followed by addition of 50 µl RPMI −Cys. Small molecules were added to wells using a D300 digital drug dispenser (HP). Viability was quantified at $T = 48$ h using the CTG luminescence assay and compared with viability at $T = 0$ h.

## Lipid peroxidation quantification

K562 cell lines growing in RPMI/rich were collected, washed 2× with PBS and incubated for 24 h in RPMI −Cys at 1 million ml$^{-1}$. After media exchange to fresh RPMI −Cys and dilution to 0.5 million ml$^{-1}$, cells were dispensed in a 12-well plate (2 ml per well). Cystine was added from a 200× aqueous stock solution, and RSL3 and serotonin were added using 1,000× stocks in DMSO. After 6 h incubation at 37 °C/5% $CO_2$, 100 µl of a solution of BODIPY 581/591 C11 (Thermo Fisher D3861) 200 µM in RPMI −Cys (5% v/v DMSO) was added to each well. After 30 min incubation at 37 °C/5% $CO_2$, cells were washed once with PBS and analysed by flow cytometry (BD LSRFortessa). The oxidized form of BODIPY C11 was quantified using 488 nm emission and 530/30 nm

emission, and the reduced form was quantified using 561 nm emission and 610/20 nm emission. Peroxide levels were estimated by the ratio of these two fluorescence intensities.

### Statistics and reproducibility

All experimental assays, excluding CRISPR screens, were performed at least in triplicate. The exact number of replicates is indicated in the figure legends. In general, data represent the mean of individual replicates and error bars represent the standard error of the mean (s.e.m.). Data analysis and display was accomplished in R (v. 3.6.0) using ggplot (v. 3.3.3). Significance probabilities, except for CRISPR screens, were calculated using two-tailed unpaired Student's $t$-tests. For western blots, experiments were reproduced twice with similar results and one of the replicates was shown as representative example. No sample size calculation was performed for experiments in cell culture. For in vivo screens, we pre-determined in initial experiments that sufficient library representation was achieved in every single tumour and we included six replicate tumours for each screen condition. To improve data quality in these screens, we excluded from each set of screens the two datasets with the highest amount of technical noise. Noise was determined in an unbiased way by quantifying the number of NTC sgRNA counts that were within $1 \log_2$ of the median after growth in vivo. No other data were excluded from the study. Transporter CRISPR screens were run in duplicates, and all attempts at replication were successful (Extended Data Fig. 2c,d). Mice were randomly chosen for the four subcutaneous injection conditions. Other experiments were not randomized as there was no allocation into different experimental groups. The researcher performing subcutaneous injections was blinded to the identity of the cells injected. All pooled CRISPR screens are inherently randomized and blinded. Other experiments were not blinded as all data were collected using unbiased methods.

### Reporting summary

Further information on research design is available in the Nature Portfolio Reporting Summary linked to this article.

### Data availability

All CRISPR screening data are provided in Supplementary Tables 5 and 6. The composition of the sgRNA libraries is provided in Supplementary Table 1. Source data are provided with this paper. All other data supporting the findings of this study are available from the corresponding authors on reasonable request.

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

### Acknowledgements

We thank A. Muir, N. Matheson, Z. Li, S. Block and the Sorger and Vander Heiden labs for discussions, and J. Weissman, D. Trono, B. Weinberg, D. Sabatini and C. Gersbach for plasmids. This work was supported by NIH/NCI grant U54-CA225088 to P.K.S., grants R35-CA242379 and P30-CA014051 to M.G.V.H., a Jane Coffin Childs Memorial Fund for Medical Research fellowship to A.M.D., Ludwig Cancer Research, the Termeer and Lustgarten Foundations, and the MIT Center for Precision Cancer Medicine.

### Author contributions

C.C. designed the study and performed experiments. C.C. analysed data with help from A.M.D. C.C. and A.M.D. performed cellular transport assays and western blots. B.L.G. assisted with experiments. E.C.L., A.M.B. and C.C. performed mouse screens. M.G.V.H. and P.K.S. supervised the project and obtained funding. C.C. and P.K.S. wrote the paper. A.M.D. edited the paper.

### Competing interests

M.G.V.H. is a scientific advisor for Agios Pharmaceuticals, iTeos Therapeutics, Sage Therapeutics, Auron Therapeutics and Droia Ventures. P.K.S. is a co-founder and member of the BOD of Glencoe Software, a member of the BOD of Applied Biomath and a member of the SAB of RareCyte, NanoString and Montai Health, and a consultant for Merck. None of these relationships has influenced the content of this manuscript. The other authors declare no competing interests.

### Additional information

**Extended data** is available for this paper at https://doi.org/10.1038/s41556-024-01402-1.

**Correspondence and requests for materials** should be addressed to Christopher Chidley or Peter K. Sorger.

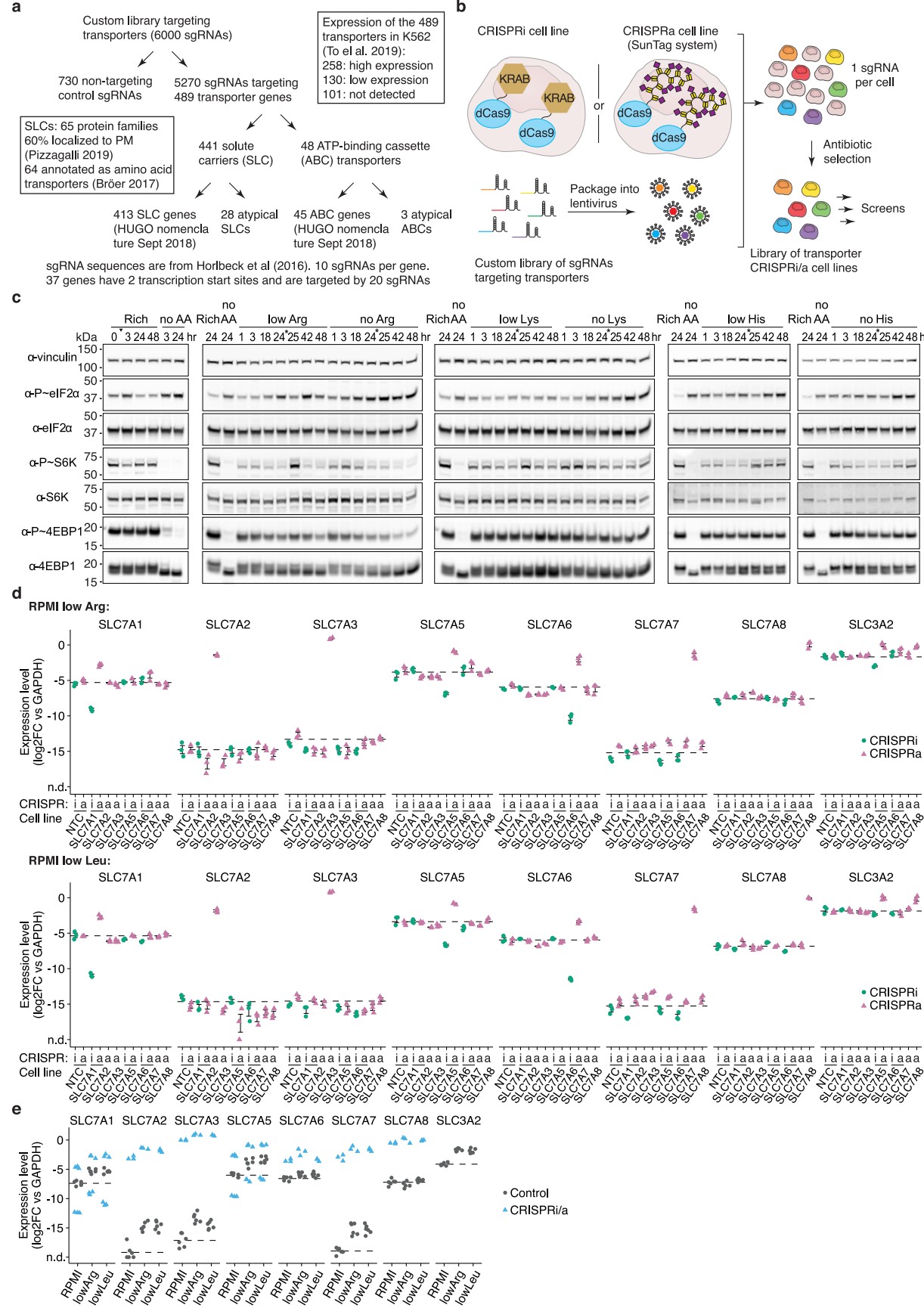

**Extended Data Fig. 1 | See next page for caption.**

**Extended Data Fig. 1 | Transporter CRISPRi/a screening optimization.**
(**a**) Composition of the CRISPRi/a transporter sgRNA libraries. (**b**) Preparation of
K562 CRISPRi or CRISPRa pooled transporter libraries. Custom sgRNA libraries
for CRISPRi and CRISPRa were constructed and packaged into lentivirus. A
K562 monoclonal cell line expressing either the CRISPRi or CRISPRa machinery
was infected with lentiviral particles, and untransduced cells were removed by
antibiotic selection. Because transporter expression is often tissue-specific[20]
and cell lines can lose transporter expression over time[71], CRISPRa bypasses
the need for transporter expression. CRISPRi was used over CRISPR/Cas9
knockout because genetic loss-of-function strategies are often complicated
by transcriptional adaptation[71,72]. (**c**) Assessment of the activity of the mTORC1
and GCN2 pathways in low amino acid screen-like conditions for Arg, Lys, and
His via Western blotting of downstream targets eIF2α (GCN2), and S6 kinase and
4E-BP1 (mTORC1). K562 cells grown in complete RPMI medium (Rich; the triangle
depicts a centrifugation step) were rapidly pelleted and transferred to either

Rich, low amino acid RPMI, or single amino acid dropout RPMI and samples were
taken at specific time points after media exchange. The media were exchanged
after 24 h (medium exchange denoted by asterisk). For all samples, cell lysates
were immunoblotted with antibodies against vinculin (loading control), and
phospho- and total antibodies against eIF2α, S6K, and 4E-BP1. (**d**) CRISPRi/a
of transporters leads to specific changes in gene expression. Transporter
expression levels in K562 CRISPRi/a cells with specific or non-targeting control
(NTC) sgRNAs and grown in low Arg or low Leu RPMI was quantified by RT-qPCR
relative to the housekeeping gene GAPDH. n = 3 technical replicates. Data are
mean ± s.e.m. n.d., not detected. (**e**) low Arg and low Leu conditions induce
a small transcriptional upregulation of most tested SLC7 transporters. Data
are a subset of expression levels determined in Fig. 1b and (d). For each gene,
expression levels of CRISPRi/a cell lines targeting that gene and NTC cell lines
were plotted. Source numerical data and unprocessed blots are available in
source data.

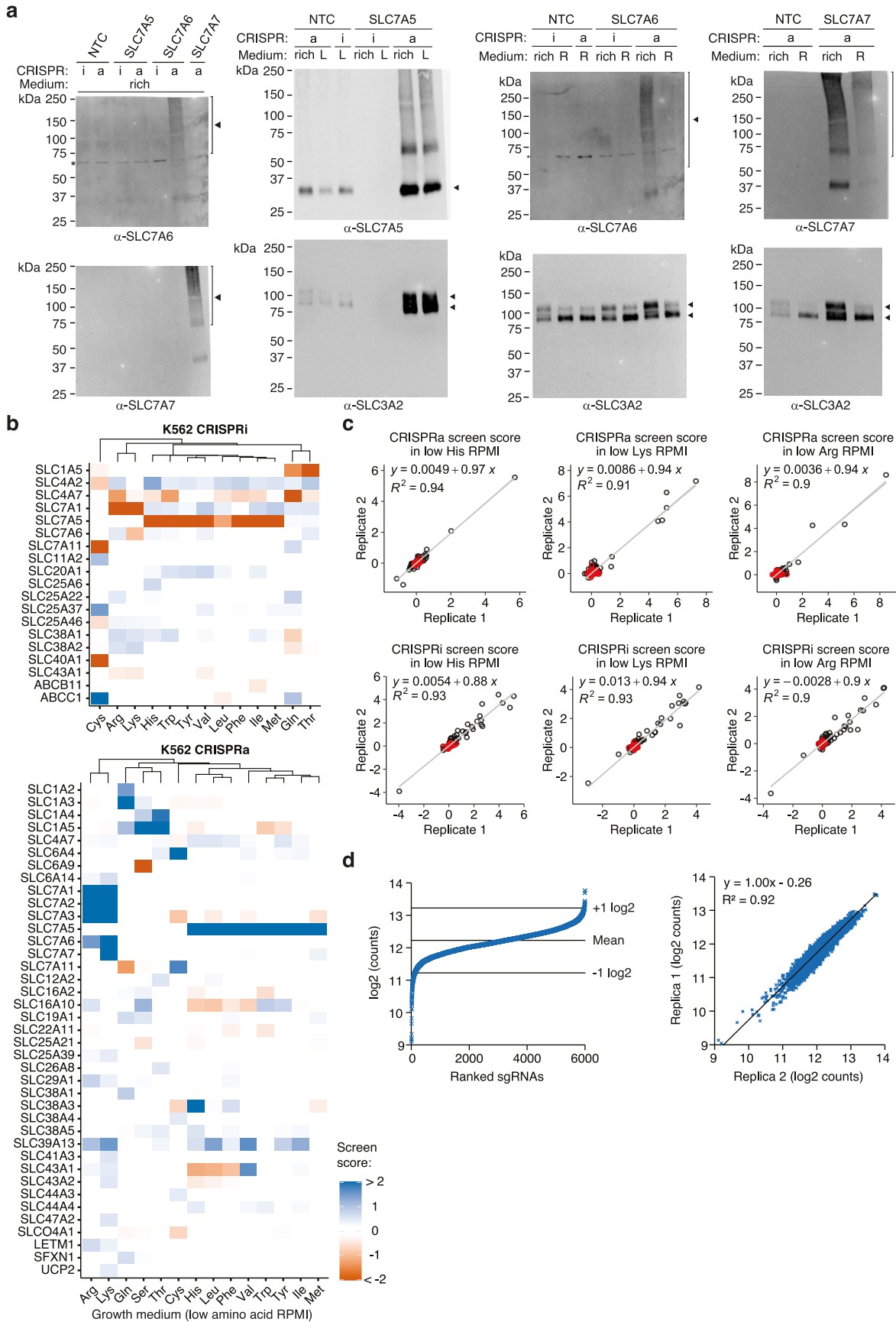

**Extended Data Fig. 2 | See next page for caption.**

**Extended Data Fig. 2 | CRISPRi/a transporter screens in low amino acid conditions.** (**a**) CRISPRi/a of transporters leads to changes in protein level at the plasma membrane. Changes in low Leu (L) and low Arg (R) conditions are similar to those observed in RPMI (rich). K562 CRISPRi/a cells were incubated with a cell-impermeable biotinylation reagent, and plasma membrane proteins were isolated by streptavidin affinity purification and analyzed by Western blotting. * denotes a non-specific band. (**b**) Tile plots displaying all significant transporter CRISPRi/a hits in low amino acid screens. Screen scores were calculated by multiplying phenotype scores by -log10(*P* value) for all genes and computed negative controls. *P* values were determined using a Mann-Whitney test. A stringent score cutoff was determined by the highest scoring negative control across all conditions for both datasets. All transporter genes that were significant in at least one condition tested were included in the tile plot. Transporter CRISPRi/a cell lines that have a growth defect in RPMI were prone to being pan-resistant in low amino acid conditions, as previously reported in other screens[25], and were removed from the analysis (see Methods). Columns were ordered based on hierarchical clustering of scores. (**c**) CRISPRi/a transporter screens are highly reproducible as shown by the strong correlation between independent screen replicates. Black circles represent individual transporter genes and red circles indicate negative control genes. (**d**) The custom sgRNA library is highly homogenous, and the library preparation introduces no sample bias. gDNA was isolated from a pool of K562 CRISPRi transporter library cells post-antibiotic selection, and the abundance of each sgRNA present in the library was determined after PCR amplification and high-throughput sequencing (Methods). For this representative sample, >97% of sgRNAs were within 1 log2 of the mean, 12 sgRNAs had less than 1000 counts, and no sgRNA had 0 counts. Source numerical data and unprocessed blots are available in source data.

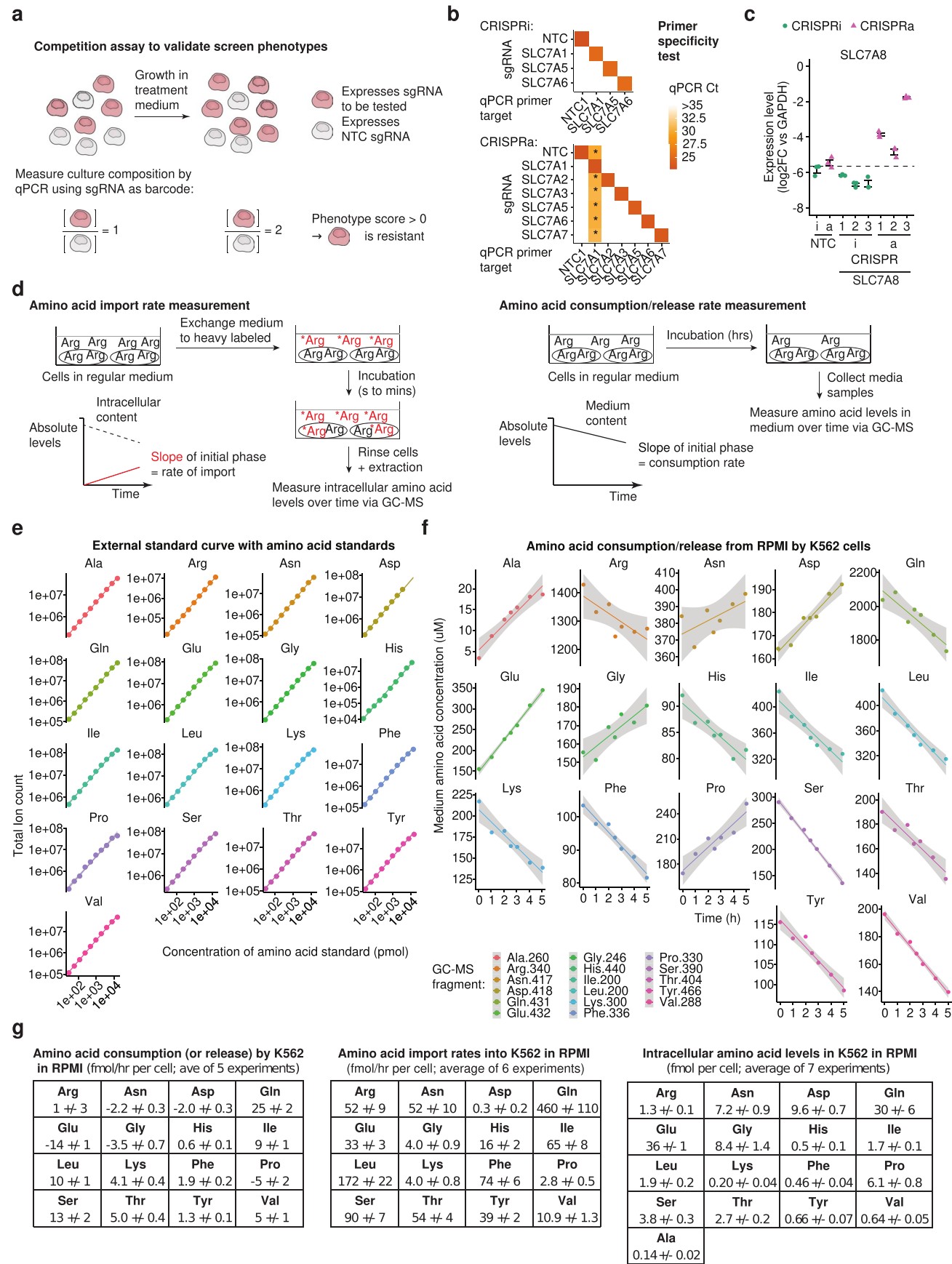

**a** Competition assay to validate screen phenotypes

**b** Primer specificity test

**c** SLC7A8

**d** Amino acid import rate measurement / Amino acid consumption/release rate measurement

**e** External standard curve with amino acid standards

**f** Amino acid consumption/release from RPMI by K562 cells

**g**

Amino acid consumption (or release) by K562 in RPMI (fmol/hr per cell; ave of 5 experiments)

| Arg | Asn | Asp | Gln |
|---|---|---|---|
| 1 +/- 3 | -2.2 +/- 0.3 | -2.0 +/- 0.3 | 25 +/- 2 |
| **Glu** | **Gly** | **His** | **Ile** |
| -14 +/- 1 | -3.5 +/- 0.7 | 0.6 +/- 0.1 | 9 +/- 1 |
| **Leu** | **Lys** | **Phe** | **Pro** |
| 10 +/- 1 | 4.1 +/- 0.4 | 1.9 +/- 0.2 | -5 +/- 2 |
| **Ser** | **Thr** | **Tyr** | **Val** |
| 13 +/- 2 | 5.0 +/- 0.4 | 1.3 +/- 0.1 | 5 +/- 1 |

Amino acid import rates into K562 in RPMI (fmol/hr per cell; average of 6 experiments)

| Arg | Asn | Asp | Gln |
|---|---|---|---|
| 52 +/- 9 | 52 +/- 10 | 0.3 +/- 0.2 | 460 +/- 110 |
| **Glu** | **Gly** | **His** | **Ile** |
| 33 +/- 3 | 4.0 +/- 0.9 | 16 +/- 2 | 65 +/- 8 |
| **Leu** | **Lys** | **Phe** | **Pro** |
| 172 +/- 22 | 4.0 +/- 0.8 | 74 +/- 6 | 2.8 +/- 0.5 |
| **Ser** | **Thr** | **Tyr** | **Val** |
| 90 +/- 7 | 54 +/- 4 | 39 +/- 2 | 10.9 +/- 1.3 |

Intracellular amino acid levels in K562 in RPMI (fmol per cell; average of 7 experiments)

| Arg | Asn | Asp | Gln |
|---|---|---|---|
| 1.3 +/- 0.1 | 7.2 +/- 0.9 | 9.6 +/- 0.7 | 30 +/- 6 |
| **Glu** | **Gly** | **His** | **Ile** |
| 36 +/- 1 | 8.4 +/- 1.4 | 0.5 +/- 0.1 | 1.7 +/- 0.1 |
| **Leu** | **Lys** | **Phe** | **Pro** |
| 1.9 +/- 0.2 | 0.20 +/- 0.04 | 0.46 +/- 0.04 | 6.1 +/- 0.8 |
| **Ser** | **Thr** | **Tyr** | **Val** |
| 3.8 +/- 0.3 | 2.7 +/- 0.2 | 0.66 +/- 0.07 | 0.64 +/- 0.05 |
| **Ala** | | | |
| 0.14 +/- 0.02 | | | |

**Extended Data Fig. 3 | See next page for caption.**

**Extended Data Fig. 3 | Measurement of amino acid import and consumption rates in K562 cells.** (**a**) A cartoon illustrating the competition assay used to validate growth phenotypes of CRISPRi/a cell lines. A CRISPRi/a cell line expressing a test sgRNA is mixed at a 1:1 ratio with a cell line expressing a non-targeting control (NTC) sgRNA. Cell line ratios pre- and post-treatment are determined by qPCR on gDNA extracted from cultures using primers specific to the sgRNA. Phenotype scores are determined by normalizing enrichments by the difference in population doublings. (**b**) Confirmation of the specificity of primers targeting sgRNAs in qPCR assays. gDNA extracted from K562 CRISPRi/a cells was amplified by qPCR with primers targeting specific and NTC sgRNA barcodes. Data are the cycle threshold (Ct) value of a representative experiment. Asterisks represent non-specific product amplification. (**c**) SLC7A8 expression level in K562 SLC7A8 CRISPRi/a cells determined by RT-qPCR relative to the housekeeping gene GAPDH. n = 3 technical replicates. Data are mean ± s.e.m. (**d**) Cartoons illustrating amino acid import and consumption rate determination.

The medium surrounding K562 cells attached to the surface of a Petri dish is rapidly exchanged to a similar medium where amino acids are heavy-isotope labelled. After extensive washing with PBS, intracellular metabolites are extracted and light and heavy amino acid levels are quantified by GC-MS. Import rates are determined from the slope of the increase in heavy amino acids over time. For consumption rate determination, the medium surrounding K562 cells is exchanged to fresh unlabelled medium and media samples are taken over time. Amino acid levels in samples are determined by GC-MS, and consumption rates are determined by the slope of the change in levels over time. (**e**) External standard curve used to calculate absolute levels of amino acids. Representative example of two independent experiments (**f**) Representative example of amino acid consumption from the medium of K562 cells growing in RPMI. Data (n = 7 biologically independent samples) were fit to a linear model and the shaded area represents 95% CI. (**g**) Numerical values of data displayed in Fig. 2c,d. Source numerical data are available in source data.

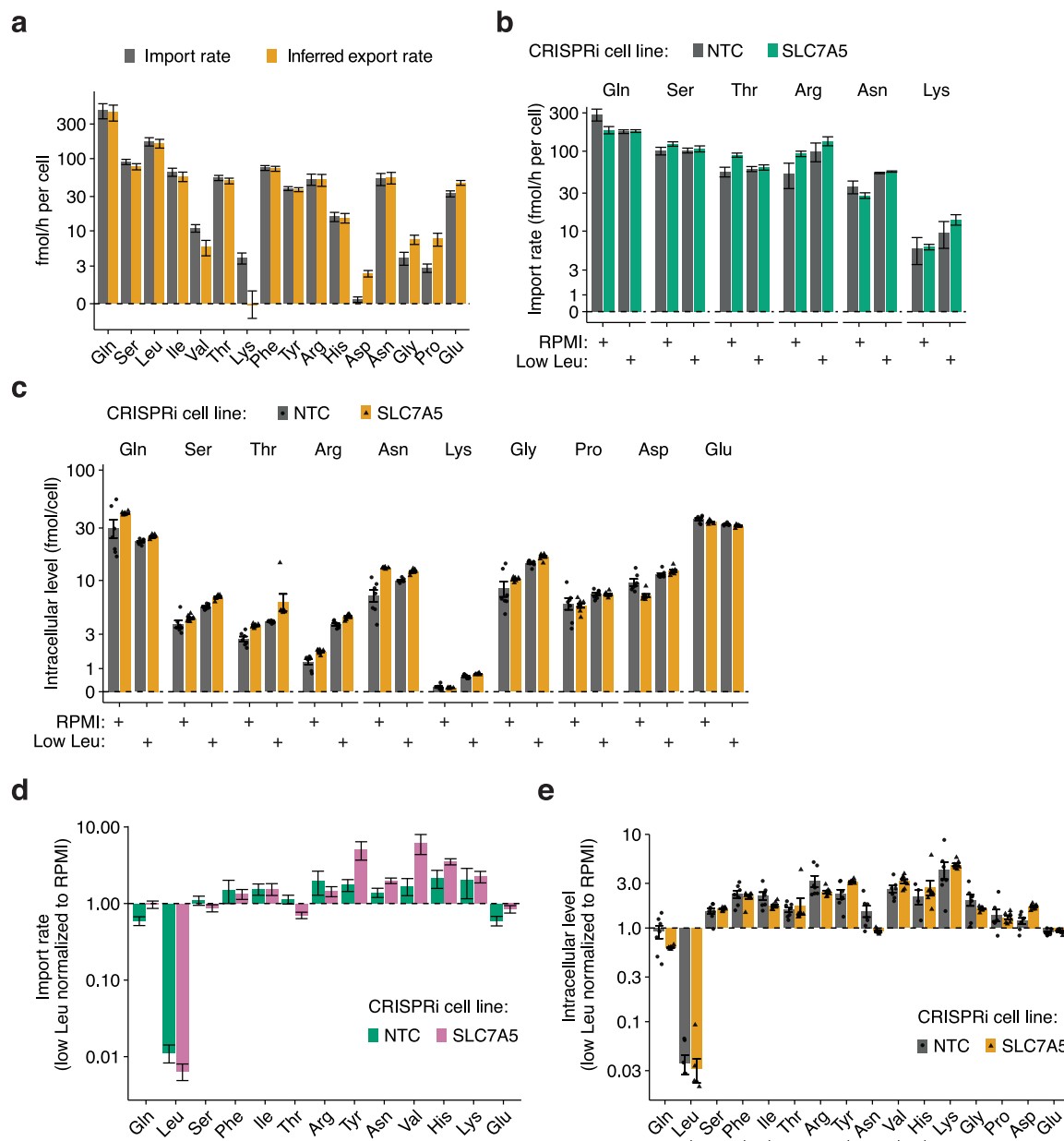

**Extended Data Fig. 4 | SLC7A5 CRISPRi specifically reduces transport of large neutral amino acids.** (**a**) Comparison of amino acid import rates and inferred export rates in K562 cells growing in RPMI. Import rates are from Fig. 2c, and export rates were calculated by subtracting consumption rates from import rates. (**b,c**) Additional data related to Fig. 2f,i for amino acids that are not SLC7A5 substrates. (**d**) Comparison of amino acid import rates in low leucine RPMI and in RPMI for K562 SLC7A5 and non-targeting control (NTC) CRISPRi determined in Fig. 2f. (**e**) Comparison of intracellular amino acid levels for K562 SLC7A5 and NTC CRISPRi in low leucine RPMI and in RPMI determined in Fig. 2i. (**d,e**) Asterisks indicate SLC7A5 substrates. Source numerical data are available in source data.

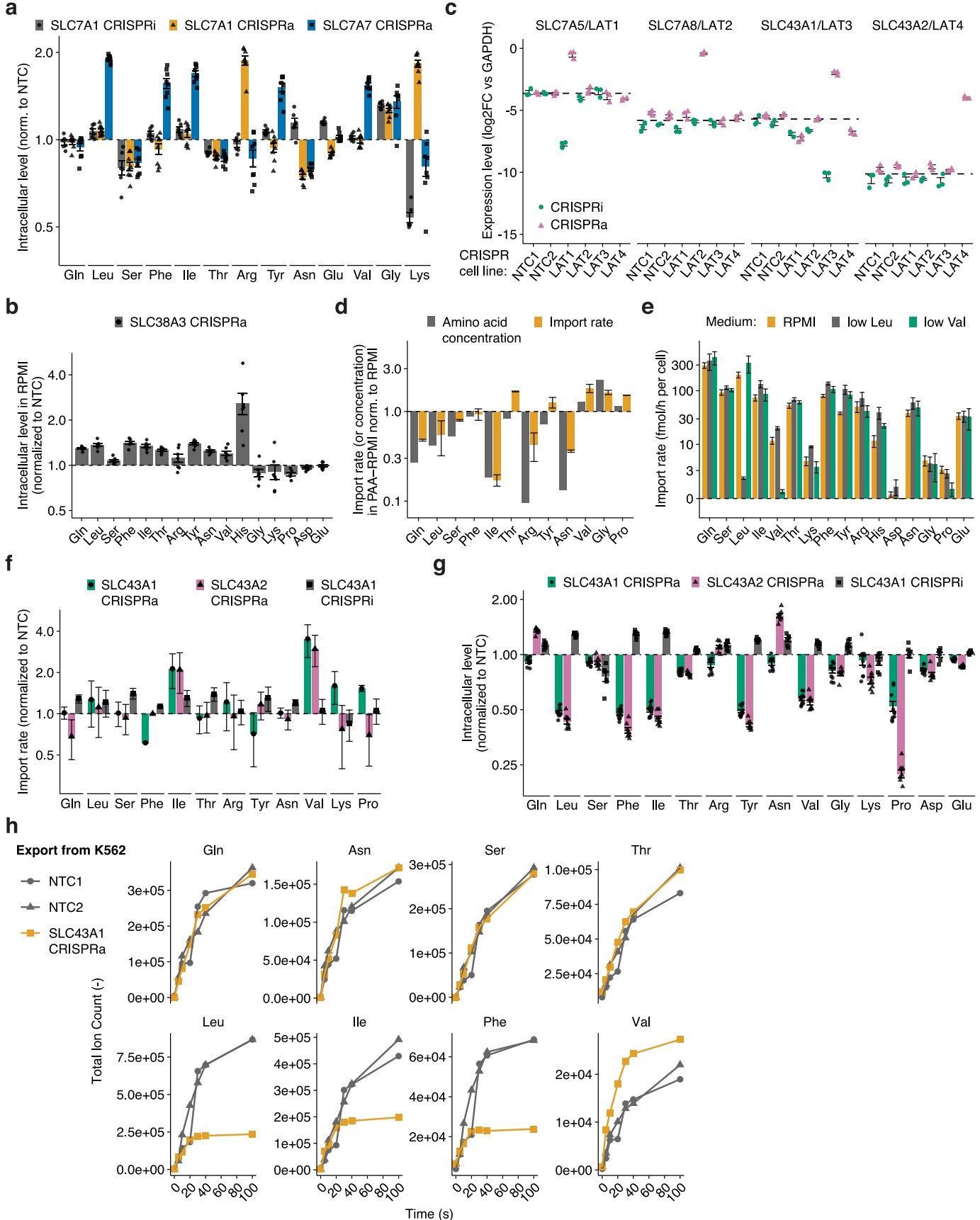

**Extended Data Fig. 5 | See next page for caption.**

**Extended Data Fig. 5 | Specific changes to amino acid levels and transport rates induced by CRISPRi/a of SLCs.** (**a**) Changes in intracellular amino acid levels induced by CRISPRi/a of SLC7A1 and SLC7A7. (**b**) Same as (a) but for SLC38A3 CRISPRa and data from Fig. 3d. (**c**) LAT1–4 expression levels in K562 CRISPRi/a cell lines were determined by RT-qPCR relative to the housekeeping gene GAPDH. n = 3 technical replicates. Data are mean ± s.e.m. (**d**) Amino acid import rates into K562 cells correlate with levels of amino acid in the growth medium. Import rates for K562 CRISPRa SLC43A1 and non-targeting control (NTC) in RPMI and PAA–RPMI were from Fig. 3f. Data represent the ratio of import rates in PAA–RPMI to that RPMI. Relative amino acid levels in the media were determined from their respective formulation. (**e**) The import of amino acids into K562 cells in low amino acid medium is selectively diminished for that specific low abundance amino acid. Data: mean ± s.e.m. of n = 2 (low Val), n = 3 (low Leu), n = 4 (RPMI) independent import rate determinations each

calculated from the linear regression of n = 6 biologically independent samples. (**f**) SLC43A2 CRISPRa increases import of isoleucine and valine into K562 cells in RPMI. Rates were determined from a linear regression of n = 6 biologically independent samples. Data represent the slope ± SE normalized to NTC. Data for SLC43A1 CRISPRa is from Fig. 3f. (**g**) SLC43A2 CRISPRa induces a decrease in intracellular levels of large neutral amino acids in K562 cells grown in RPMI. (**h**) SLC43A1 CRISPRa leads to higher export of valine and similar export of leucine, isoleucine, and phenylalanine despite lower intracellular pools. Cells grown in RPMI containing heavy-isotope labelled amino acids were rapidly washed, then incubated in regular RPMI. Accumulation of heavy-labelled amino acids in RPMI was monitored over time by GC-MS. Representative example of 3 independent experiments. (**a,b,d,g**) n = 7 biologically independent samples. Data are mean ± s.e.m. Source numerical data are available in source data.

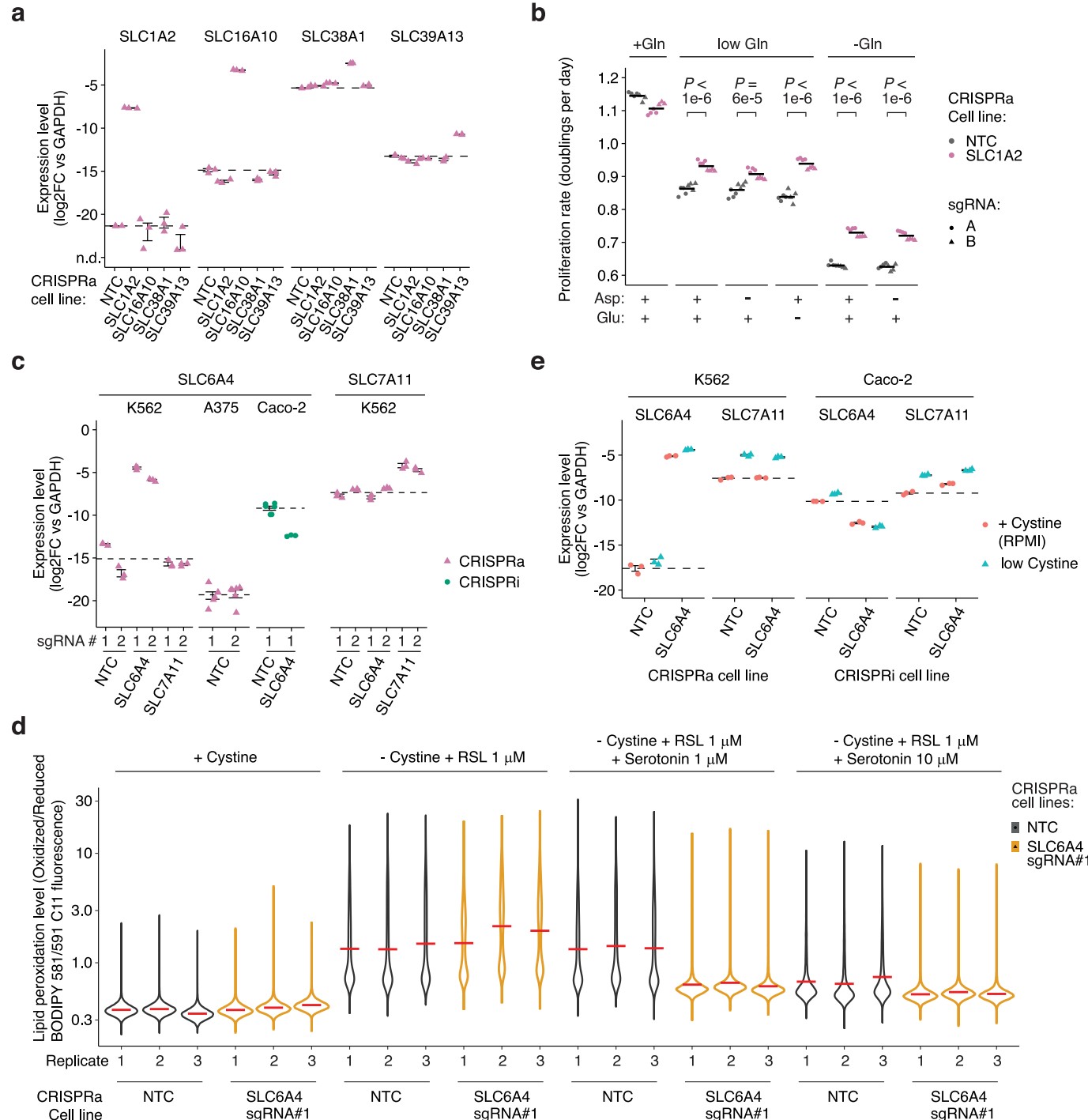

**Extended Data Fig. 6 | Specific changes to expression level of SLCs by CRISPRi/a.** (**a**) RT-qPCR analysis shows strong and specific gene upregulation by CRISPRa. n = 3 technical replicates. Data are mean ± s.e.m. (**b**) SLC1A2 CRISPRa confers a growth advantage in low and no glutamine RPMI in the presence or absence of either glutamate or aspartate in the medium. Proliferation rates were extracted from competition assays with 4 biologically independent samples from two independent sgRNAs. *P* values were determined using two-tailed unpaired Student's t-tests. (**c**) Expression level of SLC6A4 in cell lines used in this study and specific up- and down-regulation of SLC6A4 via CRISPRi/a. Levels of SLC6A4 and SLC7A11 were quantified by RT-qPCR relative to the housekeeping gene GAPDH in K562 CRISPRa, A375 CRISPRa, and Caco-2 CRISPRi cell lines with sgRNAs

targeting SLC6A4, SLC7A11 or a non-targeting control (NTC). n = 3 technical replicates, except for A375 and Caco-2 NTC where n = 6. Data are mean ± s.e.m. (**d**) Violin plots displaying the distribution of lipid peroxidation levels in single cells (~20k) as determined by flow cytometry of cells incubated with BODIPY 581/591 C11 sensor after growth in the mentioned conditions. Red bars represent the average peroxidation level of the population and were used in Fig. 4g. (**e**) Low cystine induces expression of SLC7A11 but not of SLC6A4. K562 and Caco-2 cells were grown in either RPMI or low cystine RPMI for 3 days with daily media changes. Expression levels were quantified by RT-qPCR relative to GAPDH (n = 3 technical replicates. Data are mean ± s.e.m.). Source numerical data are available in source data.

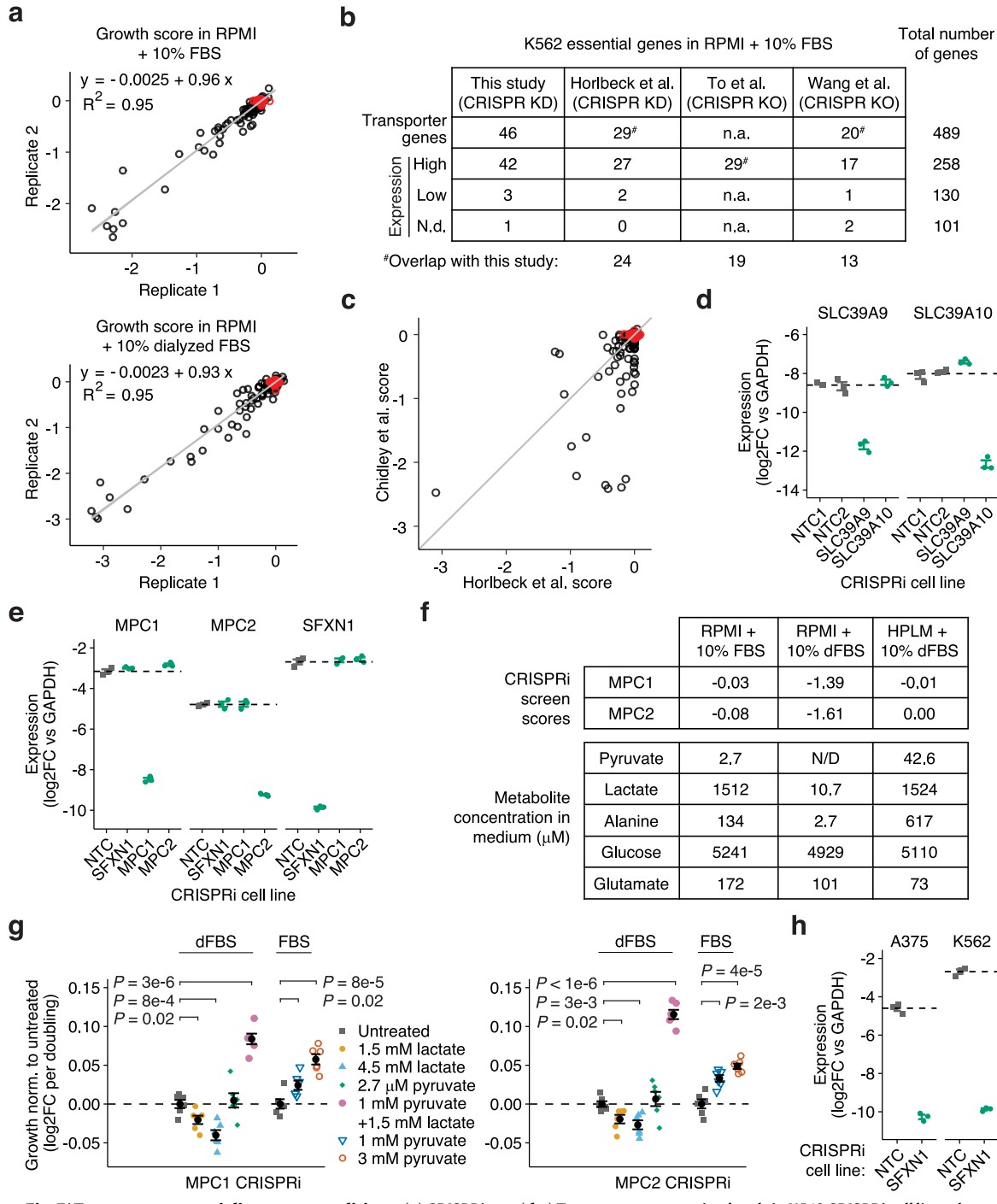

**Extended Data Fig. 7 | Transporter essentiality across conditions. (a)** CRISPRi transporter screens are highly reproducible as shown by the strong correlation of screen scores between independent replicates. Black circles represent individual transporter genes and red circles indicate computed negative control pseudogenes assembled from non-targeting control (NTC) sgRNAs. **(b)** Comparison of essential transporter genes identified in this study to previous screens in K562 cells grown in RPMI. The strong enrichment in expressed genes (98%) amongst the essential transporters in this study highlights the quality of the dataset. Essential transporters were determined by using a significance cutoff of q-value < 0.05 (ref. 37), and p < 0.05 and CS score negative[73] for the two CRISPRko screens (KO: knockout), and the first negative control gene for the genome-wide CRISPRi screen[34] (KD: knockdown). Genes were binned by expression level using publicly available data[37]. **(c)** Pairwise comparison of transporter growth scores determined in this study and in a genome-wide CRISPRi screen[34] in the same cell line, medium, and using the same gRNA library.

**(d,e)** Transporter expression levels in K562 CRISPRi cell lines determined by RT-qPCR and relative to the housekeeping gene GAPDH (n = 3 technical replicates. Data are mean ± s.e.m.). **(f)** Growth scores of K562 MPC1 and MPC2 CRISPRi determined in screens in different media and the concentration of a selection of metabolites in those media. Screen scores are from Figs. 5c, 6c and metabolite levels are from published data[22]. **(g)** Addition of pyruvate to RPMI + FBS or RPMI + dialyzed FBS (dFBS) alleviates the growth defect induced by CRISPRi of MPC1 or MPC2, and addition of lactate to RPMI + dFBS worsens the growth defect. Assays were performed as in Fig. 5g and data were normalized to untreated samples. n = 6 biologically independent samples. Data are mean ± s.e.m. P values were determined using two-tailed unpaired Student's t-tests. **(h)** Expression level of SFXN1 in K562 and A375 CRISPRi cell lines determined by RT-qPCR and relative to GAPDH (n = 3 technical replicates. Data are mean ± s.e.m.). Source numerical data are available in source data.

**a**

Determination of engraftment frequency in K562:

Proportion of GFP⁺ cells:

| # of cells injected: | 1:1000 GFP⁺ cells | 1:10,000 GFP⁺ cells | 1:100,000 GFP⁺ cells | 0 GFP⁺ cells |
|---|---|---|---|---|
| 10 mio | **10k GFP⁺ cells** | **1k GFP⁺ cells** | **100 GFP⁺ cells** | 0 GFP⁺ cells |
| 1 mio | **1k GFP⁺ cells** | **100 GFP⁺ cells** | 10 GFP⁺ cells | 0 GFP⁺ cells |
| 0.1 mio | 100 GFP⁺ cells | 10 GFP⁺ cells | 1 GFP⁺ cell | 0 GFP⁺ cells |

Subcutaneous injection into recipient mice (2 flanks per condition)

in vivo growth

Dissociation of tumor cells and detection of GFP⁺ cells by flow cytometry

Detection of GFP⁺ cells in tumor samples?

| | | | |
|---|---|---|---|
| YES | YES | YES | No |
| YES | YES | No | No |
| No | No | No | No |

→ **Engraftment rate of >1% when 1 or 10 mio cells are injected (<1% when 100k cells are injected)**

**b**

Distribution of non-targeting control (NTC) sgRNAs in samples (estimate of effect of experimental bottlenecks)

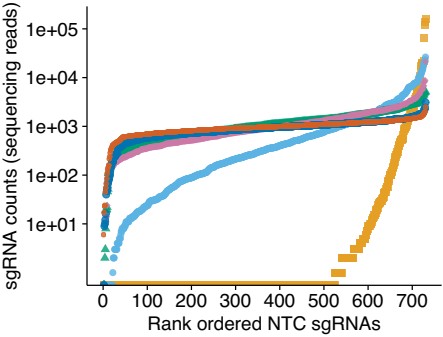

K562 CRISPRi samples:

- T0
- in vitro (RPMI 19 days)
- in vivo 10 mio cells replicate #1 (19 days)
- in vivo 10 mio cells replicate #2 (19 days)
- in vivo 1 mio cells (26 days)
- in vivo 100k cells (34 days)

**c**    **d**

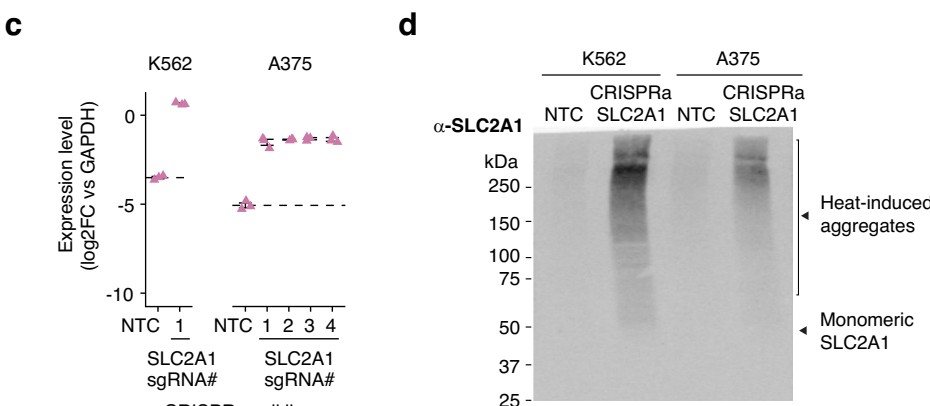

**e**

Preparation of parental K562 CRISPRi cell line

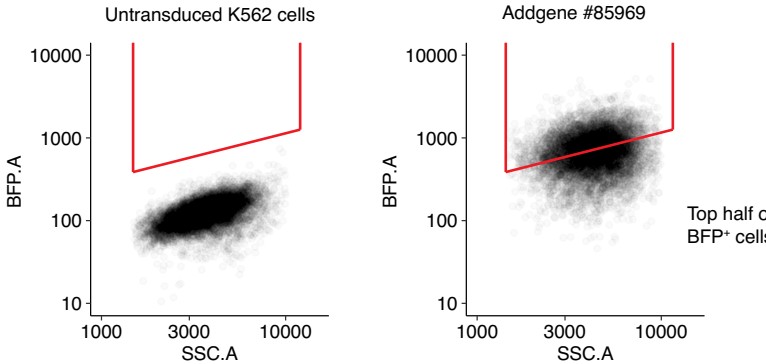

**Extended Data Fig. 8 | See next page for caption.**

**Extended Data Fig. 8 | Optimization of subcutaneous CRISPRi/a transporter screens.** (**a**) Determination of the engraftment frequency of K562 cells injected subcutaneously into immunodeficient mice. K562 CRISPRi library cells expressing GFP were mixed into pools of K562 CRISPRi library cells at a ratio ranging from 1:1000–1:100,000. 10, 1 or 0.1 million cells of these preparations were injected into the flanks of mice. Tumors formed over 19–34 days, were harvested and dissociated into single cell suspensions. The presence (or absence) of GFP+ cells in homogenized tumor samples was determined by flow cytometry and was used to determine the engraftment frequency and conditions required to preserve library complexity during in vivo transporter screens. (**b**) Plot displaying the number of counts for all non-targeting control (NTC) sgRNAs determined either from the sequencing of tumor samples from (a), from the initial pool of library cells, or from library cells grown in RPMI. (**c**) Expression level of SLC2A1 in K562 and A375 CRISPRa cell lines determined by RT-qPCR and relative to the housekeeping gene GAPDH. n = 3 technical replicates. Data are mean ± s.e.m. (**d**) Overexpression of SLC2A1 via CRISPRa leads to increased levels at the plasma membrane in K562 and A375 cells. Cells were incubated with a cell-impermeable biotinylation reagent, and plasma membrane proteins were isolated by streptavidin affinity purification and analyzed by Western blotting. (**e**) Example of gating strategy used for flow cytometry sorting of K562 cells transduced with lentivirus expressing dCas9-BFP-KRAB for K562 CRISPRi/a cell line generation. Source numerical data and unprocessed blots are available in source data.

# Reporting Summary

## Statistics

For all statistical analyses, confirm that the following items are present in the figure legend, table legend, main text, or Methods section.

| n/a | Confirmed | |
|---|---|---|
| ☐ | ☒ | The exact sample size ($n$) for each experimental group/condition, given as a discrete number and unit of measurement |
| ☐ | ☒ | A statement on whether measurements were taken from distinct samples or whether the same sample was measured repeatedly |
| ☐ | ☒ | The statistical test(s) used AND whether they are one- or two-sided *Only common tests should be described solely by name; describe more complex techniques in the Methods section.* |
| ☒ | ☐ | A description of all covariates tested |
| ☐ | ☒ | A description of any assumptions or corrections, such as tests of normality and adjustment for multiple comparisons |
| ☐ | ☒ | A full description of the statistical parameters including central tendency (e.g. means) or other basic estimates (e.g. regression coefficient) AND variation (e.g. standard deviation) or associated estimates of uncertainty (e.g. confidence intervals) |
| ☐ | ☒ | For null hypothesis testing, the test statistic (e.g. $F$, $t$, $r$) with confidence intervals, effect sizes, degrees of freedom and $P$ value noted *Give P values as exact values whenever suitable.* |
| ☒ | ☐ | For Bayesian analysis, information on the choice of priors and Markov chain Monte Carlo settings |
| ☒ | ☐ | For hierarchical and complex designs, identification of the appropriate level for tests and full reporting of outcomes |
| ☐ | ☒ | Estimates of effect sizes (e.g. Cohen's $d$, Pearson's $r$), indicating how they were calculated |

*Our web collection on statistics for biologists contains articles on many of the points above.*

## Software and code

Policy information about availability of computer code

**Data collection**   Western blots were imaged on an ImageQuant LAS4000 (Cytiva) or an iBright FL1500 Imaging System (Invitrogen).
Luminescence from Cell-Titer Glo assays was measured on a plate reader (BioTek Synergy H1).
qPCR results were collected on a QuantStudio 6 Real-Time PCR system (Thermo Fisher Scientific).
The composition of pooled libraries was determined by sequencing using an Illumina HiSeq 2500 platform and a 50 bp single read on a high output standard v4 flow cell, or using an Illumina NextSeq 500 platform and a 75 bp single read on a high output flow cell.
Flow cytometry was collected on a BD LSRFortessa or a BD Biosciences LSR II.
Derivatized amino acid samples were analyzed on an Agilent 7890B gas chromatograph linked to an Agilent 5977B mass spectrometer.
No other specific software or code was used for data collection.

**Data analysis**   CRISPR screens were analyzed using an open source code available at https://github.com/mhorlbeck/ScreenProcessing.
All other data was analyzed using R (v. 3.6.0). General Data visualization was performed using ggplot2 (v. 3.3.3). Chord plots were generated with GOplot (v. 1.0.2). Pairwise Pearson correlations (and P values) were calculated using the function rcorr in the R package Hmisc (v. 4.7.1) and data was displayed using the R package corrplot (v. 0.84).
GC-MS raw data was quantified using El-MAVEN (v. 0.11.0) and natural isotope abundances were corrected using IsoCorrectoR (v. 1.2.1).

For manuscripts utilizing custom algorithms or software that are central to the research but not yet described in published literature, software must be made available to editors and reviewers. We strongly encourage code deposition in a community repository (e.g. GitHub). See the Nature Portfolio guidelines for submitting code & software for further information.

## Data

All CRISPR screening data are provided in Supplementary Tables 5,6. The composition of the sgRNA libraries is provided in Supplementary Table 1. Source data have been provided in Source Data files. All other data supporting the findings of this study are available from the corresponding author on reasonable request.

## Research involving human participants, their data, or biological material

| | |
|---|---|
| Reporting on sex and gender | not applicable |
| Reporting on race, ethnicity, or other socially relevant groupings | not applicable |
| Population characteristics | not applicable |
| Recruitment | not applicable |
| Ethics oversight | not applicable |

Note that full information on the approval of the study protocol must also be provided in the manuscript.

# Field-specific reporting

Please select the one below that is the best fit for your research. If you are not sure, read the appropriate sections before making your selection.

☒ Life sciences    ☐ Behavioural & social sciences    ☐ Ecological, evolutionary & environmental sciences

# Life sciences study design

All studies must disclose on these points even when the disclosure is negative.

| | |
|---|---|
| Sample size | We performed experiments to determine the implantation rate of K562 leukemia cells in the subcutaneous flank tissue of NSG mice. Our initial experiments determined that sufficient library representation was achieved in each single tumor. We decided to have 6 replicate tumors for each screen.<br>No sample size calculation was performed for experiments in cell culture. The sample size for each experiment is included in the respective figure legend. We included a minimum of 3 biologically independent experiments. |
| Data exclusions | To improve data quality in the in vivo CRISPR screens, we excluded in an unbiased way 2 datasets from the 6 replicates that were collected for each condition. We excluded the 2 datasets with the highest amount of technical noise, which was determined by the lowest number of NTC sgRNA counts that were within 1 log2 of the median after growth in vivo. |
| Replication | Transporter CRISPR screens were run in duplicates to quantify reproducibility. All attempts at replication of transporter CRISPR screens were successful, as shown in Extended Data Fig 2c,d.<br>All screens include 10 sgRNAs per gene and non targeting controls to evaluate biological noise. The number of independent replicates for all other experiments is mentioned in the corresponding figure legend. |
| Randomization | Mice were randomly chosen for the 4 subcutaneous injection conditions. All pooled CRISPR screens are inherently randomized. Other experiments were not randomized as there was no allocation into different experimental groups. |
| Blinding | The researcher performing subcutaneous injections was blinded to the identity of the cells injected. All pooled CRISPR screens are inherently blinded. Other experiments were not blinded as all data was collected using unbiased methods. |

# Reporting for specific materials, systems and methods

We require information from authors about some types of materials, experimental systems and methods used in many studies. Here, indicate whether each material, system or method listed is relevant to your study. If you are not sure if a list item applies to your research, read the appropriate section before selecting a response.

## Materials & experimental systems

| n/a | Involved in the study |
|---|---|
| ☐ | ☒ Antibodies |
| ☐ | ☒ Eukaryotic cell lines |
| ☒ | ☐ Palaeontology and archaeology |
| ☐ | ☒ Animals and other organisms |
| ☒ | ☐ Clinical data |
| ☒ | ☐ Dual use research of concern |
| ☒ | ☐ Plants |

## Methods

| n/a | Involved in the study |
|---|---|
| ☒ | ☐ ChIP-seq |
| ☐ | ☒ Flow cytometry |
| ☒ | ☐ MRI-based neuroimaging |

## Antibodies

| | |
|---|---|
| Antibodies used | Cell Signaling Technology primary rabbit antibodies: vinculin (E1E9V) #13901 Dilution 1:2000; Phospho-p70 S6 Kinase (Thr389) #9234 Dilution 1:1000; p70 S6 Kinase #9202 Dilution 1:1000; Phospho-eIF2α (Ser51) (D9G8) #3398 Dilution 1:1000; eIF2α #9722 Dilution 1:2000;  Phospho-4E-BP1 (Ser65) #9451 Dilution 1:1000; 4E-BP1 (53H11) #9644 Dilution 1:2000;  SLC3A2/4F2hc (D603P) #13180 Dilution 1:1000; SLC2A1 (D3J3A) #12939 Dilution 1:1000. <br> Other primary antibodies: SLC7A5 (Proteintech, 28670-1-AP) Dilution 1:5000; SLC7A6 (Novus Biologicals, NBP2-75086) Dilution 1:500; SLC7A7 (Novus Biologicals, NBP1-82826) Dilution 1:500. <br> Secondary antibodies: anti-rabbit secondary antibody (IRDye800CW LI-COR 926-32211) Dilution 1:20,000; anti-goat secondary antibody (Donkey IgG H&L, Alexa Fluor 750 conjugate, Abcam ab175744) Dilution 1:10,000; goat HRP-linked Anti rabbit IgG (Cell Signaling Technology #7074) Dilution 1:5000. |
| Validation | Cell Signaling Technology primary antibodies used in this study are well-published commercial antibodies. Validation statements for use of these antibodies in Western blots and literature references are available on the manufacturer's website (https://www.cellsignal.com/). <br> Other primary antibodies have validation statements on the supplier's website for use in Western blots (https://www.ptglab.com/ and https://www.novusbio.com/). <br> The specificity for antibodies against SLC2A1, SLC7A5, SLC7A6, SLC7A7 was confirmed in this study using CRISPRi/a cell lines. |

## Eukaryotic cell lines

Policy information about cell lines and Sex and Gender in Research

| | |
|---|---|
| Cell line source(s) | Cell lines were from ATCC: K562 (CCL–243), A375 (CRL–1619), C2BBe1 clone of Caco-2 (CRL–2102), HEK293T (CRL–3216). |
| Authentication | Cells lines were used from low passage from ATCC stocks. Cell lines were not further authenticated. |
| Mycoplasma contamination | All cell lines were confirmed mycoplasma free using the MycoAlert mycoplasma detection kit (Lonza LT07-318). |
| Commonly misidentified lines (See ICLAC register) | none |

## Animals and other research organisms

Policy information about studies involving animals; ARRIVE guidelines recommended for reporting animal research, and Sex and Gender in Research

| | |
|---|---|
| Laboratory animals | NOD.Cg-Prkdcscid Il2rgtm1Wjl/SzJ mice (NSG mice; The Jackson Laboratory, Strain #005557). Male mice between 3 and 4 months old were used in this study. |
| Wild animals | Study did not involve wild animals. |
| Reporting on sex | Male mice were used in this study |
| Field-collected samples | Study did not involve Field-collected samples |
| Ethics oversight | All animal experiments conducted in this study were approved by the Massachusetts Institute of Technology (MIT) Institutional Animal Care and Use Committee (IACUC). |

Note that full information on the approval of the study protocol must also be provided in the manuscript.

# Flow Cytometry

## Plots

Confirm that:

☒ The axis labels state the marker and fluorochrome used (e.g. CD4-FITC).

☒ The axis scales are clearly visible. Include numbers along axes only for bottom left plot of group (a 'group' is an analysis of identical markers).

☒ All plots are contour plots with outliers or pseudocolor plots.

☒ A numerical value for number of cells or percentage (with statistics) is provided.

## Methodology

| | |
|---|---|
| Sample preparation | K562 cells in culture in RPMI. Washed once with PBS and strained through 35 micron mesh filter |
| Instrument | BD FACS Aria II |
| Software | BD FACSDiva 8.0.1 |
| Cell population abundance | BFP+ cells represented about 30-50% of initial population. We performed 2 rounds of cell sorting, isolating the top half of BFP + cells and then single cell cloned. The resulting line is 100% pure |
| Gating strategy | Untransduced K562 cells were used to define BFP- cells. We determined the BFP+ cells in transduced K562 and gated on the top half of this population. |

☒ Tick this box to confirm that a figure exemplifying the gating strategy is provided in the Supplementary Information.

