## [Peer Review File · Nature Cell Biology]

Peer Review Information

Journal: Nature Cell Biology

Manuscript Title: A CRISPRi/a screening platform to study cellular nutrient transport in diverse microenvironments

Corresponding author name(s): Dr Christopher Chidley

Editorial Notes:

Reviewer Comments & Decisions:

Decision Letter, initial version:
--

*Please delete the link to your author homepage if you wish to forward this email to co-authors.

Dear Dr Chidley,

Thank you for submitting your manuscript, "A CRISPRi/a screening platform to study cellular nutrient transport in diverse microenvironments", to Nature Cell Biology. The manuscript has now been seen by 3 referees, who are experts in cancer metabolism, nutrient dependencies (Referee #1); nutrient transport (Referee #2); and CRISPR screens, metabolism (Referee #3). As you will see from their comments (attached below), they found this work of potential interest but have raised substantial concerns, which in our view would need to be addressed with considerable revisions before we can consider publication in Nature Cell Biology.

Nature Cell Biology editors discuss the referee reports in detail within the editorial team, including the chief editor, to identify key referee points that should be addressed with priority, and requests that are overruled as being beyond the scope of the current study. To guide the scope of the revisions, I have listed these points below. Our standard revision period is six months, and we are committed to providing a fair and constructive peer-review process, so please feel free to contact me if you would like to discuss any of the referee comments further or if you anticipate any issues or delays addressing the reviews.

You will see that Rev#1 was not particularly enthusiastic about the degree of advance, from a broad cell biological perspective. While the reviewer made some good points, we feel that the study has the potential to provide a useful and interesting Resource for the field, and would therefore invite you to revise the study and resubmit it as a Resource. Reviewers #2 and #3 had consistent and overlapping points that we feel should be addressed rigorously:

- (i) Please dedicate efforts in revision to address the reviewers' comments about the clarity of the manuscript throughout, the descriptions of the results, and scholarly discussion of the data in the current context of the field.
- (ii) Please provide further validation of the hits and strengthen the biological conclusions: Rev#2 "Line 214" paragraph; "Line 256" paragraph; and Rev#3's points #1 and #2
- (iii) Finally, please pay close attention to our guidelines on statistical and methodological reporting (listed below) as failure to do so may delay the reconsideration of the revised manuscript. In particular, please provide:

We would be happy to consider a revised manuscript that would satisfactorily address these points, unless a similar paper is published elsewhere, or is accepted for publication in Nature Cell Biology in the meantime.

- ensure that it conforms to our format instructions and publication policies (see below and www.nature.com/nature/authors/).

- provide a point-by-point rebuttal to the full referee reports verbatim, as provided at the end of this letter.

- provide the completed Editorial Policy Checklist (found here <https://www.nature.com/authors/policies/Policy.pdf>), and Reporting Summary (found here <https://www.nature.com/authors/policies/ReportingSummary.pdf>). This is essential for reconsideration of the manuscript and these documents will be available to editors and referees in the event of peer review. For more information see <http://www.nature.com/authors/policies/availability.html> or contact me.

Nature Cell Biology is committed to improving transparency in authorship. As part of our efforts in this direction, we are now requesting that all authors identified as 'corresponding author' on published papers create and link their Open Researcher and Contributor Identifier (ORCID) with their account on the Manuscript Tracking System (MTS), prior to acceptance. ORCID helps the scientific community achieve unambiguous attribution of all scholarly contributions. You can create and link your ORCID from the home page of the MTS by clicking on 'Modify my Springer Nature account'. For more information please visit www.springernature.com/orcid.

[Redacted]

We hope that you will find our referees' comments and editorial guidance helpful. Please do not

hesitate to contact me if there is anything you would like to discuss. Thank you again for considering the journal for your work.

Best wishes,

Melina

Melina Casadio, PhD
Senior Editor, Nature Cell Biology
ORCID ID: <https://orcid.org/0000-0003-2389-2243>

Reviewers' Comments:

Reviewer #1:

Remarks to the Author:

The metabolic demands of unrestrained proliferation render cancer cells exquisitely dependent on the ability to acquire a variety of nutrients from the tumor microenvironment and, therefore, dependent on transporters that permit nutrient uptake across the cell membrane. Therefore, a thorough understanding of specific nutrient transporters required to support cell proliferation/cell survival has potential to identify novel metabolic vulnerabilities of cancer cells and new targets for cancer therapy. In the current study, Chidley and colleagues describe a platform that permits systematic interrogation of the contributions of individual members of the solute carrier (SLC) and ATP-binding cassette (ABC) transporter superfamilies to nutrient uptake in human cells. The authors employ their CRISPR interference (CRISPRi) and activation (CRISPRa) screening platform to identify critical cellular nutrient transporters, to characterize transporter function and to investigate the impact of the tumor microenvironment on cancer cell metabolism.

This is a very thorough study that reports a powerful platform for investigating the nutrient dependencies of cancer cells. The manuscript describes an elegant series of experiments, and all conclusions are supported by robust data that are clearly presented. As such, there are no specific experimental concerns that need to be addressed. However, a major concern exists with respect to the novelty of the current study. With respect to the screening approach, Rebsamen and colleagues recently reported a SLC-focussed CRISPR gain-of-function screening approach to identify solute carriers able to sustain proliferation under amino acid-deprived conditions (DOI: 10.26508/lsa.202201404) and metabolic CRISPRi/a libraries, which encompass SLC and ABC transporters, have now been employed in a number of in vitro and in vivo studies (e.g., DOI: 10.1016/j.cmet.2020.10.017, DOI: 10.1016/j.molcel.2022.05.024). Moreover, with respect to the authors' claim that they have discovered a novel role for serotonin in the regulation of ferroptosis, Liu and colleagues recently reported this very finding (DOI: 10.1002/adv.202204006). The lack of novelty limits the impact of the current study.

Reviewer #2:

Remarks to the Author:

A systematic crispr screen was developed to investigate knockdown and overexpression of solute

carriers. To enhance the growth phenotype, cells were cultivated in amino acid limiting media. The screen confirms many known transport activities critical for cancer cells such as *slc7a5* for BCAA and AAA, *slc38a1,2* and *slc1a5* for glutamine and *slc7a1* and *slc7a6* for cationic AA. The screen also reveals some surprising findings, which are poorly discussed but are more worthwhile investigating.

Line 128 and Fig. 1: Please clarify whether you mean cysteine (cys) or cystine. In the figure the amino acid is listed as cysteine. Please note that cysteine added to media will oxidise to cystine in a matter of hours/days.

Fig. 1 Please state explicitly, whether a negative score means the sgRNA is depleted or enriched or better whether a particular mRNA was silenced or overexpressed. Also why are *slc38a1* and *a2* sgRNA enriched under low arginine and low lysine conditions?

Line 186: Should this not read "with one significantly depleted sgRNA?"

Line 214: I am not convinced that SLC1A2 transports glutamine. Most likely SLC1A2 transported glutamate, which was converted to glutamine by glutaminase.

Line 256: Absence of growth effect in low Arg but not low Lys. This is probably due to the effect that lysine is an essential amino acid while arginine can be synthesized if ASS is expressed. The interpretation that SLC7A7 may not import arginine under growth limiting conditions is not well justified (see Gauthier-Coles, G.; Fairweather, S.J.; Bröer, A.; Bröer, S. Do Amino Acid Antiporters Have Asymmetric Substrate Specificity? *Biomolecules* 2023, 13, 301. <https://doi.org/10.3390/biom13020301>).

Line 270: Please note that dependence on arginine is reduced in the presence of citrulline.

Line 274: It is unlikely that a phenotypic score for *slc7a8* is observed when *slc7a5* is expressed at high levels.

Line 407: Please note that not all of these amino acids are substrates of SLC43A1 and that reduction of intracellular pools occurs indirectly through exchangers (see Gauthier-Coles, Gregory, et al. "Quantitative modelling of amino acid transport and homeostasis in mammalian cells." *Nature communications* 12.1 (2021): 5282. *Slc43a1* has been characterised as a controller transporter for these reasons.

Line 638: Please provide more guidance and explanation for Fig. 6A/B. If this is to be read in the same way as Fig. 1 a negative score would mean that the corresponding sgRNA were depleted.

Reviewer #3:

Remarks to the Author:

The authors presented a collection of CRISPRi/a screens targeting SLCs and ABCs in K562 and A375 cells, as well as in vivo tumors. The K562 screen data, focusing on the interplay between membrane transporters and amino acids, are potentially useful resources for the community. However, there are two main areas where the manuscript could be improved.

(1) The SLC-targeting CRISPRi/a screens in K562 require additional hit validations and information.

first, a more detailed description of the custom library is needed. It would be helpful to provide the number of SLCs, ABCs, and protein families in the library, and how many of these are expressed in K562 cells. Additionally, the number of plasma membrane transporters and putative amino acid transporters would be relevant.

second, because that genes involved in amino acid metabolism are often regulated transcriptionally by

amino acid availability, including transporter genes, it would be worthwhile to profile transporter transcript levels under low amino acid CRISPR screen conditions. This could be particularly important for screen hits that might be transcriptionally regulated by low amino acids, independent of CRISPRi/a. The authors have already comments on the GCN2 and mTOR pathways. Other pathways include ATF4 regulated by serine.

Plasma membrane transporter hit validation by CRISPRi/a needs to be strengthened. The manuscript should include gene expression data demonstrating the corresponding direction of changes by CRISPRi/a, similar to what was shown for SLC7 family genes. For a few plasma membrane transporters characterized in detail, it is essential to measure their plasma membrane protein levels and compare them under the CRISPRi/a condition. For heterodimer transporters, the author might consider comment whether the levels of other subunits change upon CRISPRi/a of one subunit, for instance, for the screen hit SLC7 genes and the lack of phenotype for SLC7A8.

(2) The findings related to the serotonin transporter SLC6A4 are surprising and need further investigation. Will the plasma membrane transporter level increase in low cysteine conditions and/or increases due to CRISPRa. The concentration of 5HT in regular culture media needs to be measured, and are the concentrations used in the study (1uM, 10uM) comparable to physiological levels. A directly measure the effect of 5HT on lipid peroxidation is needed, for example, by using the Bodipy C11 sensor.

Minor:

In Figure 1c, it would be helpful to list the concentrations of all the amino acids tested, considering that different RPMI formulations may have slightly different amino acid compositions and concentrations. Similarly, in Figures 1c, 1f, and 1g, the exact amino acid concentrations should be included in the main figure.

Although not the main focus of the manuscript, the references on recently characterized mitochondrial metabolite transporters should include BCAA (Yoneshiro 2019) and glutathione (Wang 2021 and Shi 2021 [BioRxiv]/2022), as well as other more relevant plasma membrane transporters, heme/SLCO2B1, choline/FLVCR1, and cGAMP/SLC19A1.

The data presentation could be improved with a clearer logical flow. For instance, experimental data on SLC7 family transporters are the main focus but are scattered throughout the manuscript with little connection.

READABILITY OF MANUSCRIPTS – Nature Cell Biology is read by cell biologists from diverse

backgrounds, many of whom are not native English speakers. Authors should aim to communicate their findings clearly, explaining technical jargon that might be unfamiliar to non-specialists, and avoiding non-standard abbreviations. Titles and abstracts should concisely communicate the main findings of the study, and the background, rationale, results and conclusions should be clearly explained in the manuscript in a manner accessible to a broad cell biology audience. Nature Cell Biology uses British spelling.

REFERENCES – are limited to a total of 70 for Articles, Resources, Technical Reports; and 40 for Letters. This includes references in the main text and Methods combined. References must be numbered sequentially as they appear in the main text, tables and figure legends and Methods and must follow the precise style of Nature Cell Biology references. References only cited in the Methods should be numbered consecutively following the last reference cited in the main text. References only associated with Supplementary Information (e.g. in supplementary legends) do not count toward the

total reference limit and do not need to be cited in numerical continuity with references in the main text. Only published papers can be cited, and each publication cited should be included in the numbered reference list, which should include the manuscript titles. Footnotes are not permitted.

Methods should be written concisely, but should contain all elements necessary to allow interpretation and replication of the results. As a guideline, Methods sections typically do not exceed 3,000 words. The Methods should be divided into subsections listing reagents and techniques. When citing previous methods, accurate references should be provided and any alterations should be noted. Information must be provided about: antibody dilutions, company names, catalogue numbers and clone numbers for monoclonal antibodies; sequences of RNAi and cDNA probes/primers or company names and catalogue numbers if reagents are commercial; cell line names, sources and information on cell line identity and authentication. Animal studies and experiments involving human subjects must be reported in detail, identifying the committees approving the protocols. For studies involving human subjects/samples, a statement must be included confirming that informed consent was obtained. Statistical analyses and information on the reproducibility of experimental results should be provided in a section titled "Statistics and Reproducibility".

All Nature Cell Biology manuscripts submitted on or after March 21 2016 must include a Data availability statement at the end of the Methods section. For Springer Nature policies on data availability see <http://www.nature.com/authors/policies/availability.html>; for more information on this particular policy see <http://www.nature.com/authors/policies/data/data-availability-statements-data-citations.pdf>. The Data availability statement should include:

- Accession codes for primary datasets (generated during the study under consideration and designated as "primary accessions") and secondary datasets (published datasets reanalysed during the study under consideration, designated as "referenced accessions"). For primary accessions data should be made public to coincide with publication of the manuscript. A list of data types for which submission to community-endorsed public repositories is mandated (including sequence, structure, microarray, deep sequencing data) can be found here <http://www.nature.com/authors/policies/availability.html#data>.
- Unique identifiers (accession codes, DOIs or other unique persistent identifier) and hyperlinks for datasets deposited in an approved repository, but for which data deposition is not mandated (see here for details <http://www.nature.com/sdata/data-policies/repositories>).
- At a minimum, please include a statement confirming that all relevant data are available from the authors, and/or are included with the manuscript (e.g. as source data or supplementary information), listing which data are included (e.g. by figure panels and data types) and mentioning any restrictions on availability.
- If a dataset has a Digital Object Identifier (DOI) as its unique identifier, we strongly encourage including this in the Reference list and citing the dataset in the Methods.

We recommend that you upload the step-by-step protocols used in this manuscript to the Protocol

Exchange. More details can found at www.nature.com/protocolexchange/about.

All imaging data should be accompanied by scale bars, which should be defined in the legend. Cropped images of gels/blots are acceptable, but need to be accompanied by size markers, and to retain visible background signal within the linear range (i.e. should not be saturated). The boundaries of panels with low background have to be demarked with black lines. Splicing of panels should only be considered if unavoidable, and must be clearly marked on the figure, and noted in the legend with a statement on whether the samples were obtained and processed simultaneously. Quantitative comparisons between samples on different gels/blots are discouraged; if this is unavoidable, it should only be performed for samples derived from the same experiment with gels/blots were processed in parallel, which needs to be stated in the legend.

The total number of Supplementary Figures (not including the "unprocessed scans" Supplementary

Figure) should not exceed the number of main display items (figures and/or tables (see our Guide to Authors and March 2012 editorial <http://www.nature.com/ncb/authors/submit/index.html#suppinfo>; <http://www.nature.com/ncb/journal/v14/n3/index.html#ed>). No restrictions apply to Supplementary Tables or Videos, but we advise authors to be selective in including supplemental data.

GUIDELINES FOR EXPERIMENTAL AND STATISTICAL REPORTING

REPORTING REQUIREMENTS – To improve the quality of methods and statistics reporting in our papers we have recently revised the reporting checklist we introduced in 2013. We are now asking all life sciences authors to complete two items: an Editorial Policy Checklist (found here <https://www.nature.com/authors/policies/Policy.pdf>) that verifies compliance with all required editorial policies and a reporting summary (found here <https://www.nature.com/authors/policies/ReportingSummary.pdf>) that collects information on experimental design and reagents. These documents are available to referees to aid the evaluation of the manuscript. Please note that these forms are dynamic 'smart pdfs' and must therefore be downloaded and completed in Adobe Reader. We will then flatten them for ease of use by the reviewers. If you would like to reference the guidance text as you complete the template, please access these flattened versions at <http://www.nature.com/authors/policies/availability.html>.

We strongly recommend the presentation of source data for graphical and statistical analyses as a separate Supplementary Table, and request that source data for all independent repeats are provided when representative experiments of multiple independent repeats, or averages of two independent experiments are presented. This supplementary table should be in Excel format, with data for different figures provided as different sheets within a single Excel file. It should be labelled and numbered as one of the supplementary tables, titled "Statistics Source Data", and mentioned in all relevant figure

legends.

Author Rebuttal to Initial comments

Point-by-point rebuttal to the full referee reports. Chidley et al.

Overall Comments

In response to the referees' comments, we have performed 5 new sets of experiments, added 4 Figure panels and 9 Extended Data Figure panels, and modified many sections of the manuscript. We have further validated the effects of CRISPRi/a to transporter levels by extending our expression profiling across conditions and by measuring transporter plasma membrane levels. We have also strengthened our biological conclusions, especially for SLC7A6/SLC7A7 for arginine/lysine transport and for the role of serotonin/SLC6A4 in ferroptosis. We believe that our revisions adequately address the major comments raised by the reviewers and strengthen our manuscript.

Reviewer #1:

Remarks to the Author:

The metabolic demands of unrestrained proliferation render cancer cells exquisitely dependent on the ability to acquire a variety of nutrients from the tumor microenvironment and, therefore, dependent on transporters that permit nutrient uptake across the cell membrane. Therefore, a thorough understanding of specific nutrient transporters required to support cell proliferation/cell survival has potential to identify novel metabolic vulnerabilities of cancer cells and new targets for cancer therapy. In the current study, Chidley and colleagues describe a platform that permits systematic interrogation of the contributions of individual members of the solute carrier (SLC) and ATP-binding cassette (ABC) transporter superfamilies to nutrient uptake in human cells. The authors employ their CRISPR interference (CRISPRi) and activation (CRISPRa) screening platform to identify critical cellular nutrient transporters, to characterize transporter function and to investigate the impact of the tumor microenvironment on cancer cell metabolism.

This is a very thorough study that reports a powerful platform for investigating the nutrient dependencies of cancer cells. The manuscript describes an elegant series of experiments, and all conclusions are supported by robust data that are clearly presented. As such, there are no specific experimental concerns that need to be addressed. However, a major concern exists with respect to the novelty of the current study. With respect to the screening approach, Rebsamen and colleagues recently reported a SLC-focussed CRISPR gain-of-function screening approach to identify solute carriers able to sustain proliferation under amino acid-deprived conditions (DOI: 10.26508/lsa.202201404) and metabolic CRISPRi/a libraries, which encompass SLC and ABC transporters, have now been employed in a number of in vitro and in vivo studies (e.g., DOI: 10.1016/j.cmet.2020.10.017, DOI: 10.1016/j.molcel.2022.05.024). Moreover, with respect to the authors' claim that they have discovered a novel role for serotonin in the regulation of ferroptosis, Liu and colleagues recently reported this very finding (DOI: 10.1002/adv.202204006). The lack of novelty limits the impact of the current study.

We thank Reviewer #1 for a thorough evaluation of our manuscript and for valuable feedback. Their assessment emphasizes the robustness of our conclusions and the absence of specific experimental concerns, focusing instead on a perceived lack of novelty. Regarding novelty, we acknowledge that certain aspects of our approach and some of our findings have previously been reported, as highlighted by Reviewer #1's references, but this is a strength for a systematic screening study since it helps to validate the approach. Moreover, our study offers a wealth of new observations and contributions that surpass earlier reports in several key aspects.

First and foremost, our manuscript contains the most comprehensive characterization to date of the nutrient transporters necessary to support cell proliferation under a range of conditions, including low amino acid environments, rich media conditions, and xenografts. It's worth noting that while Rebsamen et al. used a gain-of-function SLC-focused library to identify SLCs capable of overcoming nutrient limitations, they did not provide any data on the dependence of cancer cell line on transporters for proliferation, as they lacked knockdown or knockout data. While we acknowledge the existence of loss-of-function and gain-of-function "metabolic" CRISPR libraries encompassing transporters, as used, for example, in studies by the Birsoy lab, maintaining sgRNA representation in vivo is challenging (as mentioned in the reference pointed out by Reviewer #1: 10.1016/j.cmet.2020.10.017). Focusing on transporter-specific libraries, as we have done, allows substantially more robust data to be obtained from in vivo screens.

Second, as a consequence of better representation, our work made novel observations related to transporter activity in living cells, which we meticulously characterized in mass-spec based transport assays, and involving SLC43A1, SLC7A6, and SLC7A7. These (novel) findings were directly related to amino acid transport.

Third, we made the significant observation that serotonin, transported via SLC6A4, exhibits anti-ferroptosis activity. We thank Reviewer #1 for drawing our attention to a recent study by Liu and colleagues that we had missed. Liu and colleagues showed that serotonin acts as a potent chemical antioxidant against lipid peroxides and protects cells from ferroptosis. However, our current study substantially extends their work by showing that expression of SLC6A4 dramatically enhances the response to serotonin. Our findings have several key implications: (i) variability in the effect of serotonin across different cell lines observed by Liu et al. is probably due to the cell line-specific levels of SLC6A4 expression, (ii) serotonin reuptake inhibitors have the potential to modify the ferroptotic sensitivity of cells expressing SLC6A4, and (iii) transport of serotonin across the plasma membrane is necessary for it to exert its anti-ferroptotic potential.

We have updated our manuscript to highlight the recent publication by Liu et al. and to strengthen our claim that serotonin is an endogenous antioxidant that protects cells from ferroptosis.

Reviewer #2:

Remarks to the Author:

A systematic crispr screen was developed to investigate knockdown and overexpression of solute carriers. To enhance the growth phenotype, cells were cultivated in amino acid limiting media. The screen confirms many known transport activities critical for cancer cells such as slc7a5 for BCAA and AAA, slc38a1,2 and slc1a5 for glutamine and slc7a1 and slc7a6 for cationic AA. The screen also reveals some surprising findings, which are poorly discussed but are more worthwhile investigating.

We thank Reviewer #2 for their rigorous evaluation of our study, especially concerning our results related to newly discovered transporter activity in cancer cells.

Line 128 and Fig. 1: Please clarify whether you mean cysteine (cys) or cystine. In the figure the amino acid is listed as cysteine. Please note that cysteine added to media will oxidise to cystine in a matter of hours/days.

We apologize for the lack of clarity related to the identity of the chemical species added in the low cystine screens described in Figure 1, Line 128, and Lines 431-437. The amino acid titrations and the screens described in Figure 1 used media that were prepared using the oxidized form of cysteine, cystine. For

simplification, we used the abbreviation of cysteine (i.e. “Cys”) in our figures to refer to this condition because media made from cysteine or cystine are indistinguishable after a short period of time due to oxidation, as pointed out by the Reviewer and mentioned in our initial draft at lines 431-437. We rectified this imprecision in both the text and figures, explicitly referring to the condition as 'low cystine'.

Fig. 1 Please state explicitly, whether a negative score means the sgRNA is depleted or enriched or better whether a particular mRNA was silenced or overexpressed. Also why are slc38a1 and a2 sgRNA enriched under low arginine and low lysine conditions?

We thank the reviewer for pointing out a lack of clarity. We have incorporated a description of the phenotype score in all relevant panels of Figure 1, clarifying that this score reflects sgRNA depletion/enrichment. However, it is important to note that the score does not indicate whether mRNAs are silenced or overexpressed, as this is determined by whether a CRISPRi or a CRISPRa screen was used.

CRISPRi of SLC38A1 or SLC38A2 in K562 has a slight deleterious effect on growth of K562 cells in RPMI medium, most likely due to the role of these proteins as primary glutamine importers. Because these knockdown cell lines have a lower rate of proliferation, they also have lower general demand for nutrients and therefore become resistant to low nutrient conditions. This effect has been observed previously in genome-wide CRISPRi screens (<https://doi.org/10.7554/eLife.50036>) and manifests as pan-resistance. Such non-specific effects can be easily identified (deleterious effect of knockdown in RPMI + pan-resistance in low nutrient conditions) and are eliminated during our data processing stage, as detailed in the Methods section. To improve the clarity of the text, we have expanded on our description of data processing and have included a note about reduced proliferation rates causing pan-resistance. In the specific case of SLC38A1 and SLC38A2, these transporters were also hypersensitive hits in the low glutamine screen (presumably due to their role as glutamine importers), and as such, these hits were not eliminated during the data processing even though they manifested as non-specific resistant hits in the other screens. To maintain focus on the primary message of Figure 1e, we removed the labels for SLC38A1 and SLC38A2.

Line 186: Should this not read "with one significantly depleted sgRNA?"

We agree that our phrasing in line 186 is confusing and have changed the sentence to “one transporter with significantly depleted sgRNAs”. All gene scores were computed from enrichment/depletion data from 10 individual sgRNAs and scores reflect the average of the top 7 sgRNAs, as outlined in the Methods section.

Line 214: I am not convinced that SLC1A2 transports glutamine. Most likely SLC1A2 transported glutamate, which was converted to glutamine by glutaminase.

We thank the reviewer for identifying this incorrect statement introduced during the final rounds of editing of our initial draft and apologize for any confusion. We agree with the reviewer that the resistance conferred by CRISPRa of SLC1A2 in low glutamine conditions is not due to increased glutamine transport. This is consistent with findings by Rebsamen et al. (DOI: 10.26508/lsa.202201404) who showed that overexpression of SLC1A2 or SLC1A3, both of which are annotated as glutamate/aspartate transporters, conferred a proliferation advantage in HEK293T cells grown in low glutamine. The authors also showed that the proliferation advantage conferred by overexpression of SLC1A2 or SLC1A3 was also present in medium completely devoid of glutamine, indicating that resistance is not caused by upregulation of glutamine import. To fully clear this error, we performed additional experiments in this revision.

We built a K562 SLC1A2 CRISPRa cell line and confirmed strong and specific overexpression of SLC1A2 by RT-qPCR (new Extended Data Figure 6a). We next tested the proliferation rate of K562 SLC1A2 CRISPRa cells compared to an NTC control in low and no glutamine conditions. In agreement with Rebsamen et al., we found that SLC1A2 OE conferred a similar level of resistance in low and no glutamine conditions (new Extended Data Figure 6b), confirming that increased glutamine import is not causing resistance.

Previous reports (<https://doi.org/10.1016/j.cmet.2018.07.005> and <https://doi.org/10.1016/j.cmet.2018.07.021>) have shown that expression of SLC1A3 supports cells by allowing the utilization of aspartate in the absence of glutamine. Under glutamine deprivation, SLC1A3 expression maintains electron transport chain and TCA activity, which promotes de novo glutamate, glutamine, and nucleotide synthesis and rescues cell viability. Our results are consistent with those reports, suggesting that import of aspartate via SLC1A2 might reduce the dependence on exogenous glutamine in low glutamine conditions. Of note, we also identified SLC1A3 OE as a hit in our low glutamine CRISPRa screen and we believe that SLC1A2 and SLC1A3 share a similar resistance mechanism.

Interestingly, when we removed aspartate or glutamate from low glutamine or no glutamine medium, we still observed a strong resistance conferred by OE of SLC1A2 (new Extended Data Figure 6b). Aspartate and glutamate are amongst the 5 amino acids that are net secreted by K562 cells into the medium over time. It is therefore possible that re-import of one (or both) of these exported amino acids could be sufficient to confer resistance in medium without added glutamate or aspartate. Alternatively, it could indicate that aspartate (or glutamate) import is not the cause of the resistance of SLC1A2 OE cells in low or no glutamine. Tajan et al. (<https://doi.org/10.1016/j.cmet.2018.07.005>) observed that aspartate import was not the cause of SLC1A3 OE resistance in HCT116 cells in medium lacking glutamine. The authors suggested that SLC1A3 can localize to the inner mitochondrial membrane and contribute to the malate-aspartate shuttle, which sustains electron transport chain and confers an advantage in low glutamine conditions.

Overall, although these results deserve further work to fully decipher the cause of resistance, they clearly show that, as Reviewer #2 pointed out, a direct increase in glutamine import is certainly not the cause of the resistance of SLC1A2 (and presumably also SLC1A3) in low glutamine CRISPRa screens. We therefore removed SLC1A2 as our example for the interpretation of negative hits and replaced it with SLC7A7. We found that SLC7A7 CRISPRa is only able to transport lysine in growth-limiting conditions (and not any of the other tested amino acids), and this example is supported by solid experimental measurement of amino acid import in screen conditions, that was added during this revision (see next point). For the sake of clarity and because it is beyond the scope of this article, we included a short summary sentence of the SLC1A2 results in the section describing indirect transport effects that mediate resistance (lines 433-442 in current version).

*Line 256: Absence of growth effect in low Arg but not low Lys. This is probably due to the effect that lysine is an essential amino acid while arginine can be synthesized if ASS is expressed. The interpretation that SLC7A7 may not import arginine under growth limiting conditions is not well justified (see Gauthier-Coles, G.; Fairweather, S.J.; Bröer, A.; Bröer, S. Do Amino Acid Antiporters Have Asymmetric Substrate Specificity? *Biomolecules* 2023, 13, 301. <https://doi.org/10.3390/biom13020301>).*

We appreciate the reviewer's concern and have therefore performed additional experiments that support our conclusion as written. In K562 cells, arginine is a conditionally essential amino acid, as its removal from the medium leads to a complete and durable absence of proliferation (as seen in Fig. 1C). While we

did not directly assess arginine biosynthesis through ASS activation, our observations indicate that there is no evidence of such activation, given the consistent selective pressure maintained during the 14-day screen.

Our statement that “OE of SLC7A7 ... resulted in a proliferation advantage in low Lys but not in low Arg” pertains to the relative impact of SLC7A7 overexpression compared to a control. All screens were conducted under conditions where amino acid levels reduced proliferation rates by about 50% (see Supplementary Table S2). This deliberate choice enabled us to identify transporter perturbations that could either hinder or enhance proliferation. The ability to isolate resistant hits in our low arginine CRISPRa screens, including the robust resistant hits SLC7A1, SLC7A2, SLC7A3, and SLC7A6, further supports the hypothesis that cells were unable to synthesize arginine via ASS in this specific condition and require exogenous arginine import.

We acknowledge Reviewer #2's point regarding our interpretation that SLC7A7 does not import arginine under growth-limiting conditions, which was primarily based on interpretation of screen phenotypes rather than on results from direct transport assays. Measuring import rates by GC-MS under such low amino acid conditions, particularly for arginine, where growth-limiting concentrations are approximately 150 times lower than RPMI levels, posed significant technical challenges.

In response to this concern and in order to further substantiate our claims, we were able to assess amino acid import rates under growth-limiting concentrations using LC-MS instead of GC-MS. LC-MS has increased sensitivity for detection of amino acids compared to GC-MS, especially for lysine and arginine. In addition to SLC7A7, we also measured amino acid import rates for SLC7A6 in growth-limiting amino acid concentrations. The results presented in Figure 3C, which is included in the revised version, offer solid experimental confirmation of our claims. In RPMI medium, SLC7A6 or SLC7A7 overexpression increased the import rates of both arginine and lysine, aligning with the known activity of these transporters and our previous assays using GC-MS (Fig. 3B). Under low lysine conditions, we found that CRISPRa of SLC7A6 and SLC7A7 both significantly enhanced lysine import rates. However, in low arginine conditions, CRISPRa of SLC7A6 resulted in a substantial increase in arginine import, while CRISPRa of SLC7A7 did not alter arginine import rates. These changes in amino acid import rates align closely with the growth scores identified in our screens, providing stronger support for our interpretations that screen scores reflect net changes in amino acid import.

Line 270: Please note that dependence on arginine is reduced in the presence of citrulline.

We thank the reviewer for this comment but this does not impact the results of our assays. We agree that the presence of citrulline in physiological conditions could indeed impact the magnitude of the dependence on arginine import from the medium. However, as long as there is a significant difference in proliferation between full arginine and low arginine conditions, a growth phenotype score can be calculated for each gene perturbation. As explained in the Methods section, these phenotype scores represent the effect of the gene perturbation on growth and are normalized to the difference in proliferation between the screen condition (low Arg here) and the control (full Arg). This normalization step cancels out any differences in proliferation of K562 cells in RPMI (no added citrulline) and PAA-RPMI (with citrulline) and ensures that the phenotype scores only reflect the effect of transporter CRISPRi/a on exogenous amino acid import.

Importantly, our transporter CRISPR screens have demonstrated high sensitivity, capable of detecting even subtle changes in growth under mild selection.

Line 274: It is unlikely that a phenotypic score for slc7a8 is observed when slc7a5 is expressed at high levels.

We appreciate Reviewer #2's comment, which aligns with our own observations. However, we believe that the lack of phenotypic scores for SLC7A8, even in the presence of high SLC7A5, offer some interesting insights into the function of SLC7A8 as a large neutral amino acid importer. We have adapted the manuscript to add further context to the SLC7A8 and SLC7A5 findings. These statements are supported by an extension to our expression profiling of members of the SLC7 family (new Extended Data Figure 1d,e).

In these new experiments, we measured the expression levels of SLC7 family genes and CRISPRi/a perturbations of these genes in low Arg, low Leu, and RPMI. Our findings reveal that the basal expression of SLC7A8 in K562 cells grown in RPMI is almost as high as that of SLC7A5. When overexpressed via CRISPRa, SLC7A8 reaches expression levels exceeding those of basal SLC7A5 expression and even surpasses the levels achieved with CRISPRa of SLC7A5. Despite these high levels of expression, CRISPRa and CRISPRi perturbations of SLC7A8 did not have any discernible phenotype in our screens. In contrast, SLC7A5, while already expressed at high levels, demonstrated a robust growth phenotype when its expression was further increased via CRISPRa, accompanied by a substantial increase in amino acid import, demonstrating that transport is not maxed out. Collectively, these data suggest that SLC7A8 does not significantly contribute to import of large neutral amino acids under growth-limiting conditions, despite the transporter being expressed at similar levels as SLC7A5. Furthermore, the observation that low Arg and low Leu conditions result in an upregulation of SLC7A5 expression levels but not SLC7A8 expression aligns with this inference, as depicted in Extended Data Figure 1e, added during this revision. We believe that exploring the contribution of SLC7A8 to large neutral amino acid import in a cell line with low SLC7A5 expression would be an interesting avenue for future investigation.

Line 407: Please note that not all of these amino acids are substrates of SLC43A1 and that reduction of intracellular pools occurs indirectly through exchangers (see Gauthier-Coles, Gregory, et al. "Quantitative modelling of amino acid transport and homeostasis in mammalian cells." Nature communications 12.1 (2021): 5282. Slc43a1 has been characterised as a controller transporter for these reasons.

We appreciate Reviewer #2 for pointing out the publication by Gauthier-Coles et al., which significantly advanced the field's understanding of amino acid transport in mammalian cells. We believe that our research builds upon these robust foundations and facilitates the examination of transporters through perturbation experiments. In our study we chose to focus on SLC43A1 (LAT3) due to its particularly unusual CRISPRi/a phenotypes.

Our findings revealed intriguing results that were unique amongst all other amino acid transporters: CRISPRi of SLC43A1 led to a small but discernable increase in fitness in regular RPMI, while CRISPRa of SLC43A1 resulted in a fitness defect in regular RPMI. In conditions where amino acids were limiting, CRISPRa of SLC43A1 displayed diverging effects, including a growth advantage in low Val and a growth defect in low Leu.

As mentioned in the Gauthier-Coles et al. study, SLC43A1 is classified as a uniporter that can facilitate import or export of amino acids, depending on the conditions. Our results are consistent with SLC43A1 functioning primarily as an exporter of amino acids across the conditions we tested. We concur with Reviewer #2 that not all amino acids experiencing a reduction in intracellular levels are direct substrates of SLC43A1. Instead, we surmise that the substantial reduction in levels observed for many of the large neutral amino acids are direct effects of CRISPRa to SLC43A1, while the smaller reductions in levels observed for other amino acids are likely indirect effects.

While we agree with the characterization of SLC43A1 as a "controller" transporter that regulates amino acid levels to prevent excessive amino acid accumulation by loaders, our data show that SLC43A1 plays

a broader role. The negative impact on growth in RPMI, in low His, low Leu, and low Phe when SLC43A1 is overexpressed clearly suggests significant efflux of amino acids in those conditions.

Line 638: Please provide more guidance and explanation for Fig. 6A/B. If this is to be read in the same way as Fig. 1 a negative score would mean that the corresponding sgRNA were depleted.

We have made modifications to the figure to address this concern and enhance clarity. In Fig. 6A/B, we now explicitly state whether sgRNAs targeting specific transporters in CRISPRi/a screens were depleted or enriched within the tumor isolates. These revisions are in line with the adjustments made in Figure 1, explicitly indicating that negative scores correspond to sgRNA depletion.

Reviewer #3:

Remarks to the Author:

The authors presented a collection of CRISPRi/a screens targeting SLCs and ABCs in K562 and A375 cells, as well as in vivo tumors. The K562 screen data, focusing on the interplay between membrane transporters and amino acids, are potentially useful resources for the community. However, there are two main areas where the manuscript could be improved.

We thank Reviewer #3 for their assessment of our manuscript and for believing that our amino acid screen data could be a useful resource for the community. We have made every effort possible to present the data clearly and have included all screening data in Supplementary Files.

(1) The SLC-targeting CRISPRi/a screens in K562 require additional hit validations and information.

first, a more detailed description of the custom library is needed. It would be helpful to provide the number of SLCs, ABCs, and protein families in the library, and how many of these are expressed in K562 cells. Additionally, the number of plasma membrane transporters and putative amino acid transporters would be relevant.

In response to this concern we have updated the manuscript to provide a substantially more comprehensive description of the custom transporter library. In the new Extended Data Figure 1a, we included a scheme depicting library composition, including the number of SLCs and ABCs, and we provide information on the number of SLC members localized to the plasma membrane, the number of transporters expressed in K562, and the number of putative amino acid transporters. Additionally, we've incorporated this information into Supporting Table S4.

second, because that genes involved in amino acid metabolism are often regulated transcriptionally by amino acid availability, including transporter genes, it would be worthwhile to profile transporter transcript levels under low amino acid CRISPR screen conditions. This could be particularly important for screen hits that might be transcriptionally regulated by low amino acids, independent of CRISPRi/a. The authors have already comments on the GCN2 and mTOR pathways. Other pathways include ATF4 regulated by serine.

We appreciate the reviewer's insightful suggestion regarding the transcriptional regulation of transporter genes under low amino acid conditions, which could potentially impact screen hits independently of CRISPRi/a. In our initial manuscript, we noted a mild activation of GCN2/mTOR pathways in screening

conditions and postulated that this activation would not substantially interfere with the identification of stronger phenotypes resulting from the knockdown or overexpression of specific transporter proteins by CRISPRi/a. In this revision, we have added experimental data to substantiate this claim.

We conducted transcript profiling of our panel of SLC7 transporters using RT-qPCR in low Arg and low Leu screening conditions, comparing them to levels in rich medium, as previously reported in Figure 1b. Furthermore, we expanded the panel of genes profiled to include SLC7A8 (a negative hit) and SLC3A2 (the heavy subunit of many heterodimeric transporters of the SLC7 family) to address the reviewer's concerns. As shown in the new Extended Data Figure 1d,e, low Arg and low Leu screening conditions led to transcriptional upregulation of most profiled transporters, except for SLC7A6 and SLC7A8, which displayed no changes. However, despite these changes in basal transporter expression, CRISPRi/a of individual SLC7 genes resulted in comparable levels of up- and down-regulation compared to a non-targeting control (new Extended Data Figure 1d,e). These data show that the large CRISPRi/a-induced transcriptional changes add to the smaller changes induced by low amino acid levels.

We also included reference to the work of Rebsamen et al. (cited in our original manuscript), who profiled the transcriptional response of transporter genes to single amino acid dropout media in HEK293T cells. Our observations in K562 in low amino acid conditions largely align with their findings, including the upregulation of SLC7A1, SLC7A3, SLC7A5, and SLC3A2, as well as no significant changes in SLC7A6 and SLC7A8 levels. They also noted that many of the upregulated transporter genes were targets of ATF4 across most conditions tested, consistent with the reviewer's comments. We would like to emphasize that ATF4 is not another amino acid sensing pathway; rather, it is an integral component of the GCN2 pathway, serving as the primary transcription factor downstream of GCN2 (e.g. doi: 10.1038/emboj.2010.81, and Broër & Broër, *Biochem J* (2017), cited in our manuscript). Due to the availability of the resource by Rebsamen et al., we chose to focus our efforts on a detailed assessment of transporter levels in our screening conditions, particularly for hits relevant to our study, rather than conducting a family-wide expression profiling. We trust that these new data effectively address the reviewer's concerns.

Plasma membrane transporter hit validation by CRISPRi/a needs to be strengthened. The manuscript should include gene expression data demonstrating the corresponding direction of changes by CRISPRi/a, similar to what was shown for SLC7 family genes.

We would like to respectfully point out that, in addition to the expression profiling of the SLC7 family members featured in Fig 1B, the initial manuscript included expression profiling of changes induced by CRISPRi/a for an additional 10 transporter genes in K562 and in at least another cell line for 3 of these 10 genes. This encompassed SLC43A1 and SLC43A2 in K562 (Extended Data Fig. 5c), SLC6A4 in K562, Caco-2, and A375 cell lines (Extended Data Fig. 6c), SLC7A11 in K562 (Extended Data Fig. 6c), SLC39A9 and SLC39A10 in K562 (Extended Data Fig. 7d), MPC1 and MPC2 in K562 (Extended Data Fig. 7e), SFXN1 in K562 and A375 (Extended Data Fig. 7h), and SLC2A1 in K562 and A375 (Extended Data Fig. 8c).

In this revision, we expanded the panel of SLC7 family genes profiled (SLC7A8 and SLC3A2) and added expression data for CRISPRa of 4 additional genes in K562 cells: SLC1A2, SLC16A10, SLC38A1, and SLC39A13 (new Extended Data Fig. 6a).

Overall, we believe that these data provide robust support to the hypothesis that CRISPRi/a strongly and selectively modulates the expression of target transporter genes. These data also support the hypothesis that any changes to the expression of genes not targeted by CRISPRi/a are most likely not the result of non-specific CRISPRi/a effects, but rather downstream transcriptional consequences of on-target

CRISPRi/a of a specific gene; one example of this being the up-regulation of SLC3A2 in the case of CRISPRa to genes SLC7A5 or SLC7A8 (new Figure 1b and discussed in the next comment).

For a few plasma membrane transporters characterized in detail, it is essential to measure their plasma membrane protein levels and compare them under the CRISPRi/a condition. For heterodimer transporters, the author might consider comment whether the levels of other subunits change upon CRISPRi/a of one subunit, for instance, for the screen hit SLC7 genes and the lack of phenotype for SLC7A8.

To address this concern we have collected new data on the plasma membrane levels of five SLCs. In our initial manuscript, we assessed the effects of CRISPRi/a on transporters by measuring changes in mRNA levels within whole cell extracts by RT-qPCR. Subsequently, we demonstrated that alterations in transporter expression, whether up- or down-regulated, translated into changes in amino acid uptake, as shown in our transport assays. Because of these functional data, we tacitly inferred that protein levels of transporters were altered at the plasma membrane. However, we agree with the reviewer that it would be helpful for readers to explicitly show that this is true for a few select transporters.

In our revised paper, we include a detailed characterization of the plasma membrane levels of transporters SLC7A5, SLC7A6, SLC7A7 and SLC3A2 in new Fig. 1c and Extended Data Fig. 2a, and SLC2A1 in Extended Data Fig. 8d. To isolate the protein fraction localized at the plasma membrane, we incubated cells with a cell-impermeable NHS-ester biotinylation reagent. After extensive washing followed by cell lysis, we isolated biotinylated proteins in pull-downs using streptavidin-agarose beads. Proteins bound to the beads were released by reduction of a disulfide bond included in the linker region of the biotinylation reagent and analyzed by Western blotting. This method allowed us to isolate clean plasma membrane fractions, as judged by the absence of the abundant cytoplasmic protein vinculin (Western blot raw data files).

With respect to the feasibility of measuring transporter levels in membrane preparations, commercial antibodies that perform well in Western blotting are either not available for many proteins or multiple antibody preparations need to be tested (<https://doi.org/10.7554/eLife.91645.1>); this is especially true for transporter proteins (<https://doi.org/10.1016/j.cell.2015.07.022>). We therefore chose to analyze levels of SLC7A5, SLC3A2, and SLC2A1, as good quality antibodies are available for these 3 transporters and have been used in many publications. For SLC7A6 and SLC7A7, we had to screen multiple antibodies in order to identify a preparation that performed well in our Western blots. For SLC7A6, we tested antibodies 13823-1-AP from Proteintech and NBP2-75086 from Novus Biologicals. For SLC7A7, we tested antibodies ab236669 from Abcam, NBP1-59856 from Novus Biologicals, and NBP1-82826 from Novus Biologicals. Furthermore, the elution of proteins from the streptavidin-agarose beads in reducing conditions and with heating led to significant aggregation of transporters, especially SLC7A6, SLC7A7 and SLC2A1. Heat-induced aggregation of transporter proteins has been reported previously (e.g. <https://doi.org/10.1371/journal.pone.0235563>). This aggregation, however, did not prevent an unequivocal assessment of the effect of CRISPRi/a on transporter levels at the plasma membrane.

In the newly added Fig. 1c, we analyzed the effect of CRISPRi/a on SLC7A5, SLC7A6, and SLC7A7 in RPMI conditions. Consistent with RT-qPCR results, we found that CRISPRi specifically reduced the level of SLC7A5 at the membrane, and CRISPRa of SLC7A5, SLC7A6, and SLC7A7 increased the level of the targeted transporter at the membrane. We were unable to clearly assess the effect of CRISPRi to SLC7A6 as basal levels of SLC7A6 were barely detectable, and we did not assess CRISPRi of SLC7A7, as this transporter is not expressed in K562 and accordingly had no phenotype in our screens. As recommended by the reviewer, we also assessed the effect of these perturbations on their heterodimeric

binding partner SLC3A2. Consistent with RT-qPCR results, we observed that overexpression of SLC7A5, SLC7A6, and SLC7A7 led to an increase in plasma membrane levels of SLC3A2. Interestingly, we observed that CRISPRi of SLC7A5 led to a strong depletion of SLC3A2 at the membrane. As we did not see a decrease in SLC3A2 mRNA levels, these results suggest post translational regulation of plasma membrane SLC3A2 levels in the SLC7A5 CRISPRi cell line.

We next analyzed transporter levels in -Leu for SLC7A5, and -Arg for SLC7A6 and SLC7A7, and compared them to levels in RPMI (Extended Data Fig. 2a). For SLC7A5, we saw similar plasma membrane levels of SLC7A5 and its binding partner SLC3A2 in - Leu compared to RPMI. For SLC6A6 and SLC7A7, we saw lower levels of transporter at the plasma membrane in -Arg conditions compared to RPMI.

Finally, we confirmed that plasma membrane levels of SLC2A1 were strongly increased by CRISPRa in both K562 and A375 cells (Extended Data Fig. 8d) growing in regular RPMI conditions.

Overall, these data show that CRISPRi/a of transporter genes potently and specifically changes plasma membrane levels of transporters. Low amino acid conditions lead to a slight upregulation of transporter expression which does not significantly affect plasma membrane levels. Changes induced by low amino acid levels, observed either at the transcript level by RT-qPCR or at the protein level by Western blotting, are minor compared to those induced by CRISPRi/a.

(2) The findings related to the serotonin transporter SLC6A4 are surprising and need further investigation. Will the plasma membrane transporter level increase in low cysteine conditions and/or increases due to CRISPRa. The concentration of 5HT in regular culture media needs to be measured, and are the concentrations used in the study (1uM, 10uM) comparable to physiological levels. A directly measure the effect of 5HT on lipid peroxidation is needed, for example, by using the Bodipy C11 sensor.

As noted by Reviewer #1, Liu et al. have independently reported that serotonin and other tryptophan metabolites act as radical antioxidants, exhibiting anti-ferroptosis activity. Our findings align with these observations, further confirming serotonin's anti-ferroptosis activity. However, our study extends these insights by highlighting the pivotal role of the serotonin transporter, SLC6A4. Given the concordance of certain aspects of our results with this prior study, we focused our revision efforts on the novel contribution of our research, specifically, the involvement of the transporter SLC6A4.

We tested whether expression of SLC6A4 was upregulated in low cystine conditions in K562 cells, that have very low basal SLC6A4 expression, and Caco-2 cells that have relatively high basal levels of SLC6A4. In new Extended Data Figure 6e, we show that SLC6A4 expression is not significantly induced by low cystine in both cell lines. As a positive control, we observed a strong upregulation of SLC7A11 in low cystine, as has previously been observed (doi: 10.3892/ol.2021.13165).

Serotonin is not present in the formulation of RPMI-1640 but can arise from the oxidative degradation of tryptophan or from trace amounts present in dialyzed FBS. The concentration of serotonin is therefore likely to be strongly dependent on the source, age and storage conditions of the tryptophan solution (or solid) used to make medium, and also on the batch of dialyzed FBS used. Furthermore, the level of serotonin in the medium most likely increases over time along with tryptophan oxidization by ambient air. The trace amounts of serotonin in RPMI medium served as a handle to identify the anti-ferroptotic capacity of serotonin transported by SLC6A4 but do not serve a physiological role. Because of the absence of physiological role of serotonin in RPMI, the variability in its level, and the technical difficulty associated with detecting trace amounts, we did not attempt a quantification of the concentration of serotonin in our medium.

The distribution of serotonin in human physiology is highly dependent on the location and is also dynamic. We observed very strong protective effects against ferroptosis at 1 μ M. As noted in our discussion section, serotonin is mostly present in the CNS and in the gut. Concentrations of serotonin have been measured at about 5 μ M in the ileum and colon of mice (<https://doi.org/10.1111/j.1600-079X.2010.00760.x>). Similar levels have been observed in guinea pig guts (<https://doi.org/10.1111/j.1365-2982.2004.00572.x>). The plasma concentration of serotonin is around 10 nM, but can rapidly increase to 10 μ M when platelets become activated at the site of thrombus formation or inflammation (doi: 10.3389/fcvm.2017.00048). We have adapted the text to better convey the physiological relevance of the concentrations of serotonin that we tested.

As recommended by the Reviewer, we used the Bodipy C11 lipid peroxidation sensor to confirm that the deleterious effects we saw on growth were due to the formation of lipid peroxides and subsequent ferroptotic cell death. We were able to strongly induce ferroptosis by incubating cells in medium containing no cystine and with addition of 1 μ M of RSL3, a GPX4 inhibitor. As seen in Figure 4f and Extended Figure 6d added in this revision, we observed that this treatment strongly activated the lipid peroxidation sensor, as expected. Consistent with our hypothesis, we observed that addition of serotonin reduced peroxide levels, dependent on SLC6A4 expression. The results of these lipid peroxidation measurements fully match the results obtained in our viability assays, supporting the hypothesis that serotonin suppresses ferroptosis and expression of SLC6A4 is required to facilitate that effect.

Minor:

In Figure 1c, it would be helpful to list the concentrations of all the amino acids tested, considering that different RPMI formulations may have slightly different amino acid compositions and concentrations. Similarly, in Figures 1c, 1f, and 1g, the exact amino acid concentrations should be included in the main figure.

We added the concentration of each amino acid in the standard RPMI-1640 formulation to the original Figure 1c (Figure 1d in this revision) and the CRISPRi/a screen concentration to original Figures 1f,g (now Fig. 1g,h). We also kept the information in Supplementary Tables S1 and S2, as in our initial draft.

Although not the main focus of the manuscript, the references on recently characterized mitochondrial metabolite transporters should include BCAA (Yoneshiro 2019) and glutathione (Wang 2021 and Shi 2021 [BioRxiv]/2022), as well as other more relevant plasma membrane transporters, heme/SLCO2B1, choline/FLVCR1, and cGAMP/SLC19A1.

We agree with Reviewer #3 that these papers represent a beautiful selection of recent characterizations of the function of orphan transporters and we have modified the manuscript accordingly. The reference to the discovery of the mitochondrial glutathione importer SLC25A39 by the Birsoy lab was included in our initial draft. In our current revision, we added reference to work by the Shen lab that concurrently characterized SLC25A39 as a mitochondrial glutathione importer. We also added a reference to the discovery of FLVCR1 as a plasma membrane choline importer by Kenny et al. In order to keep the number of references in our article within the recommended guidelines, we did not include other examples highlighted by the reviewer.

The data presentation could be improved with a clearer logical flow. For instance, experimental data on SLC7 family transporters are the main focus but are scattered throughout the manuscript with little connection.

In our revised manuscript, we have attempted to improve the logical flow. We start the paper by presenting our approach to determining the transporters of specific nutrients, namely CRISPRi/a screens in conditions where a nutrient is limiting for growth. We test our approach with amino acids in Figure 1. Figures 2 and 3 present the validation of amino acid transporters found in these screens. Figure 4 focuses on the exploration of an unexpected hit from the screens in low cystine medium. In the last two Figures, we extend this work by showing how these libraries and the approach can be used in contexts where nutrients are not artificially limited, namely in various rich media (Figure 5) and in mice (Figure 6). We found many SLC7 transporters in our low amino acid screens and used experimental validation of these hits in Figures 1-3 to convince readers of the power of our approach and to explore novel findings.

Decision Letter, first revision:

Our ref: NCB-RS51313A

17th January 2024

Dear Dr. Chidley,

Thank you for submitting your revised manuscript "A CRISPRi/a screening platform to study cellular nutrient transport in diverse microenvironments" (NCB-RS51313A). It has now been seen by the original Referees #2 and #3 and their comments are below. The reviewers find that the paper has improved in revision, and therefore we'll be happy in principle to publish it in Nature Cell Biology, pending minor revisions to satisfy the referees' final requests and to comply with our editorial and formatting guidelines.

If the current version of your manuscript is in a PDF format, please email us a copy of the file in an editable format (Microsoft Word or LaTeX), as we can not proceed with PDFs at this stage.

We are now performing detailed checks on your paper and will send you a checklist detailing our editorial and formatting requirements in about 1-2 weeks. Please do not upload the final materials and make any revisions until you receive this additional information from us.

Thank you again for your interest in Nature Cell Biology. Please do not hesitate to contact me if you have any questions.

Sincerely,

Melina

Melina Casadio, PhD
Senior Editor, Nature Cell Biology
ORCID ID: <https://orcid.org/0000-0003-2389-2243>

Reviewer #2 (Remarks to the Author):

This is a very large and systematic study of solute carrier knock-downs and overexpression in K562 cells, which includes in vitro and in vivo data. Since this is the revised version of the study, my comments are for consideration by the authors.

Line 111 and Fig. 1c Panel slc7a6 and slc7a7, the cognate molecular weights for these transporters are approx. 37kDa. The triangles appear to be sitting at higher MW.

Line 414: "Overall these data suggest that SLC43A1 and SLC43A2 function as amino acid exchangers". The term exchanger has a specific mechanistic meaning, namely that the uptake of one amino acid is coupled to the efflux of another amino acid. This is not the case here. I would suggest saying that SLC43A1 and A2 can mediate the uptake and efflux of neutral amino acids.

Line 698-719: It would be worth commenting here on SLC16A1/2 in the CRISPRa screen.

Line 739-756: Related to the argument of limiting amount of nutrients is the concept of reserve capacity. If a transporter only needs 10% of its capacity to sustain cell growth CRISPRi has to be >90% to make an observable depletion. This might be worth adding to the discussion.

Line 757-768: It would be fair to refer to Ref. 13 here, where the same argument was made.

Reviewer #3 (Remarks to the Author):

The manuscript detailing transporter CRISPR screens in amino acid conditions is a valuable resource to the community. The authors have made commendable efforts to address my queries experimentally, and I suggest some minor edits to enhance clarity.

- In the context of a resource paper, the screen data results are important and can be presented with enhanced clarity. For example, in Fig. 1f, g, and h, as well as Fig. 5a and d, the framings and arrows appear distracting. The authors might consider removing them and aligning gene names consistently to the respective data points.
- Certain detailed data and their interpretation appear less rigorous, somewhat diminishing the impact of the screen data. Recommend relocate either to the supplemental data or after the screen data presentation. Specifically, in the opening section, while the WB for membrane SLC7A5 and SLC3A2 (Fig. 1c) are convincing, WB for membrane SLC7A6 and SLC7A7 in the CRISPRa cells seems to highlight potential modifications, possibly ubiquitinated and thus unstable transporters. The text associated with Fig. 1b implies transcriptional upregulation of SLC3A2 is expected upon CRISPRa of SLC7A5/A6/A7/A8 (lines 105-108), however, this stabilization effect of subunits within the heterodimers upon expressing the other subunit is anticipated at the protein level – suggest not emphasizing these minor changes and moving the panel to supplemental figure as CRISPR screen setup validation.

Decision Letter, final checks:

Our ref: NCB-RS51313A

22nd January 2024

Dear Dr. Chidley,

Thank you for your patience as we've prepared the guidelines for final submission of your Nature Cell Biology manuscript, "A CRISPRi/a screening platform to study cellular nutrient transport in diverse microenvironments" (NCB-RS51313A). Please carefully follow the step-by-step instructions provided in the attached file, and add a response in each row of the table to indicate the changes that you have made. Please also check and comment on any additional marked-up edits we have proposed within the text. Ensuring that each point is addressed will help to ensure that your revised manuscript can be swiftly handed over to our production team.

We would like to start working on your revised paper, with all of the requested files and forms, as

soon as possible (preferably within two weeks). Please get in contact with us if you anticipate delays.

In recognition of the time and expertise our reviewers provide to Nature Cell Biology's editorial process, we would like to formally acknowledge their contribution to the external peer review of your manuscript entitled "A CRISPRi/a screening platform to study cellular nutrient transport in diverse microenvironments". For those reviewers who give their assent, we will be publishing their names alongside the published article.

Nature Cell Biology offers a Transparent Peer Review option for new original research manuscripts submitted after December 1st, 2019. As part of this initiative, we encourage our authors to support increased transparency into the peer review process by agreeing to have the reviewer comments, author rebuttal letters, and editorial decision letters published as a Supplementary item. When you submit your final files please clearly state in your cover letter whether or not you would like to participate in this initiative. Please note that failure to state your preference will result in delays in accepting your manuscript for publication.

Cover suggestions

COVER ARTWORK: We welcome submissions of artwork for consideration for our cover. For more information, please see our guide for cover artwork.

Nature Cell Biology has now transitioned to a unified Rights Collection system which will allow our Author Services team to quickly and easily collect the rights and permissions required to publish your work. Approximately 10 days after your paper is formally accepted, you will receive an email in providing you with a link to complete the grant of rights. If your paper is eligible for Open Access, our Author Services team will also be in touch regarding any additional information that may be required to arrange payment for your article.

Please note that *Nature Cell Biology* is a Transformative Journal (TJ). Authors may publish their research with us through the traditional subscription access route or make their paper immediately open access through payment of an article-processing charge (APC). Authors will not be required to make a final decision about access to their article until it has been accepted. Find out more about Transformative Journals

Authors may need to take specific actions to achieve compliance with funder and institutional open access mandates. If your research is supported by a funder that requires immediate open access (e.g. according to Plan S principles) then you should select the gold OA route, and we will direct you to the compliant route where possible. For authors selecting the subscription publication route, the journal's standard licensing terms will need to be accepted, including self-

archiving policies. Those licensing terms will supersede any other terms that the author or any third party may assert apply to any version of the manuscript.

Please use the following link for uploading these materials:
[Redacted]

Best regards,

Kendra Donahue
Staff
Nature Cell Biology

On behalf of

Melina Casadio, PhD
Senior Editor, Nature Cell Biology
ORCID ID: <https://orcid.org/0000-0003-2389-2243>

Reviewer #2:

Remarks to the Author:

This is a very large and systematic study of solute carrier knock-downs and overexpression in K562 cells, which includes in vitro and in vivo data. Since this is the revised version of the study, my comments are for consideration by the authors.

Line 111 and Fig. 1c Panel slc7a6 and slc7a7, the cognate molecular weights for these transporters are approx. 37kDa. The triangles appear to be sitting at higher MW.

Line 414: "Overall these data suggest that SLC43A1 and SLC43A2 function as amino acid exchangers". The term exchanger has a specific mechanistic meaning, namely that the uptake of one amino acid is coupled to the efflux of another amino acid. This is not the case here. I would suggest saying that SLC43A1 and A2 can mediate the uptake and efflux of neutral amino acids.

Line 698-719: It would be worth commenting here on SLC16A1/2 in the CRISPRa screen.

Line 739-756: Related to the argument of limiting amount of nutrients is the concept of reserve capacity. If a transporter only needs 10% of its capacity to sustain cell growth CRISPRi has to be >90% to make an observable depletion. This might be worth adding to the discussion.

Line 757-768: It would be fair to refer to Ref. 13 here, where the same argument was made.

Reviewer #3:

Remarks to the Author:

The manuscript detailing transporter CRISPR screens in amino acid conditions is a valuable resource to the community. The authors have made commendable efforts to address my queries experimentally, and I suggest some minor edits to enhance clarity.

- In the context of a resource paper, the screen data results are important and can be presented with enhanced clarity. For example, in Fig. 1f, g, and h, as well as Fig. 5a and d, the framings and arrows appear distracting. The authors might consider removing them and aligning gene names consistently to the respective data points.
- Certain detailed data and their interpretation appear less rigorous, somewhat diminishing the impact of the screen data. Recommend relocate either to the supplemental data or after the screen data presentation. Specifically, in the opening section, while the WB for membrane SLC7A5 and SLC3A2 (Fig. 1c) are convincing, WB for membrane SLC7A6 and SLC7A7 in the CRISPRa cells seems to highlight potential modifications, possibly ubiquitinated and thus unstable transporters. The text associated with Fig. 1b implies transcriptional upregulation of SLC3A2 is expected upon CRISPRa of SLC7A5/A6/A7/A8 (lines 105-108), however, this stabilization effect of subunits within the heterodimers upon expressing the other subunit is anticipated at the protein level – suggest not emphasizing these minor changes and moving the panel to supplemental figure as CRISPR screen setup validation.

Author Rebuttal, first revision:

Point-by-point response to the reviewer comments. Chidley et al. March 2024.

Reviewer #2:

Remarks to the Author:

This is a very large and systematic study of solute carrier knock-downs and overexpression in K562 cells, which includes in vitro and in vivo data. Since this is the revised version of the study, my comments are for consideration by the authors.

We thank Reviewer#2 for their evaluation of our revised work and for these comments

Line 111 and Fig. 1c Panel slc7a6 and slc7a7, the cognate molecular weights for these transporters are approx. 37kDa. The triangles appear to be sitting at higher MW.

As mentioned in our previous cover letter, it is very common for transporter proteins to aggregate during the heat treatment that we used for Western blotting. We agree with both reviewers that this technical difficulty, which does not affect the conclusion that transporters are

present at higher concentration at the plasma membrane, adds an unnecessary source of confusion for the reader. As recommended by Reviewer #3 below, we have moved the panel to an Extended Data Figure. Concerning the MW of SLC7A6 and SLC7A7: these transporters are composed of 515 and 511 amino acids, respectively. Their theoretical MW is around 56-57 kDa. As observed by many others, most transporters aggregate during the heat inactivation step that was performed prior to SDS-PAGE and Western blotting. We have indicated this in our blots.

Line 414: "Overall these data suggest that SLC43A1 and SLC43A2 function as amino acid exchangers". The term exchanger has a specific mechanistic meaning, namely that the uptake of one amino acid is coupled to the efflux of another amino acid. This is not the case here. I would suggest saying that SLC43A1 and A2 can mediate the uptake and efflux of neutral amino acids.

We have corrected the text to remove the term exchanger, which we used incorrectly.

Line 698-719: It would be worth commenting here on SLC16A1/2 in the CRISPRa screen.

In order to keep the word count under 6000 words and preferably closer to 5500, we were unable to include further speculations about these interesting hits.

Line 739-756: Related to the argument of limiting amount of nutrients is the concept of reserve capacity. If a transporter only needs 10% of its capacity to sustain cell growth CRISPRi has to be >90% to make an observable depletion. This might be worth adding to the discussion.

We agree that this is an interesting concept that would be worth discussing further. However, we decided not to include a discussion about this point due to word count concerns.

Line 757-768: It would be fair to refer to Ref. 13 here, where the same argument was made.

We agree with this reviewer. Our revised manuscript did in fact already contain a reference to Ref. 13 at line 760. This reference is also present in this final version.

Reviewer #3:

Remarks to the Author:

The manuscript detailing transporter CRISPR screens in amino acid conditions is a valuable resource to the community. The authors have made commendable efforts to address my queries experimentally, and I suggest some minor edits to enhance clarity.

We thank the reviewer for their evaluation of our revised work and for these comments.

• In the context of a resource paper, the screen data results are important and can be presented with enhanced clarity. For example, in Fig. 1f, g, and h, as well as Fig. 5a and d, the framings

and arrows appear distracting. The authors might consider removing them and aligning gene names consistently to the respective data points.

We improved the clarity of the screen data presentation by removing the framings and arrows, as suggested by this reviewer.

• Certain detailed data and their interpretation appear less rigorous, somewhat diminishing the impact of the screen data. Recommend relocate either to the supplemental data or after the screen data presentation. Specifically, in the opening section, while the WB for membrane SLC7A5 and SLC3A2 (Fig. 1c) are convincing, WB for membrane SLC7A6 and SLC7A7 in the CRISPRa cells seems to highlight potential modifications, possibly ubiquitinated and thus unstable transporters. The text associated with Fig. 1b implies transcriptional upregulation of SLC3A2 is expected upon CRISPRa of SLC7A5/A6/A7/A8 (lines 105-108), however, this stabilization effect of subunits within the heterodimers upon expressing the other subunit is anticipated at the protein level – suggest not emphasizing these minor changes and moving the panel to supplemental figure as CRISPR screen setup validation.

We agree with the reviewer that the aggregation observed in the SLC7A6 and SLC7A7 WBs could be confusing for the reader. We therefore moved those WBs to Extended Data Figure 2. Concerning SLC3A2: we believe that these data are of interest to the reader, so we kept both the RT-qPCR and WB data in Figure 1. Our text (lines 105-108) describes the interesting observation that SLC3A2 transcript levels are upregulated in SLC7A5-8 CRISPRa cell lines. This upregulation does suggest that a mechanism is in place to prevent SLC3A2 levels from limiting functional heterodimer formation. However, we do not make any speculation concerning this mechanism and we merely describe the data.

Final Decision Letter:

Dear Dr Chidley,

I am pleased to inform you that your manuscript, "A CRISPRi/a screening platform to study cellular nutrient transport in diverse microenvironments", has now been accepted for publication in Nature Cell Biology.

Over the next few weeks, your paper will be copyedited to ensure that it conforms to Nature Cell Biology style. Once your paper is typeset, you will receive an email with a link to choose the appropriate publishing options for your paper and our Author Services team will be in touch regarding

any additional information that may be required.

Please note that *Nature Cell Biology* is a Transformative Journal (TJ). Authors may publish their research with us through the traditional subscription access route or make their paper immediately open access through payment of an article-processing charge (APC). Authors will not be required to make a final decision about access to their article until it has been accepted. Find out more about Transformative Journals

To assist our authors in disseminating their research to the broader community, our SharedIt initiative provides you with a unique shareable link that will allow anyone (with or without a subscription) to

read the published article. Recipients of the link with a subscription will also be able to download and print the PDF.

If you have not already done so, we strongly recommend that you upload the step-by-step protocols used in this manuscript to the Protocol Exchange (www.nature.com/protocolexchange), an open online resource established by Nature Protocols that allows researchers to share their detailed experimental know-how. All uploaded protocols are made freely available, assigned DOIs for ease of citation and are fully searchable through nature.com. Protocols and Nature Portfolio journal papers in which they are used can be linked to one another, and this link is clearly and prominently visible in the online versions of both papers. Authors who performed the specific experiments can act as primary authors for the Protocol as they will be best placed to share the methodology details, but the Corresponding Author of the present research paper should be included as one of the authors. By uploading your Protocols to Protocol Exchange, you are enabling researchers to more readily reproduce or adapt the methodology you use, as well as increasing the visibility of your protocols and papers. You can also establish a dedicated page to collect your lab Protocols. Further information can be found at www.nature.com/protocolexchange/about

With kind regards,

Melina Casadio, PhD
Senior Editor, Nature Cell Biology
ORCID ID: <https://orcid.org/0000-0003-2389-2243>
